# Decentralized Projection-free Online Upper-Linearizable Optimization with Applications to DR-Submodular Optimization

## Abstract

We introduce a novel framework for decentralized projection-free optimization, extending projection-free methods to a broader class of upper-linearizable functions. Our approach leverages decentralized optimization techniques with the flexibility of upper-linearizable function frameworks, effectively generalizing traditional DR-submodular function optimization. We obtain the regret of $O(T^{1-\theta/2})$ with communication complexity of $O(T^\theta)$ and number of linear optimization oracle calls of $O(T^{2\theta})$ for decentralized upper-linearizable function optimization, for any $0 \leq \theta \leq 1$. This approach allows for the first results for monotone up-concave optimization with general convex constraints and non-monotone up-concave optimization with general convex constraints. Further, the above results for first order feedback are extended to zeroth order, semi-bandit, and bandit feedback.

## 1 Introduction

Modern machine-learning systems increasingly encounter streaming, non-convex objectives that evolve over time and require decentralized optimization. One prominent challenge is handling adversarial continuous $\gamma$-weakly up-concave functions, which includes Diminishing-Return-Submodular (or DR-submodular) and concave functions. This problem can be formulated as an online interaction: at each time step, the learner selects an action from a convex domain, after which a reward function, satisfying $\gamma$-weak up-concavity, is revealed or partially observed. Depending on the feedback setting, the learner may observe full gradients, only the function value at the selected point, or noisy bandit-style responses. An example is mean-field variational inference on edge devices (Bian et al., 2019), where geographically dispersed sensors cooperatively maximize the evidence lower bound (ELBO) without sharing raw data. The ELBO in this setting is a *continuous DR-submodular objective* on the probability simplex, therefore $\gamma$-weakly up-concave, whose curvature coefficients shift adversarially as fresh observations arrive, making a decentralized, online solution essential. Such formulations also emerge in a wide array of other applications, including price optimization, inventory management, recommendation systems, and power network reconfiguration (Aldrighetti et al., 2021; Mishra et al., 2017; Ito & Fujimaki, 2016; Hassani et al., 2017; Mitra et al., 2021; Gu et al., 2023; Pedramfar et al., 2023; 2024). Recently, Pedramfar & Aggarwal (2024a) introduced the concept of *upper-linearizable* functions, which broadened the classical notions of concavity and DR-submodularity in various contexts. They also proposed a meta-algorithm that transforms linear maximization methods into algorithms capable of optimizing upper-linearizable functions. While the regret bounds of such algorithms have been studied in centralized settings (Pedramfar & Aggarwal, 2024a), we extend this analysis to decentralized setting, which is motivated by applications in multi-agent systems and sensor networks (Li et al., 2002; Xiao et al., 2007; Mokhtari et al., 2018; Duchi et al., 2011).

The objective of this paper is to establish regret bounds for optimizing upper-linearizable functions under different feedback types in a decentralized setup, where communication between agents is limited. In decentralized optimization, each node in the network represents a local decision maker that must select its own decision and subsequently receive a local reward function. The objective of each local decision maker is to minimize its $\alpha$-regret, which is determined by the average of local functions at each iteration. To achieve this, nodes are permitted to communicate with their neighbors and exchange local information. Decentralized optimization has been extensively studied in online convex optimization (Duchi et al., 2011; Sundhar Ram

Table 1: Decentralized Projection-free up-concave maximization

| $F$ | Set | | Feedback | Reference | Appx. | $\log_T(\alpha\text{-Regret})$ | $\log_T(\text{Communication})$ | $\log_T(\text{LOO calls})$ | Range of $\theta$ |
|---|---|---|---|---|---|---|---|---|---|
| Monotone | $0 \in \mathcal{K}$ | $\nabla F$ | full information | (Zhu et al., 2021) | $1-e^{-1}$ | $1/2$ | $5/2$ | $5/2$ | - |
| | | | full information | (Zhang et al., 2023a) | $1-e^{-1}$ | $4/5$ | $1$ | $1$ | - |
| | | | full information | (Liao et al., 2023) | $1-e^{-1}$ | $3/4$ | $1/2$ | $1$ | - |
| | | | | Theorem 2 | $1-e^{-\gamma}$ | $1-\theta/2$ | $\theta$ | $2\theta$ | $[0,1]$ |
| | | | semi-bandit | Theorem 4 | $1-e^{-\gamma}$ | $1-\theta/2$ | $\theta$ | $2\theta$ | $[0,2/3]$ |
| | | $F$ | full information | Theorem 5 | $1-e^{-\gamma}$ | $1-\theta/4$ | $\theta$ | $2\theta$ | $[0,1]$ |
| | | | bandit | Theorem 6 | $1-e^{-\gamma}$ | $1-\theta/4$ | $\theta$ | $2\theta$ | $[0,4/5]$ |
| | general | $\nabla F$ | semi-bandit | Theorem 2 | $\gamma^2/(1+\gamma^2)$ | $1-\theta/2$ | $\theta$ | $2\theta$ | $[0,1]$ |
| | | $F$ | bandit | Theorem 3 | $\gamma^2/(1+\gamma^2)$ | $1-\theta/4$ | $\theta$ | $2\theta$ | $[0,1]$ |
| Non-Mono | general | $\nabla F$ | full information | Theorem 2 | $(1-h)/4$ | $1-\theta/2$ | $\theta$ | $2\theta$ | $[0,1]$ |
| | | | semi-bandit | Theorem 4 | $(1-h)/4$ | $1-\theta/2$ | $\theta$ | $2\theta$ | $[0,2/3]$ |
| | | $F$ | full information | Theorem 5 | $(1-h)/4$ | $1-\theta/4$ | $\theta$ | $2\theta$ | $[0,1]$ |
| | | | bandit | Theorem 6 | $(1-h)/4$ | $1-\theta/4$ | $\theta$ | $2\theta$ | $[0,4/5]$ |

This table compares the different results for the regret for up-concave maximization. Here $h := \min_{\mathbf{z} \in \mathcal{K}} \|\mathbf{z}\|_\infty$. Communication refers to the communication complexity, and LOO calls refers to the number of calls to the Linear Optimization Oracle. Further, the results hold for any $\theta$ in the range specified in the last column.

et al., 2010; Yan et al., 2012; Wang et al., 2022; Zhang et al., 2017; 2023b; Elgabli et al., 2020). Wan et al. (2022) proposed a projection-free distributed algorithm for online convex optimization with sublinear communication complexity. Recently, this problem has been explored within the scope of DR-submodular optimization (Xie et al., 2019; Gao et al., 2021; Zhang et al., 2022; Liao et al., 2023). Specifically, for monotone DR-submodular functions over constraint sets that include the origin, algorithms with regret guarantees have been proposed. Notably, this is just one instance of an upper-linearizable function class proposed by Pedramfar & Aggarwal (2024a), a broader class that encompasses several cases, including (i) monotone $\gamma$-weakly up-concave functions (generalizing DR-submodular functions) over general convex sets, (ii) monotone $\gamma$-weakly up-concave functions over convex sets containing the origin, and (iii) non-monotone up-concave optimization over general convex sets. Therefore, in this work, we focus on decentralized algorithms for the more general class of upper-linearizable functions and provide results applicable to all these cases.

Existing methods for minimizing $\alpha$-regret in DR-submodular optimization typically fall into two broad classes. One class (Chen et al., 2018b; Zhang et al., 2022) uses projection-based strategies which involve computing projections onto the feasible set at each round, while the other class (Chen et al., 2018a;b; Zhang et al., 2019; Liao et al., 2023) leverages projection-free techniques which replace the projection step with linear optimization oracles to update decisions more efficiently. In certain machine learning tasks (Hazan & Luo, 2016; Lacoste-Julien et al., 2013), the computational cost of projection onto a complex feasible set can substantially exceed that of linear optimization. For example, in matrix completion over nuclear norm balls (Chandrasekaran et al., 2009), computing a projection requires full singular value decomposition, which is cubic in matrix size, whereas linear optimization only requires computing a leading singular vector pair. Notably, many of these tasks involve objective functions that are either possess DR-submodular property or admit concave surrogates, both of which fall within the broader class of upper-linearizable functions. This connection motivates us to provide projection-free approaches to the more general upper-linearizable functions under decentralized online setting.

In this paper, we present an algorithm for optimizing upper-linearizable functions in different setups, including monotone functions over convex set containing the origin, monotone functions over general convex set and non-monotone functions over general convex set with different types of feedback including full information first order feedback, full information zeroth order feedback, semi-bandit feedback, and bandit feedback. In particular, for optimizing monotone upper-linearizable functions over convex set containing the origin, the proposed approach achieves a regret bound of $O(T^{1-\theta/2})$ with a communication complexity of $O(T^\theta)$ and number of linear optimization oracle calls of $O(T^{2\theta})$ , where $\theta \in [0,1]$ is a trade-off parameter that controls the blocking effect. When $\theta$ decreases, the total calls to the linear optimization oracle would decrease, while regret and communication complexity would increase. We note that for $\theta = 1$, this achieves the optimal regret of $O(\sqrt{T})$ while having the number of linear optimization oracle calls as $O(T^2)$, while still improving the results in Zhu et al. (2021) reducing the communication complexity and number of linear optimization oracle calls in their special case. Further, for $\theta = 1$, the result matches that by Liao et al. (2023) in their special case

and improves all metrics as compared to the result by Zhang et al. (2023a). The result in this paper gives a tradeoff between the metrics that shows that decreasing regret causes an increase in the communication complexity and the number of linear optimization oracle calls.

The algorithm requires a querying function $\mathcal{G}$, which corresponds to a semi-bandit query when dealing with monotone $\gamma$-weakly up-concave functions over general convex sets (Appendix A.1), and a first-order full-information query in another two cases: monotone $\gamma$-weakly up-concave functions over convex sets containing the origin (Algorithm 6), and non-monotone up-concave optimization over general convex sets (Algorithm 7). By obtaining results for these queries, we adapt the meta-algorithms by Pedramfar & Aggarwal (2024a) to derive new results for different feedback types across various settings, including first-order full information, zeroth-order full information, semi-bandit, and bandit feedback. A summary of the results across these cases is provided in Table 1. It is important to note that prior work only addressed the case of monotone 1-weakly DR-submodular functions over convex sets containing the origin with first-order full-information feedback, while the remaining results in the table are novel contributions to the literature.

We summarize the contributions and technical novelties of this work as follows.

**Contributions:**

1. First result for online monotone DR-submodular maximization over general convex sets and online non-monotone DR-submodular maximization over general convex sets in decentralized setting. Further, the result for monotone DR-submodular with $0 \in \mathcal{K}$ has been extended to $\gamma$-weakly DR-submodular functions while the previous work only considered 1-weakly DR-submodular functions (See Table 1 for the results).

2. Even in the 1-weakly DR-submodular functions with $0 \in \mathcal{K}$, there were results on multiple tradeoff points on regret-communication complexity tradeoff. We combined these results demonstrating a tradeoff between the two objectives (See Table 1 for such comparisons).

3. First results for the more general class of up-concave functions, which includes DR-submodular functions as a subset. DR-submodular functions satisfy $\nabla f(x) \leq \nabla f(y)$ for all $x \geq y$ while up-concave functions only satisfy $\langle \nabla f(x), x - y \rangle \leq \langle \nabla f(y), x - y \rangle$ for all $x \geq y$.

4. First result for online decentralized optimization problem for upper-linearizable functions, a function class much broader than monotone DR-submodular maximization over convex sets containing the origin, giving opportunity to solve problems beyond submodular maximization.

5. A meta-algorithm extending the framework of Pedramfar & Aggarwal (2024a) from centralized to decentralized setting, generating first results for other feedbacks. In total we proposed 10 algorithms, 9 of which are the first in their respective settings, including trivial vs non-trivial queries, first vs zeroth order oracle, which handles different function instances of our upper-linearizable function class. Indeed, the approach allows us to come up with the first results on decentralized DR-submodular optimization with bandit feedback in any of the classes.

**Technical Novelty**

1. We note that previous works on decentralized DR-submodular optimization assumed monotone 1-weakly DR-submodular functions with $0 \in \mathcal{K}$. Thus, we needed a non-trivial approach to extend the setup to include non-monotonicity of the function, $\gamma$-weakly DR-submodular functions, and general convex set constraints. This is done using the notion of upper-linearizable functions, allowing us to obtain the first results for (i) monotone $\gamma$-weakly up-concave functions over convex sets with $0 \in \mathcal{K}$, (ii) monotone up-concave functions with general convex sets, and (iii) non-monotone up-concave functions with general convex sets.

2. We proved that the framework proposed by Pedramfar & Aggarwal (2024a) which was proposed for and examined in the centralized setting, can be extended to the decentralized setting with proper adaptations. The notion of regret changes in the decentralized setting, where we have to consider the average of loss function among all agents instead of just one function, as is the case in a the centralized setting. Note that the changes in the definition of online optimization between centralized and decentralized optimization

makes applications of meta-algorithms used in Pedramfar & Aggarwal (2024a) non-trivial. The centralized version have no notion of communication between nodes and it requires nuance in how one goes about applying meta-algorithms designed for centralized setting to base algorithms that are decentralized.

3. In the cases of monotone functions over convex sets containing the origin and non-monotone functions, the main algorithm, i.e., DOLCO, requires first order full-information feedback. In the centralized setting, the SFTT meta-algorithm, described in Pedramfar & Aggarwal (2024a), is designed to convert the such algorithms into algorithms only requiring semi-bandit feedback. However, this algorithm does not directly work in the decentralized setup. We design Algorithms 3 and 5 using the idea of SFTT meta-algorithm. The challenge here is to ensure that the SFTT blocking mechanism interacts properly with the existing blocking mechanism of the base algorithm.

## 2  Related Works

**Upper-Linearizable Function:** Pedramfar & Aggarwal (2024a) introduced the concept of upper-linearizable functions, a class that generalizes concavity and DR-submodularity across various settings, including both monotone and non-monotone cases over different convex sets. They also explored projection-free algorithms in centralized settings for these setups. Additionally, they proposed several meta-algorithms that adapt the feedback type, converting full-information queries to trivial queries and transitioning from first-order to zeroth-order feedback.

**Decentralized Online DR-Submodular Maximization:** Zhu et al. (2021) introduced the first decentralized online projection-free algorithm for monotone DR-submodular maximization with a constraint set containing the origin, known as the Decentralized Meta-Frank-Wolfe (DMFW). This algorithm provides a $1 - 1/e$ approximation guarantee and achieves an expected regret bound of $O(\sqrt{T})$, where $T$ denotes the time horizon. However, in each round, DMFW requires $T^{3/2}$ stochastic gradient estimates for each local objective function, followed by the transmission of these gradient messages across the network, resulting in significant computational and communication overhead, especially for large $T$ (with a total communication complexity of $O(T^{5/2})$). To address these limitations, Zhang et al. (2023a) proposed the One-shot Decentralized Meta-Frank-Wolfe Algorithm, which achieves $(1 - 1/e)$-regret of $O(T^{4/5})$ while lowering the communication complexity and the number of linear optimization oracle (LOO) calls to $O(T)$. Additionally, Liao et al. (2023) introduced a projection-free online boosting gradient ascent algorithm that achieves $(1 - 1/e)$-regret of $O(T^{3/4})$ with a communication complexity of $O(T^{1/2})$ and number of LOO calls as $O(T)$. On the other hand, the projection-based Decentralized Online Boosting Gradient Ascent algorithm proposed in Zhang et al. (2023a) obtains $(1 - 1/e)$-regret of $O(\sqrt{T})$ with a communication complexity of $O(T)$.

In this paper, we present algorithms for optimizing upper-linearizable functions, achieving a regret bound of $O(T^{1-\theta/2})$, in first order feedback case, and $O(T^{1-\theta/4})$, in zeroth order feedback case, with a communication complexity of $O(T^\theta)$ and number of LOO calls of $O(T^{2\theta})$. In first order full-information case, for $\theta = 1$, we show that the regret and the communication complexity matches the best known projection-based algorithm in Zhang et al. (2023a) and for $\theta = 1/2$, the results match that in Liao et al. (2023) in the special case of monotone 1-weakly DR-submodular functions with the convex set containing the origin.

## 3  Preliminaries

**Notations and Definitions** This paper considers a decentralized setting involving $N$ agents connected over a network represented by an undirected graph $G = (V, E)$, where $V = \{1, \cdots, N\}$ is the set of nodes, and $E \subseteq V \times V$ is the set of edges. Each agent $i \in V$ acts as a local decision maker and can communicate only with its neighbors, defined as $\mathcal{N}_i = \{j \in V \mid (i, j) \in E\} \cup \{i\}$. To model the communication between agents, we introduce a non-negative weight matrix $A = [a_{i,j}] \in \mathbb{R}_+^{N \times N}$, which is supported on the graph $G$. The matrix $A$ is symmetric and doubly stochastic, with $a_{i,j} > 0$ only if $(i, j) \in E$ or $i = j$. Since $A^\top = A$ and $A\mathbf{1} = \mathbf{1}$, where $\mathbf{1}$ denotes the vector of all ones, $\mathbf{1}$ is an eigenvector of $A$ with eigenvalue of 1.

A set $\mathcal{K} \in \mathbb{R}^d$ is convex if $\forall \mathbf{x}, \mathbf{y} \in \mathcal{K}$ and $\forall \alpha \in [0, 1]$, we have $\alpha \mathbf{x} + (1 - \alpha)\mathbf{y} \in \mathcal{K}$. For a constrained set $\mathcal{K}$, we denote the radius of the set as $R \triangleq \max \|\mathbf{x}\|, \forall \mathbf{x} \in \mathcal{K}$. For two vectors $\mathbf{x}, \mathbf{y} \in \mathcal{K}$, we say $\mathbf{x} \leq \mathbf{y}$ if

every element in $\mathbf{x}$ is less than or equal to the corresponding element in $\mathbf{y}$. Given a set $\mathcal{K}$, we define a function class $\mathcal{F}$ as a subset of all real-valued functions over $\mathcal{K}$. For a set $\mathcal{K} \subseteq \mathbb{R}^d$, we define its *affine hull* $\mathrm{aff}(\mathcal{D})$ to be the set of $\alpha\mathbf{x} + (1-\alpha)\mathbf{y}$ for all $\mathbf{x}, \mathbf{y} \in \mathcal{K}$ and $\alpha \in \mathbb{R}$. The *relative interior* of $\mathcal{K}$ is defined as $\mathrm{relint}(\mathcal{D}) := \{\mathbf{x} \in \mathcal{K} \mid \exists r > 0, \mathbb{B}_r(\mathbf{x}) \cap \mathrm{aff}(\mathcal{K}) \subseteq \mathcal{K}\}$.

Given $0 < \gamma \le 1$, we say a differentiable function $f : \mathcal{K} \to \mathbb{R}$ is $\gamma$-*weakly up-concave* if it is $\gamma$-weakly concave along positive directions. Specifically if, $\forall \mathbf{x} \le \mathbf{y}$ in $\mathcal{K}$, we have

$$\gamma\left(\langle \nabla f(\mathbf{y}), \mathbf{y} - \mathbf{x}\rangle\right) \le f(\mathbf{y}) - f(\mathbf{x}) \le \frac{1}{\gamma}\left(\langle \nabla f(\mathbf{x}), \mathbf{y} - \mathbf{x}\rangle\right). \tag{1}$$

We say $\tilde{\nabla} f : \mathcal{K} \to \mathbb{R}^d$ is a $\gamma$-*weakly up-super-gradient* of $f$ if for all $\mathbf{x} \le \mathbf{y}$ in $\mathcal{K}$, the above holds with $\tilde{\nabla}$ instead of $\nabla$. We say $f$ is $\gamma$-weakly up-concave if it is continuous and it has a $\gamma$-weakly up-super-gradient. When it is clear from the context, we simply refer to $\tilde{\nabla} f$ as an up-super-gradient for $f$. A differentiable function $f : \mathcal{K} \to \mathbb{R}$ is called $\gamma$-*weakly continuous DR-submodular* if $\forall \mathbf{x} \le \mathbf{y}$, we have $\nabla f(\mathbf{x}) \ge \gamma\nabla f(\mathbf{y})$. It follows that any $\gamma$-weakly continuous DR-submodular functions is $\gamma$-weakly up-concave.

**Upper-Linearizable Functions** Recently, Pedramfar & Aggarwal (2024a) proposed the notion of upper-linearizable function, which generalizes the notion of up-concave and DR-submodular. The function class $\mathcal{F}$ is upper-linearizable if there exists $\mathfrak{g} : \mathcal{F} \times \mathcal{K} \to \mathbb{R}^d$, $h : \mathcal{K} \to \mathcal{K}$, and constant $0 < \alpha \le 1$ and $\beta > 0$ such that $\forall \mathbf{x}, \mathbf{y} \in \mathcal{K}, \forall f \in \mathcal{F}$:

$$\alpha f(\mathbf{y}) - f(h(\mathbf{x})) \le \beta\left(\langle \mathfrak{g}(f, \mathbf{x}), \mathbf{y} - \mathbf{x}\rangle\right). \tag{2}$$

Pedramfar & Aggarwal (2024a) demonstrated the generality of this class of functions by showing that this includes concave functions as well as up-concave functions in many scenarios. More precisely, this class includes (i) monotone $\gamma$-weakly up-concave functions over general convex sets (Lemma 2), (ii) monotone $\gamma$-weakly up-concave functions over convex sets containing the origin (Lemma 3), and (iii) non-monotone up-concave optimization over general convex sets (Lemma 5). We note that $h(\cdot)$ takes identity function for case (i) and (ii), while $h(\mathbf{x}) = \frac{\mathbf{x} + \bar{\mathbf{x}}}{2}$ for some constant $\bar{\mathbf{x}} \in \mathcal{K}$ for case (iii). The details of these functions with the corresponding query functions, $h(\cdot)$ mapping, $\alpha$ and $\beta$ are provided in Appendix A for completeness.

**Problem Formulation** In a decentralized setting, in $t^{th}$ round, agent $i$ chooses a point $\mathbf{x}_t^i \in \mathcal{K}$, and plays an action $\hat{\mathbf{x}}_t^i = h(\mathbf{x}_t^i)$. The adversary then chooses an upper-linearizable objective function $f_{t,i} \in \mathcal{F}$ and a query oracle $\mathcal{Q}_{t,i}$ for $f_{t,i}$.

Agent $i$ uses a *query function* $\mathcal{G}$ (possibly randomized) that takes, as input, a point $\mathbf{x}_t^i$ and returns a point $\mathbf{w}_t^i \in \mathcal{K}$ at which the agent wants to query and gain information about $f_{t,i}$ [1]. The query function is defined as the function which returns $\mathbf{w}_t^i$ such that $\mathbb{E}[\nabla f(\mathbf{w}_t^i)] = \mathfrak{g}(f_{t,i}, \mathbf{x}_t^i)$. We say the agent query function $\mathcal{G}$ is *trivial* if $\mathcal{G}$ is an identity function ($\mathbf{w}_t^i = \mathbf{x}_t^i$), otherwise we say the agent query function is *non-trivial*. Note that in the three up-concave functions we mention above and in Appendix A, $\mathcal{G}$ is trivial for case (i) and non-trivial for case (ii) and (iii), with $\mathcal{G}$ given by Algorithm 6 for (ii) and Algorithm 7 for (iii), respectively.

The query oracle $\mathcal{Q}_{t,i}$, depending on the feedback type, returns a noisy response $\mathbf{o}_{t,i} = \tilde{\mathfrak{q}}(\mathbf{w}_t^i)$ whose mean is either the gradient (first order feedback) or the value (zeroth order feedback) of $f_{t,i}(\mathbf{w}_t^i)$. We denote $\mathfrak{q}(\mathbf{w}_t^i)$ as the expected value of $\mathcal{Q}_{t,i}(\mathbf{w}_t^i)$. In other words, in the first order feedback case, $\mathfrak{q}(\mathbf{w}_t^i) = \nabla f_{t,i}(\mathbf{w}_t^i)$, and in the zeroth order feedback case $\mathfrak{q}(\mathbf{w}_t^i) = f_{t,i}(\mathbf{w}_t^i)$. For the first order oracle, we use $\tilde{\mathfrak{g}}_{t,i}(\mathbf{x}_t^i) = \tilde{\mathfrak{q}}(\mathbf{w}_t^i)$ to denote the noisy, unbiased estimate of $\mathfrak{g}_{t,i}$ since $\mathbb{E}[\tilde{\mathfrak{g}}_{t,i}(\mathbf{x}_t^i)] = \mathbb{E}[\tilde{\mathfrak{q}}(\mathbf{w}_t^i)] = \mathbb{E}[\nabla f(\mathbf{w}_t^i)] = \mathfrak{g}(f_{t,i}, \mathbf{x}_t^i)$.

The goal for each agent $i$ is to optimize the aggregate function over the network over time: $\sum_{t=1}^T \sum_{j=1}^N f_{t,j}(\mathbf{x}_t^i)$ (Zhu et al., 2021; Zhang et al., 2023a). Thus, with an approximation coefficient $0 < \alpha \le 1$, we define the $\alpha$-*regret* for agent $i$ to be:

$$\mathcal{R}_\alpha^i := \alpha \max_{\mathbf{u} \in \mathcal{K}} \frac{1}{N} \sum_{t=1}^T \sum_{j=1}^N f_{t,j}(\mathbf{u}) - \frac{1}{N} \sum_{t=1}^T \sum_{j=1}^N f_{t,j}(\hat{\mathbf{x}}_t^i).$$

---

[1] In general, the agent may query more than one point. However, in all algorithms considered in this paper, agents only require a single query.

We note that the optimization problem is NP-hard even in the centralized offline setup. For example, when $f$ is a monotone continuous DR-submodular function and $\mathcal{K} \subseteq [0,1]^d$ contains the origin, finding a point $\mathbf{x} \in \mathcal{K}$ such that $f(\mathbf{x})$ is optimal is NP-hard. More generally, for any $\epsilon > 0$, finding any point $\mathbf{x} \in \mathcal{K}$ such that $f(\mathbf{x})$ is at least $(1 - e^{-1} + \epsilon)$ times the optimal value is NP-hard. (See Proposition 3 in Bian et al. (2017)). However, there are polynomial times algorithms that achieve the ratio of $1 - e^{-1}$. The ratio with this property is referred to as the *optimal approximation ratio*, in this case being $1 - e^{-1}$. In the three settings considered in this paper, (in the special case of $\gamma = 1$) the approximation ratios for monotone function over convex sets containing the origin and non-monotone functions over general convex sets are known to be optimal.(See Bian et al. (2017); Mualem & Feldman (2023)) In the case of monotone functions over general convex sets (when $\gamma = 1$), the approximation ratio of $1/2$ is the best known in the literature so far and it is conjectured to be optimal (See Pedramfar et al. (2023)).

We say the agent takes *semi-bandit feedback* if the oracle is first-order gradient feedback on the function and the agent query function is trivial. More formally, the query oracle returns a noisy response with mean of $\nabla f_{t,i}(\mathbf{x}_t^i)$. Similarly, it takes *bandit feedback* if the oracle is zeroth-order and the agent query function is trivial. If the agent query function is non-trivial, then we say the agent requires *full-information feedback*. More formally, first-order full-information feedback returns a noisy response with mean of $\nabla f_{t,i}(\mathbf{w}_t^i)$ and similar for zeroth-order.

We now introduce following assumptions formally.

**Assumption 1.** We assume the action set $\mathcal{K}$ is convex and compact, of which the radius is bounded by $R$, i.e., $R = \max_{\mathbf{x} \in \mathcal{K}} \|\mathbf{x}\|$.

**Assumption 2.** We assume all functions $f \in \mathcal{F} : \mathcal{K} \to \mathbb{R}$ are $M_1$-Lipschitz continuous, differentiable and upper-linearizable with $\alpha, \beta, \mathfrak{g}$ and $h$ as described in Equation 2. We further assume the first order query oracles for $\mathcal{F}$ are bounded by $G$, which implies that, $\|\tilde{\mathfrak{g}}_{t,i}(\mathbf{x})\| \leq G$.

**Assumption 3.** We assume that $\mathfrak{g}(f_{t,i}, \mathbf{x})$ is $L_1$-Lipschitz with respect to the second argument, $\mathbf{x}$.

**Remarks:** We note that the function $\mathfrak{g}$ for the three cases of upper-linearizable functions we introduced above is Lipschitz continuous if the up-concave function is Lipschitz smooth, which is a commonly used assumption in the literature (Liao et al., 2023; Fazel & Sadeghi, 2023; Zhang et al., 2022; 2024). The details of the up-concave functions with the corresponding query oracles is provided in the Supplementary for completeness.

**Assumption 4.** We assume that the second largest eigenvalue of the communication weight matrix $A$, $\lambda_2$, is strictly less than 1.

We also introduce a Lemma for the infeasible projection algorithm.

**Lemma 1** (Lemma 1 in Liao et al. (2023)). *Let $\mathcal{B}$ be the unit ball centered at the origin. There exists an algorithm $\mathcal{O}_{IP}$ referred to as infeasible projection oracle over any convex set $\mathcal{K} \subseteq R\mathcal{B}$ (where $R\mathcal{B}$ means a ball with radius $R$), which takes the set $\mathcal{K}$, a pair of points $(\mathbf{x}_0, \mathbf{y}_0) \in \mathcal{K} \times \mathbb{R}^n$, and an error tolerance parameter $\epsilon$ as the input, and can output*

$$(\mathbf{x}, \tilde{\mathbf{y}}) = \mathcal{O}_{IP}(\mathcal{K}, \mathbf{x}_0, \mathbf{y}_0, \epsilon)$$

*such that $(\mathbf{x}, \tilde{\mathbf{y}}) \in \mathcal{K} \times R\mathcal{B}$, $\|\mathbf{x} - \tilde{\mathbf{y}}\|^2 \leq 3\epsilon$, and $\forall \mathbf{z} \in \mathcal{K}$, $\|\tilde{\mathbf{y}} - \mathbf{z}\|^2 \leq \|\mathbf{y}_0 - \mathbf{z}\|^2$. Moreover, such an oracle $\mathcal{O}_{IP}$ can be implemented with total LOO calls bounded by*

$$\left\lceil \frac{27R^2}{\epsilon} - 2 \right\rceil \max\left(1, \frac{\|\mathbf{x}_0 - \mathbf{y}_0\|^2(\|\mathbf{x}_0 - \mathbf{y}_0\|^2 - \epsilon)}{4\epsilon^2} + 1\right). \tag{3}$$

## 4 Main Result

### 4.1 Algorithm

We introduce a `DOCLO` algorithm which is given in Algorithm 1. The algorithm uses the hyper-parameters $\mathcal{K}$, $\eta$, and $\epsilon$, which will be described in the results. Each agent $i$ in the network stores two variables, decision variable $\mathbf{x}^i \in \mathcal{K}$ and an auxiliary infeasible variable $\tilde{\mathbf{y}}^i$ to help with infeasible projection, initialized in line 2 as $\mathbf{x}_1^i$ and $\tilde{\mathbf{y}}_1^i$ to be some $\mathbf{c} \in \mathcal{K}$. We divide time $T$ into $T/K$ blocks, each of size $K$. Line 3 loops over the

---
**Algorithm 1** Decentralized Online Continuous Upper-Linearizable Optimization - `DOCLO`

---
1: **Input:** decision set $\mathcal{K}$, horizon $T$, block size $K$, step size $\eta$, error tolerance $\epsilon$, number of agents $N$, weight matrix $\mathbf{A} = [a_{ij}]$, query function $\mathcal{G}$, transformation map $h(\cdot)$, first order oracle $\mathcal{Q}$
2: Set $\mathbf{x}_1^i = \tilde{\mathbf{y}}_1^i = \mathbf{c} \in \mathcal{K}$ for any $i = 1, \cdots, N$
3: **for** $m = 1, \cdots, T/K$ **do**
4:     **for** each agent $i = 1, \cdots, N$ in parallel **do**
5:         **for** $t = (m-1)K + 1, \ldots, mK$ **do**
6:             Play $\hat{\mathbf{x}}_m^i = h(\mathbf{x}_m^i)$
7:             Query the oracle $\mathcal{Q}$ at $\mathbf{w}_t^i = \mathcal{G}(\mathbf{x}_t^i)$ and get response $\mathbf{o}_t^i$
8:         **end for**
9:         Communicate $\mathbf{x}_m^i$ and $\tilde{\mathbf{y}}_m^i$ with neighbors
10:         $\mathbf{y}_{m+1}^i \leftarrow \sum_{j \in \mathcal{N}_i} a_{ij} \tilde{\mathbf{y}}_m^j + \eta \sum_{t \in \mathcal{T}_m} \mathbf{o}_t^i$
11:         $(\mathbf{x}_{m+1}^i, \tilde{\mathbf{y}}_{m+1}^i) \leftarrow \mathcal{O}_{IP}(\mathcal{K}, \sum_{j \in \mathcal{N}_i} a_{ij} \mathbf{x}_m^j, \mathbf{y}_{m+1}^i, \epsilon)$
12:     **end for**
13: **end for**

---

blocks, and we denote the set of all time steps in block $m$ as $\mathcal{T}_m$. In each time $t$ inside block $m$, agent $i$ plays $\hat{\mathbf{x}}_t^i = h(\mathbf{x}_m^i)$ (line 6). Then each agent $i$ uses $\mathcal{G}$ to select a point $\mathbf{w}_t^i$, at which it queries the first order oracle $\mathcal{Q}$ and obtains response $\mathbf{o}_t^i$ (line 7). At the end of each block, each agent $i$ communicates $\mathbf{x}_m^i$ and $\tilde{\mathbf{y}}_m^i$ with neighbors (line 9), which are combined with the weight matrix $A$. Thus, each agent $i$ receives $\sum_{j \in \mathcal{N}_i} a_{ij} \tilde{\mathbf{y}}_m^j$ and $\sum_{j \in \mathcal{N}_i} a_{ij} \mathbf{x}_m^j$, where $\mathcal{N}_i$ denotes the neighbors of agent $i$. Based on the information received, each agent $i$ computes $\mathbf{y}_{m+1}^i = \sum_{j \in \mathcal{N}_i} a_{ij} \tilde{\mathbf{y}}_m^j + \eta \sum_{t \in \mathcal{T}_m} \mathbf{o}_t^i$ (line 10), then performs an infeasible projection to compute $(\mathbf{x}_{m+1}^i, \tilde{\mathbf{y}}_{m+1}^i) = \mathcal{O}_{IP}(\mathcal{K}, \sum_{j \in \mathcal{N}_i} a_{ij} \mathbf{x}_m^j, \mathbf{y}_{m+1}^i, \epsilon)$ using the algorithm $\mathcal{O}_{IP}$ described in Lemma 1 (line 11).

## 4.2 Result and Analysis

We will provide the key results for the proposed algorithm, including the regret, communication complexity, and the number of total LOO calls used by the proposed algorithm. The result is given in Theorem 1.

**Theorem 1.** *Algorithm 1 ensures that the $\alpha$-regret for agent $i$ is bounded as*

$$\mathbb{E}\left[R_\alpha^i\right] \leq \beta \left[\frac{2R^2}{\eta} + \frac{18\epsilon T}{\eta K} + 7\eta T K G^2 + 13 T G \sqrt{3\epsilon}\right]$$
$$+ \frac{\beta}{1 - \lambda_2}\left(\frac{12\epsilon T}{\eta K} + 9\eta T K G^2 + 12 T G \sqrt{3\epsilon}\right)$$
$$+ (G + 2L_1 R)(N^{1/2} + 1)\beta \left(3\sqrt{2\epsilon} + \frac{(3\eta K G + 2\sqrt{3\epsilon})}{1 - \lambda_2}\right).$$

*Further, the communication complexity is $O(T/K)$. Finally, the number of LOO calls are upper bounded as $\frac{27TR^2}{\epsilon K}\left(8.5 + 5.5\frac{K^2\eta^2 G^2}{\epsilon} + \frac{K^4\eta^4 G^4}{\epsilon^2}\right)$. In particular, if we set $\epsilon = K^2\eta^2 G^2$, then we have*

$$\mathbb{E}\left[R_\alpha^i\right] = O\left(\frac{1}{\eta} + \eta T K G^2\right),$$

*and the number of LOO calls is $O(\frac{T}{\epsilon K})$.*

*Proof.* The detailed proof of Theorem 1 is provided in the Appendix. For the completeness of argument, we provide a high-level outline of proof for the regret bound and the communication complexity.

**Regret:** Note that instead of a 1-weakly DR-submodular function which has the nice property of $\nabla f(\mathbf{x}) \geq f(\mathbf{y}), \forall \mathbf{x} \leq \mathbf{y}$, we are dealing with upper linearizable functions, a much more generalized function class that includes any function that satisfies $\alpha f(\mathbf{y}) - f(h(\mathbf{x})) \leq \beta \left( \langle \mathfrak{g}(f, \mathbf{x}), \mathbf{y} - \mathbf{x} \rangle \right)$ for some functional $\mathfrak{g}$, function $h$, and constants $0 < \alpha \leq 1$ and $\beta > 0$ as previously described, of which 1-weakly DR-submodular function is an instance.

As is common to bound regret of convex algorithm with first order linear approximation (Orabona, 2019), we bound the regret using the inner product of the distance in action space and $\mathfrak{g}$ function space to approximate the function value, with the help of law of iterated expectation. Let $\mathbf{x}^* \in \mathrm{argmax}_{\mathbf{u} \in \mathcal{K}} \frac{1}{N} \sum_{t=1}^{T} \sum_{i=1}^{N} f_{t,i}(\mathbf{u})$ denote the optimal action, we have

$$\mathbb{E}[\mathcal{R}_\alpha^j] = \frac{1}{N} \sum_{i=1}^{N} \sum_{m=1}^{T/K} \sum_{t \in \mathcal{T}_m} \mathbb{E}\left[ \alpha f_{t,i}(\mathbf{x}^*) - f_{t,i}(h(\mathbf{x}_m^j)) \right]$$

$$\leq \frac{\beta}{N} \sum_{i=1}^{N} \sum_{m=1}^{T/K} \sum_{t \in \mathcal{T}_m} \mathbb{E}\left[ \langle \mathbf{x}^* - \mathbf{x}_m^j, \tilde{\mathfrak{g}}_{t,i}(\mathbf{x}_m^j) \rangle \right]$$

Rearranging terms to better leverage the communication structure of the decentralized network, we have:

$$\frac{1}{\beta} \mathbb{E}[\mathcal{R}_\alpha^j] \leq \mathbb{E}\left[ \frac{1}{N} \sum_{i=1}^{N} \sum_{m=1}^{T/K} \sum_{t \in \mathcal{T}_m} \langle \mathbf{x}^* - \mathbf{x}_m^i, \tilde{\mathfrak{g}}_{t,i}(\mathbf{x}_m^i) \rangle \right]$$

$$+ \mathbb{E}\left[ \frac{1}{N} \sum_{i=1}^{N} \sum_{m=1}^{T/K} \sum_{t \in \mathcal{T}_m} \langle \mathbf{x}_m^i - \mathbf{x}_m^j, \tilde{\mathfrak{g}}_{t,i}(\mathbf{x}_m^i) \rangle \right] \qquad (4)$$

$$+ \mathbb{E}\left[ \frac{1}{N} \sum_{i=1}^{N} \sum_{m=1}^{T/K} \sum_{t \in \mathcal{T}_m} \langle \mathbf{x}^* - \mathbf{x}_m^j, \tilde{\mathfrak{g}}_{t,i}(\mathbf{x}_m^j) - \tilde{\mathfrak{g}}_{t,i}(\mathbf{x}_m^i) \rangle \right]$$

Let the three parts in Equation (4) be respectively $P_1$, $P_2$, $P_3$. Through exploitation of properties of the loss functions and domain, the update rule (line 10) and the infeasible projection operation (line 11) in Algorithm 1, we obtain the upper bound of the expectation of each part:

$$\mathbb{E}\left[P_1\right] \leq \frac{R^2}{\eta} + \frac{18\epsilon T}{\eta K} + 7\eta T K G^2 + 13 T G \sqrt{3\epsilon}$$

$$+ \frac{1}{1 - \lambda_2} \left( \frac{12\epsilon T}{\eta K} + 9\eta T K G^2 + 12 T G \sqrt{3\epsilon} \right)$$

$$\mathbb{E}\left[P_2\right] \leq G(N^{1/2} + 1) \left( 3\sqrt{2\epsilon} + \frac{3\eta K G + 2\sqrt{3\epsilon}}{1 - \lambda_2} \right)$$

$$\mathbb{E}\left[P_3\right] \leq 2 L_1 R(N^{1/2} + 1) \left( 3\sqrt{2\epsilon} + \frac{3\eta K G + 2\sqrt{3\epsilon}}{1 - \lambda_2} \right)$$

Adding $P_1, P_2, P_3$, we obtain the upper bound for $\alpha$-regret for agent $i$ as given in the statement of the Theorem.

**LOO calls:** Based on Lemma 1 for the infeasible projection oracle, we have the number of LOO calls for agent $i$ in block $m$ as:

$$l_m^i = \frac{27 R^2}{\epsilon} \max \left( \frac{1}{4\epsilon^2} (\|\mathbf{y}_{m+1}^i - \sum_{j \in \mathcal{N}_i} a_{ij} \mathbf{x}_m^j\|^2) \right.$$

$$\left. (\|\mathbf{y}_{m+1}^i - \sum_{j \in \mathcal{N}_i} a_{ij} \mathbf{x}_m^j\|^2 - \epsilon) + 1, 1 \right) \qquad (5)$$

Through exploitation of the update rule (line 10) and the infeasible projection operation (line 11) in Algorithm 1, we have

$$
\left\| \mathbf{y}_{m+1}^i - \sum_{j \in \mathcal{N}_i} a_{ij} \mathbf{x}_m^j \right\|^2 \leq 2\eta^2 K^2 G^2 + 6\epsilon \tag{6}
$$

Substituting (6) to (5), we obtain the total LOO calls, $\sum_{m=1}^{T/K} l_m^i$, as in the statement of the Theorem. □

With appropriate selection of parameters block size $K$, update step $\eta$, and infeasible projection error tolerance $\epsilon$, we have final results for the main Algorithm 1 in Theorem 2. Motivated by the trade-off between block size and the time complexity, we introduce a hyper parameter $\theta$, through which users adjust block size accordingly, allowing resilience against practical communication limitations.

**Theorem 2.** *Choosing* $K = T^{1-\theta}$, $\eta = \frac{1}{\sqrt{KT}}$, *and* $\epsilon = K^2 \eta^2$, *we get that for each agent i the*

$$
\mathbb{E}\left[ R_\alpha^i \right] = O(T^{1-\theta/2}) \tag{7}
$$

*Further, the communication complexity is* $O(T^\theta)$ *and the number of LOO calls is* $O(T^{2\theta})$.

We note that in the special case of $\theta = 1$, there will be no block effect, and we achieve a regret of $O(\sqrt{T})$, with a communication complexity of $O(T)$ and number of LOO calls of $O(T^2)$. Further, in the special case of $\theta = 1/2$, we achieve a regret of $O(T^{3/4})$, with a communication complexity of $O(\sqrt{T})$ and number of LOO calls of $O(T)$.

# 5 Extension of the Result to Different Feedback Types for Up-Concave Optimization

In Theorem 1, we provided the result for a query function $\mathfrak{g}$. For the different cases of up-concave function summarized in Table 1, the query oracle $\mathcal{Q}$ is a gradient oracle on the function. Further, the query function $\mathcal{G}$ is trivial in the case of monotone $\gamma$-weakly up-concave functions over general convex sets, thus making the feedback a semi-bandit feedback for the function gradient. In this section, we will consider different extensions for the proposed results, that provide the regret guarantees for the different feedback settings, following the meta-algorithms proposed by Pedramfar & Aggarwal (2024a).

## 5.1 Bandit Feedback for Trivial Query Function

When the query function is trivial, Algorithm 1 has semi-bandit feedback, so we provide an algorithm that uses the semi-bandit to bandit feedback meta-algorithm, STB, by Pedramfar & Aggarwal (2024b) to get the results for bandit feedback. The algorithm designed to handle bandit feedback for upper-linearizable functions with trivial query function is described in Algorithm 2.

Before detailing steps in Algorithm 2, we introduce several mathematical notations that are being used by the STB meta-algorithm. Recall we have defined affine hull and relative interior of set $\mathcal{K}$ in Section 3. We choose a point $\mathbf{c} \in \text{relint}(\mathcal{K})$ and a real number $r > 0$ such that $\text{aff}(\mathcal{K}) \cap \mathbb{B}_r(\mathbf{c}) \subseteq \mathcal{K}$. Then, for any shrinking parameter $0 \leq \delta < r$, we define $\hat{\mathcal{K}}_\delta := (1 - \frac{\delta}{r})\mathcal{K} + \frac{\delta}{r}\mathbf{c}$. For a function $f : \mathcal{K} \to \mathbb{R}$ defined on a convex set $\mathcal{K} \subseteq \mathbb{R}^d$, its $\delta$-smoothed version $\hat{f}_\delta : \hat{\mathcal{K}}_\delta \to \mathbb{R}$ is given as

$$
\begin{aligned}
\hat{f}_\delta(\mathbf{x}) &:= \mathbb{E}_{\mathbf{z} \sim \text{aff}(\mathcal{K}) \cap \mathbb{B}_\delta(\mathbf{x})}[f(\mathbf{z})] \\
&= \mathbb{E}_{\mathbf{v} \sim \mathcal{L}_0 \cap \mathbb{B}_1(\mathbf{0})}[f(\mathbf{x} + \delta\mathbf{v})],
\end{aligned}
$$

where $\mathcal{L}_0 = \text{aff}(\mathcal{K}) - \mathbf{x}$, for any $\mathbf{x} \in \mathcal{K}$, is the linear space that is a translation of the affine hull of $\mathcal{K}$ and $\mathbf{v}$ is sampled uniformly at random from the $k = \dim(\mathcal{L}_0)$-dimensional ball $\mathcal{L}_0 \cap \mathbb{B}_1(\mathbf{0})$. Thus, the function value $\hat{f}_\delta(\mathbf{x})$ is obtained by "averaging" $f$ over a sliced ball of radius $\delta$ around $\mathbf{x}$. For a function class $\mathbf{F}$ over $\mathcal{K}$, we use $\hat{\mathbf{F}}_\delta$ to denote $\{\hat{f}_\delta \mid f \in \mathbf{F}\}$. We will drop the subscript $\delta$ when there is no ambiguity.

The important property of this notion is that it allows for construction of a one-point gradient estimator.

Specifically, it is known[2] that

$$\nabla \hat{f}_\delta(\mathbf{x}) = \frac{k}{\delta} \mathbb{E}_{\mathbf{v} \sim \mathcal{L}_0 \cap \mathbb{S}^1} [f(\mathbf{x} + \delta \mathbf{v})].$$

This allows us to convert Algorithm 1 to allow for zeroth order feedback. Specifically, we run Algorithm 1 on functions $\hat{f}_{t,i}$ instead of $f_{t,i}$ and when it requires an unbiased estimate of the gradient of $\hat{f}_{t,i}(\mathbf{x})$, we use $f_{t,i}(\mathbf{x} + \delta \mathbf{v})$ where $\mathbf{v}$ is sampled uniformly from $\mathcal{L}_0 \cap \mathbb{S}^1$. More generally, if we have access to $o_{t,i}$, an unbiased estimate of $f_{t,i}(\mathbf{x} + \delta \mathbf{v})$, then $\frac{k}{\delta} o_{t,i}$ is an unbiased estimate of $\nabla \hat{f}_{t,i}(\mathbf{x})$. If the zeroth order oracle, from which $\mathbf{o}_{t,i}$ is sampled, is bounded by $B_0$, then we see that this new one-point gradient estimator of $\nabla \hat{f}_{t,i}(\mathbf{x})$ is bounded by $G' := \frac{k}{\delta} B_0$. Therefore, if we set $\epsilon = K^2 \eta^2 (G')^2$, we may use Theorem 1 to see that the regret is bounded by $O\left(\frac{1}{\eta} + \eta T K (G')^2\right)$, and the number of LOO calls is $O(\frac{T}{\epsilon K})$. However, it should be noted that the functions $\hat{f}_{t,i}$ are defined over $\mathcal{K}_\delta$ and this regret is computed against the best point in $\mathcal{K}_\delta$ which can be $O(\delta)$ away from the best point in $\mathcal{K}$. Hence, we see that

$$\mathbb{E}\left[R_\alpha^i\right] = O\left(\frac{1}{\eta} + \eta T K (G')^2 + \delta T\right)$$
$$= O\left(\frac{1}{\eta} + \eta T K \delta^{-2} + \delta T\right).$$

Putting these results together, we obtain the following result, with detailed discussion and proof in Appendix C.

---

**Algorithm 2** Bandit Algorithm for Trivial Query

---

1: **Input:** decision set $\mathcal{K}$, horizon $T$, block size $K$, step size $\eta$, error tolerance $\epsilon$, number of agents $N$, weight matrix $\mathbf{A} = [a_{ij}]$, query function $\mathcal{G}$, transformation map $h(\cdot)$, smoothing parameter $\delta \leq \alpha$, shrunk set $\hat{\mathcal{K}}_\delta$, linear space $\mathcal{L}_0$, zeroth order oracle $\mathcal{Q}$
2: Let $k = \dim(\mathcal{L}_0)$
3: Set $\mathbf{x}_1^i = \tilde{\mathbf{y}}_1^i = \mathbf{c} \in \hat{\mathcal{K}}_\delta$ for any $i = 1, \cdots, N$
4: **for** $m = 1, \cdots, T/K$ **do**
5:     **for** each node $i = 1, \cdots, N$ in parallel **do**
6:         **for** $t = (m-1)K + 1, \ldots, mK$ **do**
7:             Sample $\mathbf{v}_t^i \in \mathbb{S}^1 \cap \mathcal{L}_0$ uniformly
8:             Play $\hat{\mathbf{x}}_t^i = h(\mathbf{x}_m^i) + \delta \mathbf{v}_t^i$
9:             Query the oracle $\mathcal{Q}$ at $\hat{\mathbf{x}}_t^i$ and get response $\mathbf{o}_t^i$
10:            Let $\mathbf{o}_t^i \leftarrow \frac{k}{\delta} \mathbf{o}_t^i \mathbf{v}_t^i$
11:         **end for**
12:         Communicate $\mathbf{x}_m^i$ and $\tilde{\mathbf{y}}_m^i$ with neighbors
13:         $\mathbf{y}_{m+1}^i \leftarrow \sum_{j \in \mathcal{N}_i} a_{ij} \tilde{\mathbf{y}}_m^j + \eta \sum_{t \in \mathcal{T}_m} \mathbf{o}_t^i$
14:         $(\mathbf{x}_{m+1}^i, \tilde{\mathbf{y}}_{m+1}^i) \leftarrow O_{IP}\left(\hat{\mathcal{K}}_\delta, \sum_{j \in \mathcal{N}_i} a_{ij} \mathbf{x}_m^j, \mathbf{y}_{m+1}^i, \epsilon\right)$
15:     **end for**
16: **end for**

---

**Theorem 3.** *If $\mathcal{G}$ is trivial, and the zeroth order oracle is bounded and we set $\epsilon = K^2 \eta^2 \delta^{-2}$, then Algorithm 2 ensures a regret bound of*

$$\mathbb{E}\left[R_\alpha^i\right] = O\left(\frac{1}{\eta} + \eta T K \delta^{-2} + \delta T\right),$$

*with at most $O(\frac{T}{\epsilon K})$ LOO calls and $O(T/K)$ communication complexity. In particular, if we set $K = T^{1-\theta}$, $\delta = T^{-\theta/4}$ and $\eta = \frac{\delta}{\sqrt{KT}}$, we see that $\mathbb{E}\left[R_\alpha^i\right] = O(T^{1-\theta/4})$ with at most $O(T^{2\theta})$ LOO calls and $O(T^\theta)$ communication complexity.*

---

[2]When $k = d$ and therefore $\mathcal{L}_0 = \mathbb{R}^d$, this equality is well known, e.g. see Nemirovskiĭ & Ĭûdin (1983); Flaxman et al. (2005). The more general case where $k \leq d$ may be found in Remark 4 in Pedramfar et al. (2023).

## 5.2 Semi-Bandit Algorithm for Non-Trivial Query Function

When the query function is non-trivial, Algorithm 1 has first order full-information feedback. To have the algorithm with semi-bandit feedback, we use the idea used by the meta-algorithm "Stochastic Full-information To Trivial query" SFTT, proposed by Pedramfar & Aggarwal (2024a) to convert Algorithm 1 to one that can work with semi-bandit feedback. Note that SFTT itself uses a blocking mechanism and we must ensure that its blocking mechanism interacts properly with both the DOLCO blocking mechanism and the inter-node communication. The resulting algorithm, which is designed to handle semi-bandit feedback for upper-linearizable functions with non-trivial query, is summarized in Algorithm 3.

Let $L \geq 1$ be an integer. The main idea here is to consider the functions $(\bar{f}_{q,i})_{1 \leq q \leq T/L, 1 \leq i \leq N}$ where $\bar{f}_{q,i} = \frac{1}{L} \sum_{t=(q-1)L+1}^{qL} f_{t,i}$. We want to run Algorithm 1 against this sequence of functions. To do this, we need to construct unbiased estimates of the gradient of $\bar{f}_{q,i}$. This can be achieved by considering a random permutation $t'_1, \cdots, t'_L$ of $(q-1)L+1, \cdots, qL$ and picking $f_{t'_1, i}$. Since we want an algorithm with semi-bandit feedback, at time-step $t'_1$ we select the point where the original algorithm, i.e., Algorithm 1, needed to query. In the other $L-1$ time-steps within this block, we pick the action that Algorithm 1 wants to take and ignore the returned value of the query function. Thus, at one time-step per each block of length $L$, we have no control over the regret, which adds $O(T/L)$ to the total regret. In the remaining time-steps, the behavior is similar to Algorithm 1, with each action repeated $L-1$ times. We note that we are running Algorithm 1 against $\bar{f}_{q,i}$ with a horizon of $T' := T/L$. Hence, using the discussion above and Theorem 1, if we set $\epsilon = K^2 \eta^2 G^2$, then we see that the regret is bounded by

$$\mathbb{E}\left[R_\alpha^i\right] = (L-1)O\left(\frac{1}{\eta} + \eta T' K G^2\right) + O\left(\frac{T}{L}\right)$$

$$= O\left(\frac{L}{\eta} + \eta T K G^2 + \frac{T}{L}\right).$$

the number of LOO calls is $O(\frac{T'}{\epsilon K})$, and the communication complexity is bounded by $O(T'/K)$.

The key result is summarized in the following theorem, and the detailed discussion and proof can be found in Appendix D.

**Theorem 4.** *If $\mathcal{G}$ is non-trivial, and we set $\epsilon = K^2 \eta^2 G^2$, then Algorithm 3 ensures a regret bound of*

$$\mathbb{E}\left[R_\alpha^i\right] = O\left(\frac{L}{\eta} + \eta T K G^2 + \frac{T}{L}\right).$$

*with at most $O(\frac{T}{\epsilon K L})$ LOO calls and $O(\frac{T}{K L})$ communication complexity. In particular, if $0 \leq \theta \leq 2/3$ and we set $K = T^{1-3\theta/2}, L = T^{\theta/2}$, and $\eta = T^{\theta-1}$, we see that $\mathbb{E}\left[R_\alpha^i\right] = O(T^{1-\theta/2})$ with at most $O(T^{2\theta})$ LOO calls and $O(T^\theta)$ communication complexity.*

## 5.3 Zeroth Order Feedback for Non-Trivial Query Function

When the query function is non-trivial, Algorithm 1 has first order full-information feedback. To have the algorithm with zeroth order feedback, we use the "First Order To Zeroth Order" meta-algorithm, FOTZO, proposed by Pedramfar & Aggarwal (2024b) to obtain the result. The algorithm designed to handle zeroth-order full-information feedback for upper-linearizable functions with non-trivial query is summarized in Algorithm 4. The discussion and analysis of this meta-algorithm is quite similar to that of STB, and the details are provided in Appendix E.

**Theorem 5.** *If $\mathcal{G}$ is non-trivial, and the zeroth order oracle is bounded and we set $\epsilon = K^2 \eta^2 \delta^{-2}$, then Algorithm 4 ensures a regret bound of*

$$\mathbb{E}\left[R_\alpha^i\right] = O\left(\frac{1}{\eta} + \eta T K \delta^{-2} + \delta T\right),$$

*with at most $O(\frac{T}{\epsilon K})$ LOO calls and $O(T/K)$ communication complexity. In particular, if we set $K = T^{1-\theta}$, $\delta = T^{-\theta/4}$ and $\eta = \frac{\delta}{\sqrt{KT}}$, we see that $\mathbb{E}\left[R_\alpha^i\right] = O(T^{1-\theta/4})$ with at most $O(T^{2\theta})$ LOO calls and $O(T^\theta)$ communication complexity.*

---

**Algorithm 3** Semi-Bandit Algorithm for Non-trivial Query

---

1: **Input:** decision set $\mathcal{K}$, horizon $T$, `DOCLO` block size $K$, step size $\eta$, error tolerance $\epsilon$, number of agents $N$, weight matrix $\mathbf{A} = [a_{ij}]$, query function $\mathcal{G}$, map $h(\cdot)$, `SFTT` block size $L > 1$, first order oracle $\mathcal{Q}$
2: Set $\mathbf{x}_1^i = \tilde{\mathbf{y}}_1^i = \mathbf{c} \in \mathcal{K}$ for any $i = 1, \cdots, N$
3: **for** $m = 1, \cdots, \frac{T}{LK}$ **do**
4:     **for** each node $i = 1, \cdots, N$ in parallel **do**
5:         **for** q $= (m-1)K + 1, \cdots, mK$ **do**
6:             Play $\hat{\mathbf{x}}_q^i = h(\mathbf{x}_m^i)$
7:             Let $\mathbf{w}_q^i$ be the point returned by $\mathcal{G}(\mathbf{x}_m^i)$
8:             Sample $t_q'$ uniformly from $\{(q-1)L + 1, \ldots, qL\}$
9:             **for** $t = (q-1)L + 1, \ldots, qL$ **do**
10:                 **if** $t = t_q'$ **then**
11:                     Play the action $\mathbf{z}_t^i = \mathbf{w}_q^i$
12:                     Query the oracle $\mathcal{Q}$ at $\mathbf{w}_q^i$ and get response $\mathbf{o}_q^i$
13:                 **else**
14:                     Play the action $\mathbf{z}_t^i = \hat{\mathbf{x}}_q^i$
15:                 **end if**
16:             **end for**
17:         **end for**
18:         Communicate $\mathbf{x}_m^i$ and $\tilde{\mathbf{y}}_m^i$ with neighbors
19:         $\mathbf{y}_{m+1}^i \leftarrow \sum_{j \in \mathcal{N}_i} a_{ij} \tilde{\mathbf{y}}_m^j + \eta \sum_{q=(m-1)K+1}^{mK} \mathbf{o}_q^i$
20:         $(\mathbf{x}_{m+1}^i, \tilde{\mathbf{y}}_{m+1}^i) \leftarrow \mathcal{O}_{IP}(\mathcal{K}, \sum_{j \in \mathcal{N}_i} a_{ij} \mathbf{x}_m^j, \mathbf{y}_{m+1}^i, \epsilon)$
21:     **end for**
22: **end for**

---

**Algorithm 4** Zeroth Order Algorithm for Non-Trivial Query

---

1: **Input:** decision set $\mathcal{K}$, horizon $T$, block size $K$, step size $\eta$, error tolerance $\epsilon$, number of agents $N$, weight matrix $\mathbf{A} = [a_{ij}]$, query function $\mathcal{G}$, transformation map $h(\cdot)$, smoothing parameter $\delta \leq \alpha$, shrunk set $\hat{\mathcal{K}}_\delta$, linear space $\mathcal{L}_0$, zeroth order oracle $\mathcal{Q}$
2: Let $k = \dim(\mathcal{L}_0)$
3: Set $\mathbf{x}_1^i = \tilde{\mathbf{y}}_1^i = \mathbf{c} \in \hat{\mathcal{K}}_\delta$ for any $i = 1, \cdots, N$
4: **for** $m = 1, \cdots, T/K$ **do**
5:     **for** each node $i = 1, \cdots, N$ in parallel **do**
6:         **for** $t = (m-1)K + 1, \ldots, mK$ **do**
7:             Play $h(\mathbf{x}_m^i)$
8:             Sample $\mathbf{v}_t^i \in \mathbb{S}^1 \cap \mathcal{L}_0$ uniformly
9:             Let $\mathbf{w}_t^i$ be the point returned by $\mathcal{G}(\mathbf{x}_m^i)$
10:             Query the oracle $\mathcal{Q}$ at $\mathbf{w}_t^i + \delta \mathbf{v}_t^i$ and get response $\mathbf{o}_t^i$
11:             Let $\mathbf{o}_t^i \leftarrow \frac{k}{\delta} \mathbf{o}_t^i \mathbf{v}_t^i$
12:         **end for**
13:         Communicate $\mathbf{x}_m^i$ and $\tilde{\mathbf{y}}_m^i$ with neighbors
14:         $\mathbf{y}_{m+1}^i \leftarrow \sum_{j \in \mathcal{N}_i} a_{ij} \tilde{\mathbf{y}}_m^j + \eta \sum_{t \in \mathcal{T}_m} \mathbf{o}_t^i$
15:         $(\mathbf{x}_{m+1}^i, \tilde{\mathbf{y}}_{m+1}^i) \leftarrow \mathcal{O}_{IP}(\hat{\mathcal{K}}_\delta, \sum_{j \in \mathcal{N}_i} a_{ij} \mathbf{x}_m^j, \mathbf{y}_{m+1}^i, \epsilon)$
16:     **end for**
17: **end for**

---

### 5.4 Bandit Algorithm for Non-Trivial Query Function

In this case, we apply the Stochastic Full-information To Trivial query meta-algorithm, `SFTT`, discussed above, to convert Algorithm 4, which requires zeroth order full-information feedback, to an algorithm that works with bandit feedback. The resulting algorithm, given in Algorithm 5, can handle bandit feedback for upper-linearizable functions with non-trivial query function.

Similar to the discussion in Section 5.2, we see that the regret bound is equal to $O(T/L)$ plus $L-1$ times the regret bound of Algorithm 4 when applied over a horizon of $T/L$. In other words, if $T' = T/L$ and $\epsilon = K^2\eta^2\delta^{-2}$, then we have

$$\mathbb{E}\left[R_\alpha^i\right] = (L-1)O\left(\frac{1}{\eta} + \eta T'K\delta^{-2} + \delta T'\right) + O\left(\frac{T}{L}\right)$$

$$= O\left(\frac{L}{\eta} + \eta TK\delta^{-2} + \delta T + \frac{T}{L}\right).$$

Hence, we obtain the following result, with detailed discussion and proof in Appendix F.

---

**Algorithm 5** Bandit Algorithm for Non-trivial Query

1: **Input:** decision set $\mathcal{K}$, horizon $T$, `DOCLO` block size $K$, step size $\eta$, error tolerance $\epsilon$, number of agents $N$, weight matrix $\mathbf{A} = [a_{ij}]$, query function $\mathcal{G}$, transformation map $h(\cdot)$, `SFTT` block size $L > 1$, smoothing parameter $\delta \leq \alpha$, shrunk set $\hat{\mathcal{K}}_\delta$, linear space $\mathcal{L}_0$, zeroth order oracle $\mathcal{Q}$
2: Let $k = \dim(\mathcal{L}_0)$
3: Set $\mathbf{x}_1^i = \tilde{\mathbf{y}}_1^i = \mathbf{c} \in \hat{\mathcal{K}}_\delta$ for any $i = 1, \cdots, N$
4: **for** $m = 1, \cdots, T/LK$ **do**
5:   **for** each node $i = 1, \cdots, N$ in parallel **do**
6:     **for** q $= 1, 2, \ldots, K$ **do**
7:       Let $\hat{\mathbf{x}}_q^i = h(\mathbf{x}_m^i)$
8:       Sample $\mathbf{v}_q^i \in \mathbb{S}^1 \cap \mathcal{L}_0$ uniformly
9:       Let $\mathbf{w}_q^i$ be the point returned by $\mathcal{G}(\mathbf{x}_m^i)$
10:      Let $\hat{\mathbf{w}}_q^i = \mathbf{w}_q^i + \delta\mathbf{v}_q^i$
11:      Sample $t_q'$ uniformly from $\{(m-1)KL + (q-1)L + 1, \ldots, (m-1)KL + qL\}$
12:      **for** $t = (m-1)KL + (q-1)L + 1, \ldots, (m-1)KL + qL$ **do**
13:       **if** $t = t_q'$ **then**
14:         Play the action $\mathbf{z}_t = \hat{\mathbf{w}}_q^i$
15:         Query the oracle $\mathcal{Q}$ at $\hat{\mathbf{w}}_q^i$ and get response $\mathbf{o}_q^i$
16:         Let $\mathbf{o}_q^i \leftarrow \frac{k}{\delta}\mathbf{o}_q^i\mathbf{v}_q^i$
17:       **else**
18:         Play the action $\mathbf{z}_t = \hat{\mathbf{x}}_q^i$
19:       **end if**
20:      **end for**
21:     **end for**
22:     Communicate $\mathbf{x}_m^i$ and $\tilde{\mathbf{y}}_m^i$ with neighbours
23:     $\mathbf{y}_{m+1}^i \leftarrow \sum_{j \in \mathcal{N}_i} a_{ij}\tilde{\mathbf{y}}_m^j + \eta \sum_{q=1}^K \mathbf{o}_q^i$
24:     $(\mathbf{x}_{m+1}^i, \tilde{\mathbf{y}}_{m+1}^i) \leftarrow \mathcal{O}_{IP}(\hat{\mathcal{K}}_\delta, \sum_{j \in \mathcal{N}_i} a_{ij}\mathbf{x}_m^j, \mathbf{y}_{m+1}^i, \epsilon)$
25:   **end for**
26: **end for**

---

**Theorem 6.** *If $\mathcal{G}$ is non-trivial, and the zeroth order oracle is bounded and we set $\epsilon = K^2\eta^2\delta^{-2}$, then Algorithm 5 ensures a regret bound of*

$$\mathbb{E}\left[R_\alpha^i\right] = O\left(\frac{L}{\eta} + \eta TK\delta^{-2} + \delta T + \frac{T}{L}\right),$$

*with at most $O(\frac{T}{\epsilon KL})$ LOO calls and $O(\frac{T}{KL})$ communication complexity. In particular, if $0 \leq \theta \leq 4/5$ and we set $K = T^{1-5\theta/4}$, $\delta = T^{-\theta/4}$, $L = T^{\theta/4}$, and $\eta = T^{\theta/2-1}$, we see that $\mathbb{E}\left[R_\alpha^i\right] = O(T^{1-\theta/4})$ with at most $O(T^{2\theta})$ LOO calls and $O(T^\theta)$ communication complexity.*

## 6 Conclusions

In this paper, we presented a decentralized, projection-free approach to optimizing upper-linearizable functions, which extends the analysis of classical DR-submodular and concave functions. By incorporating projection-free methods, our framework provides efficient regret bounds of $O(T^{1-\theta/2})$, in first order feedback case, $O(T^{1-\theta/4})$, in zeroth order feedback case, with a communication complexity of $O(T^\theta)$ and number of linear optimization oracle calls of $O(T^{2\theta})$ for suitable choices of $0 \leq \theta \leq 1$, making it scalable for large decentralized networks. This illustrates a tradeoff between the regret and the communication complexity. The versatility of our approach allows it to handle a variety of feedback models, including full information, semi-bandit, and bandit settings. This is the first known result that provides such generalized guarantees for monotone and non-monotone up-concave functions over general convex sets.

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

# A   Up-concave Functions are Linearizable

In this section, we provide for completeness that three cases of common up-concave functions can be formulated as upper-linearizable functions, and we give the exact query algorithm to obtain estimates of $\mathfrak{g}$ functions. We further show that $\mathfrak{g}$ given by these algorithms is Lipschitz-continuous, if $f$ is $L$-smooth.

## A.1   Monotone Up-concave optimization over general convex set

Following Lemma 1 by Pedramfar & Aggarwal (2024a), we note that monotone up-concave optimization over general convex set can be formulated as an online maximization by quantization algorithm with trivial query $\mathcal{G}(\mathbf{x}) = \mathbf{x}$.

**Lemma 2** (Pedramfar & Aggarwal (2024a)). *Let $f : [0,1]^d \to \mathbb{R}$ be a a non-negative monotone $\gamma$-weakly up-concave function with curvature bounded by $c$. Then, for all $\mathbf{x}, \mathbf{y} \in [0,1]^d$, we have*

$$\frac{\gamma^2}{1 + c\gamma^2} f(\mathbf{y}) - f(\mathbf{x}) \le \frac{\gamma}{1 + c\gamma^2} (\langle \tilde{\nabla} f(\mathbf{x}), \mathbf{y} - \mathbf{x} \rangle),$$

*where $\tilde{\nabla} f$ is an up-super-gradient for $f$.*

## A.2   Monotone up-concave optimization over convex set containing the origin

Following Lemma 2 by Pedramfar & Aggarwal (2024a), we note that monotone up-concave optimization over convex set containing the origin can be formulated as an online maximization by quantization algorithm with non-trivial query $\mathcal{G} = \texttt{BQMO}$, which is described in Algorithm 6.

**Lemma 3** (Pedramfar & Aggarwal (2024a)). *Let $f : [0,1]^d \to \mathbb{R}$ be a non-negative monotone $\gamma$-weakly up-concave differentiable function and let $F : [0,1]^d \to \mathbb{R}$ be the function defined by*

$$F(\mathbf{x}) := \int_0^1 \frac{\gamma e^{\gamma(z-1)}}{(1 - e^{-\gamma})z} (f(z * \mathbf{x}) - f(\mathbf{0})) dz.$$

*Then $F$ is differentiable and, if the random variable $\mathcal{Z} \in [0,1]$ is defined by the law*

$$\forall z \in [0,1], \quad \mathbb{P}(\mathcal{Z} \le z) = \int_0^z \frac{\gamma e^{\gamma(u-1)}}{1 - e^{-\gamma}} du, \tag{8}$$

*then we have $\mathbb{E}\left[\nabla f(\mathcal{Z} * \mathbf{x})\right] = \nabla F(\mathbf{x})$. Moreover, we have*

$$(1 - e^{-\gamma}) f(\mathbf{y}) - f(\mathbf{x}) \le \frac{1 - e^{-\gamma}}{\gamma} \langle \nabla F(\mathbf{x}), \mathbf{y} - \mathbf{x} \rangle.$$

---

**Algorithm 6** Boosted Query Algorithm for Monotone up-concave functions over convex sets containing the origin – BQMO

---
1: **Input:** point $\mathbf{x}$
2: Sample $z \in [0,1]$ according to Equation (8) in Lemma 3
3: **Return:** $z * \mathbf{x}$

---

In this case, $\mathfrak{g} = \nabla F(\mathbf{x})$, $h(\mathbf{x}) = \text{Id}$. Further, if $f$ is smooth, $\mathfrak{g}$ is Lipschitz continuous, as shown in the following Lemma.

**Lemma 4** (Theorem 2(iii), Zhang et al. (2022)). *If $f$ is $L$-smooth and satisfies other assumptions of Lemma 3, $F$ is $L'$-smooth, where $L' = L \frac{\gamma + e^{-\gamma} - 1}{\gamma(1 - e^{-\gamma})}$, i.e., $\nabla F(\mathbf{x})$ is $L'$-Lipschitz continuous.*

### A.3 Non-monotone up-concave optimization over general convex set

Following Lemma 3 by Pedramfar & Aggarwal (2024a), we note that Non-monotone up-concave optimization over general convex set can be formulated as an online maximization by quantization algorithm with non-trivial query $\mathcal{G} = \texttt{BQN}$, as described in Algorithm **??**.

**Lemma 5** (Pedramfar & Aggarwal (2024a)). *Let $f : [0, 1]^d \to \mathbb{R}$ be a non-negative continuous up-concave differentiable function and let $\underline{\mathbf{x}} \in \mathcal{K}$. Define $F : [0, 1]^d \to \mathbb{R}$ as the function*

$$F(\mathbf{x}) := \int_0^1 \frac{2}{3z(1 - \frac{z}{2})^3} \left( f \left( \frac{z}{2} * (\mathbf{x} - \underline{\mathbf{x}}) + \underline{\mathbf{x}} \right) - f(\underline{\mathbf{x}}) \right) dz,$$

*Then $F$ is differentiable and, if the random variable $\mathcal{Z} \in [0, 1]$ is defined by the law*

$$\forall z \in [0, 1], \quad \mathbb{P}(\mathcal{Z} \leq z) = \int_0^z \frac{1}{3(1 - \frac{u}{2})^3} du, \tag{9}$$

*then we have $\mathbb{E}\left[ \nabla f \left( \frac{\mathcal{Z}}{2} * (\mathbf{x} - \underline{\mathbf{x}}) + \underline{\mathbf{x}} \right) \right] = \nabla F(\mathbf{x})$. Moreover, we have*

$$\frac{1 - \|\underline{\mathbf{x}}\|_\infty}{4} f(\mathbf{y}) - f \left( \frac{\mathbf{x} + \underline{\mathbf{x}}}{2} \right) \leq \frac{3}{8} \langle \nabla F(\mathbf{x}), \mathbf{y} - \mathbf{x} \rangle.$$

---

**Algorithm 7** Boosted Query oracle for non-monotone up-concave functions over general convex sets – $\texttt{BQN}$

1: **Input:** point $\mathbf{x}$
2: Sample $z \in [0, 1]$ according to Equation 9
3: **Return:** $\frac{z}{2} * (\mathbf{x} - \underline{\mathbf{x}}) + \underline{\mathbf{x}}$

---

In this case, $\mathfrak{g} = \nabla F(\mathbf{x})$, $h(\mathbf{x}) : \mathbf{x} \mapsto \frac{\mathbf{x}_t + \underline{\mathbf{x}}}{2}$. Further, if $f$ is $L$-smooth, $\mathfrak{g}$ is Lipschitz continuous, as given in the following Lemma.

**Lemma 6** (Theorem 18, Zhang et al. (2024)). *If $f$ is $L$-smooth, $L_1$-Lipschitz and $f$ satisfies assumptions in Lemma 5, $\nabla F(\mathbf{x})$ is $\frac{1}{8}L$-smooth and $\frac{3}{8}L_1$-Lipschitz continuous.*

## B   Proof of Theorem 1

Let $\mathbf{x}^* = \text{argmax}_{\mathbf{u} \in \mathcal{K}} \frac{1}{N} \sum_{t=1}^{T} \sum_{i=1}^{N} f_{t,i}(\mathbf{u})$, $\mathcal{T}_m = \{(m-1)K+1, \cdots, mK\}$. By the definition of $\alpha$-regret for agent $j$, we have

$$\mathbb{E}[\mathcal{R}_\alpha^j] = \frac{1}{N} \sum_{t=1}^{T} \sum_{i=1}^{N} \mathbb{E}\left[\alpha f_{t,i}(\mathbf{x}^*) - f_{t,i}(h(\mathbf{x}_t^j))\right]$$

$$= \frac{1}{N} \sum_{i=1}^{N} \sum_{m=1}^{T/K} \sum_{t \in \mathcal{T}_m} \mathbb{E}\left[\alpha f_{t,i}(\mathbf{x}^*) - f_{t,i}(h(\mathbf{x}_m^j))\right]$$

$$\stackrel{(a)}{\leq} \frac{\beta}{N} \sum_{i=1}^{N} \sum_{m=1}^{T/K} \sum_{t \in \mathcal{T}_m} \mathbb{E}\left[\langle \mathbf{x}^* - \mathbf{x}_m^j, \mathfrak{g}_{t,i}(\mathbf{x}_m^j) \rangle\right]$$

$$\stackrel{(b)}{=} \frac{\beta}{N} \sum_{i=1}^{N} \sum_{m=1}^{T/K} \sum_{t \in \mathcal{T}_m} \mathbb{E}\left[\langle \mathbf{x}^* - \mathbf{x}_m^j, \mathbb{E}\left[\tilde{\mathfrak{g}}_{t,i}(\mathbf{x}_m^j)|\mathbf{x}_m^j\right] \rangle\right]$$

$$\stackrel{(c)}{=} \frac{\beta}{N} \sum_{i=1}^{N} \sum_{m=1}^{T/K} \sum_{t \in \mathcal{T}_m} \mathbb{E}\left[\mathbb{E}\left[\langle \mathbf{x}^* - \mathbf{x}_m^j, \tilde{\mathfrak{g}}_{t,i}(\mathbf{x}_m^j) \rangle | \mathbf{x}_m^j\right]\right]$$

$$\stackrel{(d)}{=} \frac{\beta}{N} \sum_{i=1}^{N} \sum_{m=1}^{T/K} \sum_{t \in \mathcal{T}_m} \mathbb{E}\left[\langle \mathbf{x}^* - \mathbf{x}_m^j, \tilde{\mathfrak{g}}_{t,i}(\mathbf{x}_m^j) \rangle\right]$$

$$\stackrel{(e)}{=} \mathbb{E}\left[\underbrace{\frac{\beta}{N} \sum_{i=1}^{N} \sum_{m=1}^{T/K} \sum_{t \in \mathcal{T}_m} \langle \mathbf{x}^* - \mathbf{x}_m^i, \tilde{\mathfrak{g}}_{t,i}(\mathbf{x}_m^i) \rangle}_{:=P_1}\right] + \mathbb{E}\left[\underbrace{\frac{\beta}{N} \sum_{i=1}^{N} \sum_{m=1}^{T/K} \sum_{t \in \mathcal{T}_m} \langle \mathbf{x}_m^i - \mathbf{x}_m^j, \tilde{\mathfrak{g}}_{t,i}(\mathbf{x}_m^i) \rangle}_{:=P_2}\right]$$

$$+ \mathbb{E}\left[\underbrace{\frac{\beta}{N} \sum_{i=1}^{N} \sum_{m=1}^{T/K} \sum_{t \in \mathcal{T}_m} \langle \mathbf{x}^* - \mathbf{x}_m^j, \tilde{\mathfrak{g}}_{t,i}(\mathbf{x}_m^j) - \tilde{\mathfrak{g}}_{t,i}(\mathbf{x}_m^i) \rangle}_{:=P_3}\right] \qquad (10)$$

where step (a) is because $f_{t,i}$'s are upper linearizable, step (b) is because $\tilde{\mathfrak{g}}_{t,i}(\cdot)$ is unbiased, step (c) is due to the linearity of conditional expectation, step (d) is due to law of iterated expectation, and step (e) rewrites $(\mathbf{x}^* - \mathbf{x}_m^j)$ as $[(\mathbf{x}^* - \mathbf{x}_m^i) + (\mathbf{x}_m^i - \mathbf{x}_m^j)]$ and $\tilde{\mathfrak{g}}_{t,i}(\mathbf{x}_m^j)$ as $[(\tilde{\mathfrak{g}}_{t,i}(\mathbf{x}_m^j) - \tilde{\mathfrak{g}}_{t,i}(\mathbf{x}_m^i)) + \tilde{\mathfrak{g}}_{t,i}(\mathbf{x}_m^i)]$.

As illustrated in step (e) in Equation 10, total regret can be divided into three parts, $P_1, P_2, P_3$, and we provide upper bound for each of these three parts with proof in the following sections, B.2, B.3, B.4, respectively. In Section B.1, we introduce some auxiliary variables and lemmas that are useful to the following proof.

### B.1   Auxiliary variables and lemmas

In this section, we introduce some auxiliary variables and lemmas to better present the proof of bound for each of the three parts of the total regret presented in Equation 10.

Let $\mathbf{y}_1^i = \tilde{\mathbf{y}}_m^i = \mathbf{c}$, for any $i \in [N]$, since Algorithm 1 would only generate $\mathbf{y}_m^i$ for $m = 2, \cdots, T/K$. Let $\mathbf{r}_m^i = \tilde{\mathbf{y}}_m^i - \mathbf{y}_m^i$ for any $i \in [N]$ and $m \in [T/K]$. For any $m \in [T/K]$, we denote the averages by:

$$\bar{\mathbf{x}}_m = \frac{\sum_{i=1}^{N} \mathbf{x}_m^i}{N}, \ \bar{\mathbf{y}}_m = \frac{\sum_{i=1}^{N} \mathbf{y}_m^i}{N}, \ \hat{\mathbf{y}}_m = \frac{\sum_{i=1}^{N} \tilde{\mathbf{y}}_m^i}{N}, \ \text{and} \ \bar{\mathbf{r}}_m = \frac{\sum_{i=1}^{N} \mathbf{r}_m^i}{N}.$$

There are two lemmas that would be useful to the proofs, and we provide proof of each of these in the following appendices.

**Lemma 7.** *For any $i \in [N]$ and $m \in [T/K]$, Algorithm 1 ensures*

$$\|\mathbf{r}_m^i\| \leq 2\sqrt{3\epsilon} + 2\eta K G.$$

*Proof.* See Appendix B.7 for complete proof for Lemma 7. □

**Lemma 8.** *For any $i \in [N]$ and $m \in [T/K]$, Algorithm 1 ensures*

$$\sqrt{\sum_{i=1}^{N} \|\hat{\mathbf{y}}_m - \tilde{\mathbf{y}}_m^i\|^2} \leq \frac{\sqrt{N}(3\eta K G + 2\sqrt{3\epsilon})}{1 - \lambda_2}, \tag{11}$$

$$\sqrt{\sum_{i=1}^{N} \|\hat{\mathbf{y}}_m - \mathbf{y}_{m+1}^i\|^2} \leq \frac{\sqrt{N}(3\eta K G + 2\sqrt{3\epsilon})}{1 - \lambda_2}, \ and \tag{12}$$

$$\sum_{i=1}^{N} \|\mathbf{x}_m^i - \mathbf{x}_m^j\| \leq \left(3\sqrt{2\epsilon} + \frac{(3\eta K G + 2\sqrt{3\epsilon})}{1 - \lambda_2}\right)(N^{3/2} + N). \tag{13}$$

*Proof.* See Appendix B.8 for complete proof of Lemma 8. □

## B.2   Bound of $P_1$

By replacing $(\mathbf{x}^* - \mathbf{x}_m^i)$ with $[(\mathbf{x}^* - \hat{\mathbf{y}}_m) + (\hat{\mathbf{y}}_m - \tilde{\mathbf{y}}_m^i) + (\tilde{\mathbf{y}}_m^i - \mathbf{x}_m^i)]$, $P_1$ can be decomposed as:

$$\frac{1}{\beta}\P_1 = \frac{1}{N}\sum_{i=1}^{N}\sum_{m=1}^{T/K}\sum_{t \in \mathcal{T}_m} \langle \hat{\mathbf{y}}_m - \tilde{\mathbf{y}}_m^i, \tilde{\mathfrak{g}}_{t,i}(\mathbf{x}_m^i)) \rangle + \frac{1}{N}\sum_{i=1}^{N}\sum_{m=1}^{T/K}\sum_{t \in \mathcal{T}_m} \langle \tilde{\mathbf{y}}_m^i - \mathbf{x}_m^i, \tilde{\mathfrak{g}}_{t,i}(\mathbf{x}_m^i) \rangle$$

$$+ \underbrace{\frac{1}{N}\sum_{i=1}^{N}\sum_{m=1}^{T/K}\sum_{t \in \mathcal{T}_m} \langle \mathbf{x}^* - \hat{\mathbf{y}}_m, \tilde{\mathfrak{g}}_{t,i}(\mathbf{x}_m^i) \rangle}_{:=P_4}$$

$$\overset{(a)}{\leq} \frac{G}{N}\left[\sum_{i=1}^{N}\sum_{m=1}^{T/K}\sum_{t \in \mathcal{T}_m} \|\hat{\mathbf{y}}_m - \tilde{\mathbf{y}}_m^i\| + \sum_{i=1}^{N}\sum_{m=1}^{T/K}\sum_{t \in \mathcal{T}_m} \|\tilde{\mathbf{y}}_m^i - \mathbf{x}_m^i\|\right] + P_4$$

$$\overset{(b)}{\leq} G\left(\sum_{m=1}^{T/K}\sum_{t \in \mathcal{T}_m} \sqrt{\frac{1}{N}\sum_{i=1}^{N} \|\hat{\mathbf{y}}_m - \tilde{\mathbf{y}}_m^i\|^2} + T\sqrt{3\epsilon}\right) + \mathcal{R}_4$$

$$\overset{(c)}{\leq} TG\left(\frac{3\eta K G + 2\sqrt{3\epsilon}}{1 - \lambda_2} + \sqrt{3\epsilon}\right) + \mathcal{R}_4 \tag{14}$$

where step (a) follows from Cauchy-Schwartz inequality and the bound of $\tilde{\mathfrak{g}}_{t,i}(\cdot)$ function, step (b) follows from Arithmetic Mean-Quadratic Mean inequality and Lemma 1, and step (c) follows from Equation (11) in Lemma 8.

To attain upper bound on $P_4$, we notice that,

$$\bar{\mathbf{y}}_{m+1} = \frac{1}{N}\sum_{i=1}^{N}\mathbf{y}_{m+1}^i = \frac{1}{N}\sum_{i=1}^{N}\left(\sum_{j \in \mathcal{N}_i} a_{ij}\tilde{\mathbf{y}}_m^j + \eta\sum_{t \in \mathcal{T}_m}\tilde{\mathfrak{g}}_{t,i}(\mathbf{x}_m^i)\right)$$

$$\overset{(a)}{=} \frac{1}{N}\sum_{i=1}^{N}\sum_{j=1}^{N} a_{ij}\tilde{\mathbf{y}}_m^j + \frac{\eta}{N}\sum_{i=1}^{N}\sum_{t \in \mathcal{T}_m}\tilde{\mathfrak{g}}_{t,i}(\mathbf{x}_m^i)$$

$$\overset{(b)}{=} \hat{\mathbf{y}}_m + \frac{\eta}{N}\sum_{i=1}^{N}\sum_{t \in \mathcal{T}_m}\tilde{\mathfrak{g}}_{t,i}(\mathbf{x}_m^i) \tag{15}$$

where step (a) is because $a_{ij} = 0$ for any agent $j \notin \mathcal{N}_i$, and step (b) is because $\mathbf{A1} = \mathbf{1}$.

Substituting $\bar{\mathbf{y}}_{m+1}$ using Equation (14), we have for any $m \in [T/K]$,

$$\hat{\mathbf{y}}_{m+1} = \hat{\mathbf{y}}_{m+1} - \bar{\mathbf{y}}_{m+1} + \bar{\mathbf{y}}_{m+1}$$

$$\overset{(15)}{=} \bar{\mathbf{r}}_{m+1} + \hat{\mathbf{y}}_m + \frac{\eta}{N} \sum_{i=1}^{N} \sum_{t \in \mathcal{T}_m} \tilde{\mathfrak{g}}_{t,i}(\mathbf{x}_m^i) \tag{16}$$

because $\bar{\mathbf{r}}_{m+1} = \hat{\mathbf{y}}_{m+1} - \bar{\mathbf{y}}_{m+1}$.

By replacing $\hat{\mathbf{y}}_{m+1}$ with Equation (16) and then expand the equation, we have

$$\|\hat{\mathbf{y}}_{m+1} - \mathbf{x}^*\|^2 \overset{(16)}{=} \left\| \bar{\mathbf{r}}_{m+1} + \hat{\mathbf{y}}_m + \frac{\eta}{N} \sum_{i=1}^{N} \sum_{t \in \mathcal{T}_m} \tilde{\mathfrak{g}}_{t,i}(\mathbf{x}_m^i) - \mathbf{x}^* \right\|^2$$

$$= \|\hat{\mathbf{y}}_m - \mathbf{x}^*\|^2 + 2 \left\langle \hat{\mathbf{y}}_m - \mathbf{x}^*, \frac{\eta}{N} \sum_{i=1}^{N} \sum_{t \in \mathcal{T}_m} \tilde{\mathfrak{g}}_{t,i}(\mathbf{x}_m^i) \right\rangle$$

$$+ 2\langle \hat{\mathbf{y}}_m - \mathbf{x}^*, \bar{\mathbf{r}}_{m+1} \rangle + \left\| \bar{\mathbf{r}}_{m+1} + \frac{\eta}{N} \sum_{i=1}^{N} \sum_{t \in \mathcal{T}_m} \tilde{\mathfrak{g}}_{t,i}(\mathbf{x}_m^i) \right\|^2. \tag{17}$$

Following Lemma 1, we deduce that for any $m \in [T/K]$,

$$\|\tilde{\mathbf{y}}_{m+1}^i - \mathbf{x}^*\|^2 \leq \|\mathbf{y}_{m+1}^i - \mathbf{x}^*\|^2 = \|\mathbf{y}_{m+1}^i - \tilde{\mathbf{y}}_{m+1}^i + \tilde{\mathbf{y}}_{m+1}^i - \mathbf{x}^*\|^2$$

$$= \|\mathbf{y}_{m+1}^i - \tilde{\mathbf{y}}_{m+1}^i\|^2 + 2\langle \mathbf{y}_{m+1}^i - \tilde{\mathbf{y}}_{m+1}^i, \tilde{\mathbf{y}}_{m+1}^i - \mathbf{x}^* \rangle + \|\tilde{\mathbf{y}}_{m+1}^i - \mathbf{x}^*\|^2$$

$$= \|\mathbf{r}_{m+1}^i\|^2 - 2\langle \mathbf{r}_{m+1}^i, \tilde{\mathbf{y}}_{m+1}^i - \mathbf{x}^* \rangle + \|\tilde{\mathbf{y}}_{m+1}^i - \mathbf{x}^*\|^2$$

where the last equality is due to the definition of $\mathbf{r}_{m+1}^i$.

Omitting $\|\tilde{\mathbf{y}}_{m+1}^i - \mathbf{x}^*\|^2$ on both sides and moving $\langle \tilde{\mathbf{y}}_{m+1}^i - \mathbf{x}^*, \mathbf{r}_{m+1}^i \rangle$ to the left, we have

$$\langle \tilde{\mathbf{y}}_{m+1}^i - \mathbf{x}^*, \mathbf{r}_{m+1}^i \rangle \leq \frac{1}{2} \|\mathbf{r}_{m+1}^i\|^2. \tag{18}$$

Thus, we can bound the term $\langle \hat{\mathbf{y}}_m - \mathbf{x}^*, \bar{\mathbf{r}}_{m+1} \rangle$ in Equation (17) as follows

$$\langle \hat{\mathbf{y}}_m - \mathbf{x}^*, \bar{\mathbf{r}}_{m+1} \rangle = \frac{1}{N} \sum_{i=1}^{N} \langle \hat{\mathbf{y}}_m - \mathbf{x}^*, \mathbf{r}_{m+1}^i \rangle$$

$$\overset{(a)}{\leq} \frac{1}{N} \sum_{i=1}^{N} \langle \hat{\mathbf{y}}_m - \mathbf{y}_{m+1}^i, \mathbf{r}_{m+1}^i \rangle + \frac{1}{N} \sum_{i=1}^{N} \langle \tilde{\mathbf{y}}_{m+1}^i - \mathbf{x}^*, \mathbf{r}_{m+1}^i \rangle$$

$$\overset{(b)}{\leq} \frac{1}{N} \sum_{i=1}^{N} \|\hat{\mathbf{y}}_m - \mathbf{y}_{m+1}^i\| \|\mathbf{r}_{m+1}^i\| + \frac{1}{2N} \sum_{i=1}^{N} \|\mathbf{r}_{m+1}^i\|^2$$

$$\overset{(c)}{\leq} \frac{2\eta K G + 2\sqrt{3\epsilon}}{\sqrt{N}} \sqrt{\sum_{i=1}^{N} \|\hat{\mathbf{y}}_m - \mathbf{y}_{m+1}^i\|^2} + \frac{1}{2N} \sum_{i=1}^{N} \|\mathbf{r}_{m+1}^i\|^2$$

$$\overset{(d)}{\leq} \frac{1}{1 - \lambda_2} \left( 6\eta^2 K^2 G^2 + 10\eta K G \sqrt{3\epsilon} + 12\epsilon \right)$$

$$+ 2 \left( \eta^2 K^2 G^2 + 2\eta K G \sqrt{3\epsilon} + 3\epsilon \right) \tag{19}$$

where step (a) replaces $\hat{\mathbf{y}}_m - \mathbf{x}^*$ as $(\hat{\mathbf{y}}_m - \mathbf{y}_{m+1}^i) + (\mathbf{y}_{m+1}^i - \tilde{\mathbf{y}}_{m+1}^i) + (\tilde{\mathbf{y}}_{m+1}^i - \mathbf{x}^*)$ and omits $\langle \mathbf{y}_{m+1}^i - \tilde{\mathbf{y}}_{m+1}^i, \mathbf{r}_{m+1}^i \rangle \leq 0$, step (b) is due to Cauchy-Schwartz inequality and Equation (18), step (c) comes from Lemma 7 and Arithmetic Mean-Quadratic Mean inequality, and step (d) follows from Equation 11 in Lemma 8.

Also, we can bound the last term in Equation (17) as follows

$$
\begin{aligned}
\left\| \bar{\mathbf{r}}_{m+1} + \frac{\eta}{N} \sum_{i=1}^{N} \sum_{t \in \mathcal{T}_m} \tilde{\mathfrak{g}}_{t,i}(\mathbf{x}_m^i) \right\|^2 &\overset{(a)}{\leq} 2\|\bar{\mathbf{r}}_{m+1}\|^2 + 2 \left\| \frac{\eta}{N} \sum_{i=1}^{N} \sum_{t \in \mathcal{T}_m} \tilde{\mathfrak{g}}_{t,i}(\mathbf{x}_m^i) \right\|^2 \\
&\overset{(b)}{\leq} \frac{2}{N} \sum_{i=1}^{N} \|\mathbf{r}_{m+1}^i\|^2 + \frac{2K\eta^2}{N} \sum_{i=1}^{N} \sum_{t \in \mathcal{T}_m} \|\tilde{\mathfrak{g}}_{t,i}(\mathbf{x}_m^i)\|^2 \\
&\overset{(c)}{\leq} 8\left(\sqrt{3\epsilon} + \eta KG\right)^2 + 2\eta^2 K^2 G^2 \\
&= 24\epsilon + 10\eta^2 K^2 G^2 + 16\eta KG\sqrt{3\epsilon}
\end{aligned}
\tag{20}
$$

where both step (a) and step (b) utilize Cauchy-Schwartz inequality and step (c) is due to Lemma 7 and the bound of $\tilde{\mathfrak{g}}_{t,i}(\cdot)$ functions.

Substitute Equation (19) and Equation (20) into Equation (17) and taking expectation on both sides, we have

$$
\begin{aligned}
\mathbb{E}\left[\|\mathbf{x}^* - \hat{\mathbf{y}}_{m+1}\|^2\right] \leq {} & \mathbb{E}\left[\|\mathbf{x}^* - \hat{\mathbf{y}}_m\|^2\right] - \frac{2\eta}{N} \sum_{i=1}^{N} \sum_{t \in \mathcal{T}_m} \mathbb{E}\left[\langle \mathbf{x}^* - \hat{\mathbf{y}}_m, \tilde{\mathfrak{g}}_{t,i}(\mathbf{x}_m^i)\rangle\right] \\
& + \frac{1}{1-\lambda_2}\left(12\eta^2 K^2 G^2 + 20\eta KG\sqrt{3\epsilon} + 24\epsilon\right) \\
& + \left(14\eta^2 K^2 G^2 + 24\eta KG\sqrt{3\epsilon} + 36\epsilon\right).
\end{aligned}
\tag{21}
$$

Moving terms to different sides in Equation (21), we have

$$
\begin{aligned}
\mathbb{E}\left[P_4\right] = {} & \frac{1}{N} \sum_{i=1}^{N} \sum_{m=1}^{T/K} \sum_{t \in \mathcal{T}_m} \mathbb{E}\left[\langle \mathbf{x}^* - \hat{\mathbf{y}}_m, \tilde{\mathfrak{g}}_{t,i}(\mathbf{x}_m^i)\rangle\right] \\
& \overset{(21)}{\leq} \sum_{m=1}^{T/K} \left[\frac{\mathbb{E}\left[\|\hat{\mathbf{y}}_m - \mathbf{x}^*\|^2\right] - \mathbb{E}\left[\|\hat{\mathbf{y}}_{m+1} - \mathbf{x}^*\|^2\right]}{2\eta}\right] \\
& + \frac{T}{K}\left[\frac{18\epsilon}{\eta} + 7\eta K^2 G^2 + 12KG\sqrt{3\epsilon}\right] \\
& + \frac{T}{K}\left[\frac{1}{1-\lambda_2}\left(6\eta K^2 G^2 + 10KG\sqrt{3\epsilon} + \frac{12\epsilon}{\eta}\right)\right] \\
& \overset{(a)}{\leq} \frac{2R^2}{\eta} + \frac{18\epsilon T}{\eta K} + 7\eta TKG^2 + 12TG\sqrt{3\epsilon} \\
& + \frac{1}{1-\lambda_2}\left(\frac{12\epsilon T}{\eta K} + 6\eta TKG^2 + 10TG\sqrt{3\epsilon}\right)
\end{aligned}
\tag{22}
$$

where the last inequality is due to $\mathbb{E}\left[\|\hat{\mathbf{y}}_{T/K+1} - \mathbf{x}^*\|^2\right] \geq 0$ and $\|\hat{\mathbf{y}}_1 - \mathbf{x}^*\|^2 \leq 4R^2$, which is derived by combining $\hat{\mathbf{y}}_1 = \mathbf{c} \in \mathcal{K}$, $\mathbf{x}^* \in \mathcal{K}$, and $R = \max_{\mathbf{x} \in \mathcal{K}} \|\mathbf{x}\|$, and Cauchy-Schwarz Inequality.

Therefore, plugging the result for term $\mathbb{E}\left[P_4\right]$ from Equation (22) into Equation (14), we have

$$
\mathbb{E}\left[P_1\right] \leq TG\beta \left(\frac{3\eta KG + 2\sqrt{3\epsilon}}{1-\lambda_2} + \sqrt{3\epsilon}\right)
\tag{23}
$$

$$
\begin{aligned}
& + \frac{2\beta R^2}{\eta} + \frac{18\beta\epsilon T}{\eta K} + 7\beta\eta TKG^2 + 12TG\beta\sqrt{3\epsilon} \\
& + \frac{\beta}{1-\lambda_2}\left(\frac{12\epsilon T}{\eta K} + 6\eta TKG^2 + 10TG\sqrt{3\epsilon}\right)
\end{aligned}
\tag{24}
$$

### B.3 Bound of $P_2$

Next, we bound $P_2$ in (10)

$$
\begin{aligned}
\P_2 &\overset{(10)}{=} \frac{\beta}{N} \sum_{i=1}^{N} \sum_{m=1}^{T/K} \sum_{t \in \mathcal{T}_m} \langle \mathbf{x}_m^i - \mathbf{x}_m^j, \tilde{\mathfrak{g}}_{t,i}(\mathbf{x}_m^i) \rangle \\
&\overset{(a)}{\leq} \frac{\beta}{N} \sum_{i=1}^{N} \sum_{m=1}^{T/K} \sum_{t \in \mathcal{T}_m} \|\mathbf{x}_m^i - \mathbf{x}_m^j\| \|\tilde{\mathfrak{g}}_{t,i}(\mathbf{x}_m^i)\| \\
&\overset{(b)}{\leq} \frac{G\beta}{N} \sum_{m=1}^{T/K} \sum_{t \in \mathcal{T}_m} \sum_{i=1}^{N} \|\mathbf{x}_m^i - \mathbf{x}_m^j\| \\
&\overset{(c)}{\leq} G\beta(N^{1/2} - +1) \left( 3\sqrt{2}\epsilon + \frac{3\eta K G + 2\sqrt{3}\epsilon}{1 - \lambda_2} \right).
\end{aligned}
\tag{25}
$$

where step (a) is due to Cauchy-Schwartz inequality, step (b) follows from the bound of $\tilde{\mathfrak{g}}_{t,i}(\cdot)$ functions, and step (c) uses the inequality (13) in Lemma 8.

### B.4 Bound of $P_3$

Recall that $\mathfrak{g}_{t,i}(\cdot)$ are $L_1$-Lipschitz continuous, i.e.,

$$
\|\mathfrak{g}_{t,i}(\mathbf{x}_m^j) - \mathfrak{g}_{t,i}(\mathbf{x}_m^i)\| \leq L_1 \|\mathbf{x}_m^i - \mathbf{x}_m^j\|.
$$

Thus, we have

$$
\begin{aligned}
\mathbb{E}[P_3] &\overset{(10)}{=} \frac{\beta}{N} \sum_{i=1}^{N} \sum_{m=1}^{T/K} \sum_{t \in \mathcal{T}_m} \mathbb{E} \langle \mathbf{x}^* - \mathbf{x}_m^j, \tilde{\mathfrak{g}}_{t,i}(\mathbf{x}_m^j) - \tilde{\mathfrak{g}}_{t,i}(\mathbf{x}_m^i) \rangle \\
&\overset{(a)}{=} \frac{\beta}{N} \sum_{i=1}^{N} \sum_{m=1}^{T/K} \sum_{t \in \mathcal{T}_m} \mathbb{E} \left[ \langle \mathbf{x}^* - \mathbf{x}_m^j, \mathbb{E} \left[ \tilde{\mathfrak{g}}_{t,i}(\mathbf{x}_m^j) | \mathbf{x}_m^j \right] - \mathbb{E} \left[ \tilde{\mathfrak{g}}_{t,i}(\mathbf{x}_m^i) | \mathbf{x}_m^i \right] \rangle \right] \\
&\overset{(b)}{=} \frac{\beta}{N} \sum_{i=1}^{N} \sum_{m=1}^{T/K} \sum_{t \in \mathcal{T}_m} \mathbb{E} \left[ \langle \mathbf{x}^* - \mathbf{x}_m^j, \mathfrak{g}_{t,i}(\mathbf{x}_m^j) - \mathfrak{g}_{t,i}(\mathbf{x}_m^i) \rangle \right] \\
&\overset{(c)}{\leq} \frac{\beta}{N} \sum_{i=1}^{N} \sum_{m=1}^{T/K} \sum_{t \in \mathcal{T}_m} \mathbb{E} \left[ \|\mathbf{x}^* - \mathbf{x}_m^j\| \|\mathfrak{g}_{t,i}(\mathbf{x}_m^j) - \mathfrak{g}_{t,i}(\mathbf{x}_m^i)\| \right] \\
&\overset{(d)}{\leq} \frac{L_1 \beta}{N} \sum_{i=1}^{N} \sum_{m=1}^{T/K} \sum_{t \in \mathcal{T}_m} \mathbb{E} \left[ \|\mathbf{x}^* - \mathbf{x}_m^j\| \|\mathbf{x}_m^i - \mathbf{x}_m^j\| \right] \\
&\overset{(e)}{\leq} \frac{2 L_1 R \beta}{N} \mathbb{E} \left[ \sum_{m=1}^{T/K} \sum_{t \in \mathcal{T}_m} \sum_{i=1}^{N} \|\mathbf{x}_m^i - \mathbf{x}_m^j\| \right] \\
&\overset{(f)}{\leq} 2 L_1 R \beta(N^{1/2} + 1) \left( 3\sqrt{2}\epsilon + \frac{3\eta K G + 2\sqrt{3}\epsilon}{1 - \lambda_2} \right)
\end{aligned}
\tag{26}
$$

where step (a) and (b) is due to the law of iterated expectations, step (c) comes from Cauchy-Schwartz inequality, step (d) follows from continuity of $\mathfrak{g}_{t,i}(\cdot)$ functions, step (e) follows from the bound on $\mathcal{K}$ and Cauchy-Schwartz inequality, and step (f) is due to Equation (13) in Lemma 8.

### B.5 Final Regret Bound

Plugging Equation (23), Equation (25) and Equation (26) into Equation (10), for any $j \in [N]$, we have

$$
\begin{aligned}
\mathbb{E}\left[\mathcal{R}_{T,\alpha}^j\right] \leq\ & \frac{2\beta R^2}{\eta} + \frac{18\epsilon\beta T}{\eta K} + 7\eta\beta TKG^2 + 13TG\beta\sqrt{3\epsilon} \\
& + \frac{\beta}{1-\lambda_2}\left(\frac{12\epsilon T}{\eta K} + 9\eta TKG^2 + 12TG\sqrt{3\epsilon}\right) \\
& + G\beta(N^{1/2}+1)\left(3\sqrt{2\epsilon} + \frac{(3\eta KG + 2\sqrt{3\epsilon})}{1-\lambda_2}\right) \\
& + 2L_1 RT\beta(N^{1/2}+1)\left(3\sqrt{2\epsilon} + \frac{(3\eta KG + 2\sqrt{3\epsilon})}{1-\lambda_2}\right)
\end{aligned}
$$

### B.6 Number of Linear Optimization Oracle Calls

Finally, we analyze the total number of linear optimization oracle calls for each agent $i$. In Lemma 1, the term $\mathcal{R}_5 = \left\|\mathbf{y}_{m+1}^i - \sum_{j\in\mathcal{N}_i} a_{ij}\mathbf{x}_m^j\right\|^2$ can be bounded as follows

$$
\begin{aligned}
\mathcal{R}_5 = \left\|\mathbf{y}_{m+1}^i - \sum_{j\in\mathcal{N}_i} a_{ij}\mathbf{x}_m^j\right\|^2 
&\overset{(a)}{\leq} 2\left\|\mathbf{y}_{m+1}^i - \sum_{j\in\mathcal{N}_i} a_{ij}\tilde{\mathbf{y}}_m^j\right\|^2 + 2\left\|\sum_{j\in\mathcal{N}_i} a_{ij}\tilde{\mathbf{y}}_m^j - \sum_{j\in\mathcal{N}_i} a_{ij}\mathbf{x}_m^j\right\|^2 \\
&\overset{(b)}{\leq} 2\left\|\eta\sum_{t\in\mathcal{T}_m} \tilde{\mathfrak{g}}_{t,i}(\mathbf{x}_m^i)\right\|^2 + 2\sum_{j\in\mathcal{N}_i} a_{ij}\left\|\tilde{\mathbf{y}}_m^j - \mathbf{x}_m^j\right\|^2 \\
&\overset{(c)}{\leq} 2\eta^2 K^2 G^2 + 6\epsilon
\end{aligned}
\tag{27}
$$

where both step (a) follows from Cauchy-Schwartz inequality, step (b) follows from Cauchy-Schwartz inequality and Line (10) in Algorithm 1

From Equation (3) in Lemma 1, in each block $m$, each agent $i$ in Algorithm 1 at most utilizes

$$
\begin{aligned}
l_m^i &= \frac{27R^2}{\epsilon}\max\left(\frac{\|\mathbf{y}_{m+1}^i - \sum_{j\in\mathcal{N}_i} a_{ij}\mathbf{x}_m^j\|^2(\|\mathbf{y}_{m+1}^i - \sum_{j\in\mathcal{N}_i} a_{ij}\mathbf{x}_m^j\|^2 - \epsilon)}{4\epsilon^2} + 1, 1\right) \\
&= \frac{27R^2}{\epsilon}\max\left(\frac{\mathcal{R}_5(\mathcal{R}_5 - \epsilon)}{4\epsilon^2} + 1, 1\right) \\
&\overset{(27)}{\leq} \frac{27R^2}{\epsilon}\max\left(\frac{(2\eta^2 K^2 G^2 + 6\epsilon)(2\eta^2 K^2 G^2 + 6\epsilon - \epsilon)}{4\epsilon^2} + 1, 1\right) \\
&= \frac{27R^2}{\epsilon}\frac{(2\eta^2 K^2 G^2 + 6\epsilon)(2\eta^2 K^2 G^2 + 5\epsilon) + 4\epsilon^2}{4\epsilon^2}
\end{aligned}
\tag{28}
$$

linear optimization oracle calls, where the last equality is due to the fact that $(2\eta^2 K^2 G^2 + 6\epsilon)(2\eta^2 K^2 G^2 + 5\epsilon) \geq 0$.

Thus, by summing Equation (28) over $\frac{T}{K}$ blocks, we have that the total number of linear optimization steps required by each agent $i$ of Algorithm 1 is at most

$$
\sum_{m=1}^{T/K} l_m^i \leq \frac{27TR^2}{\epsilon K}\left(8.5 + 5.5\frac{K^2\eta^2 G^2}{\epsilon} + \frac{K^4\eta^4 G^4}{\epsilon^2}\right)
$$

## B.7 Proof of Lemma 7

$$
\begin{aligned}
\|\mathbf{r}_{m+1}^i\| = \|\tilde{\mathbf{y}}_{m+1}^i - \mathbf{y}_{m+1}^i\| &\overset{(a)}{\leq} \left\| \tilde{\mathbf{y}}_{m+1}^i - \sum_{j \in \mathcal{N}_i} a_{ij} \mathbf{x}_m^j \right\| + \left\| \sum_{j \in \mathcal{N}_i} a_{ij} \mathbf{x}_m^j - \mathbf{y}_{m+1}^i \right\| \\
&\overset{(b)}{\leq} 2 \left\| \sum_{j \in \mathcal{N}_i} a_{ij} \mathbf{x}_m^j - \mathbf{y}_{m+1}^i \right\| \overset{(c)}{=} 2 \left\| \sum_{j \in \mathcal{N}_i} a_{ij} \mathbf{x}_m^j - \sum_{j \in \mathcal{N}_i} a_{ij} \tilde{\mathbf{y}}_m^j - \eta \sum_{t=(m-1)K+1}^{mK} \tilde{\mathbf{g}}_{t,i}(\mathbf{x}_m^i) \right\| \\
&\overset{(d)}{\leq} 2 \sum_{j \in \mathcal{N}_i} a_{ij} \|\mathbf{x}_m^j - \tilde{\mathbf{y}}_m^j\| + 2\eta K G \\
&\overset{(e)}{\leq} 2\sqrt{3\epsilon} + 2\eta K G
\end{aligned}
$$

where step (a) comes from triangle inequality, step (b) follows by Lemma 1, step (c) replaces $\mathbf{y}_{m+1}^i$ with update rule described in Algorithm 1, step (d) comes from triangle inequality, and step (e) follows by Lemma 1.

Moreover, if $m = 1$, we can verify that

$$
\|\mathbf{r}_1^i\| = \|\mathbf{0}\| \leq 2\sqrt{3\epsilon} + 2\eta K G.
$$

## B.8 Proof of Lemma 8

To prove Lemma 8, we introduce additional auxiliary variables as follows:

$$
\mathbf{x}_m' = [\mathbf{x}_m^1; \cdots; \mathbf{x}_m^N] \in \mathbb{R}^{Nd}, \ \mathbf{y}_m' = [\mathbf{y}_m^1; \cdots; \mathbf{y}_m^N] \in \mathbb{R}^{Nd}, \ \tilde{\mathbf{y}}_m' = [\tilde{\mathbf{y}}_m^1; \cdots; \tilde{\mathbf{y}}_m^N] \in \mathbb{R}^{Nd}
$$

and

$$
\mathbf{r}_m' = [\mathbf{r}_m^1; \cdots; \mathbf{r}_m^N] \in \mathbb{R}^{Nd}, \ \mathbf{g}_m' = \sum_{t=(m-1)K+1}^{mK} [\tilde{\mathbf{g}}_{t,1}(\mathbf{x}_m^1); \cdots; \tilde{\mathbf{g}}_{t,N}(\mathbf{x}_m^N)] \in \mathbb{R}^{Nd}.
$$

According to step 10 in Algorithm 1, for any $m \in \{2, \ldots, T/K\}$, we have

$$
\mathbf{y}_{m+1}' = (\mathbf{A} \otimes \mathbf{I}) \tilde{\mathbf{y}}_m' + \eta \mathbf{g}_m' = \sum_{k=1}^{m-1} (\mathbf{A} \otimes \mathbf{I})^{m-k} \mathbf{r}_{k+1}' + \sum_{k=1}^{m} (\mathbf{A} \otimes \mathbf{I})^{m-k} \eta \mathbf{g}_k' \tag{29}
$$

where the notation $\otimes$ indicates the Kronecker product and $\mathbf{I}$ denotes the identity matrix of size $n \times n$.

In the same manner, for any $m \in [T/K]$, we have

$$
\begin{aligned}
\tilde{\mathbf{y}}_{m+1}' = \mathbf{r}_{m+1}' + \mathbf{y}_{m+1}' &= \mathbf{r}_{m+1}' + (\mathbf{A} \otimes \mathbf{I}) \tilde{\mathbf{y}}_m' + \eta \mathbf{g}_m' \\
&= \sum_{k=1}^{m} (\mathbf{A} \otimes \mathbf{I})^{m-k} \mathbf{r}_{k+1}' + \sum_{k=1}^{m} (\mathbf{A} \otimes \mathbf{I})^{m-k} \eta \mathbf{g}_k'
\end{aligned} \tag{30}
$$

where the second equality follows the fact that $\mathbf{r}_1' = \tilde{\mathbf{y}}_1' - \mathbf{y}_1' = \mathbf{0}$.

By the definition of $\hat{\mathbf{y}}_{m+1}$, for any $m \in [T/K]$, we have

$$
[\hat{\mathbf{y}}_{m+1}; \cdots; \hat{\mathbf{y}}_{m+1}] = \left( \frac{\mathbf{1}\mathbf{1}^T}{N} \otimes \mathbf{I} \right) \tilde{\mathbf{y}}_{m+1}' \tag{31}
$$

where the second equality comes from $\mathbf{1}^\top \mathbf{A} = \mathbf{1}^\top$.

### B.8.1 Proof of Equation (11)

For any $m \in [T/K]$, we have

$$
\sqrt{\sum_{i=1}^{N} \|\hat{\mathbf{y}}_{m+1} - \tilde{\mathbf{y}}_{m+1}^i\|^2} \overset{(31)}{=} \left\| \left( \frac{\mathbf{1}\mathbf{1}^T}{N} \otimes \mathbf{I} \right) \tilde{\mathbf{y}}'_{m+1} - \tilde{\mathbf{y}}'_{m+1} \right\|
$$

$$
\overset{(30)}{=} \left\| \sum_{k=1}^{m} \left( \left( \frac{\mathbf{1}\mathbf{1}^T}{N} - \mathbf{A}^{m-k} \right) \otimes \mathbf{I} \right) \mathbf{r}'_{k+1} + \sum_{k=1}^{m} \left( \left( \frac{\mathbf{1}\mathbf{1}^T}{N} - \mathbf{A}^{m-k} \right) \otimes \mathbf{I} \right) \eta \mathbf{g}'_k \right\|
$$

$$
\overset{(a)}{\leq} \left\| \sum_{k=1}^{m} \left( \left( \frac{\mathbf{1}\mathbf{1}^T}{N} - \mathbf{A}^{m-k} \right) \otimes \mathbf{I} \right) \mathbf{r}'_{k+1} \right\| + \left\| \sum_{k=1}^{m} \left( \left( \frac{\mathbf{1}\mathbf{1}^T}{N} - \mathbf{A}^{m-k} \right) \otimes \mathbf{I} \right) \eta \mathbf{g}'_k \right\|
$$

$$
\overset{(b)}{\leq} \sum_{k=1}^{m} \left\| \frac{\mathbf{1}\mathbf{1}^T}{N} - \mathbf{A}^{m-k} \right\| \|\mathbf{r}'_{k+1}\| + \sum_{k=1}^{m} \left\| \frac{\mathbf{1}\mathbf{1}^T}{N} - \mathbf{A}^{m-k} \right\| \|\eta \mathbf{g}'_k\|
$$

$$
\overset{(c)}{\leq} \sqrt{N} \sum_{k=1}^{m} \lambda_2^{m-k} (3\eta K G + 2\sqrt{3}\epsilon) \overset{(d)}{\leq} \frac{\sqrt{N}(3\eta K G + 2\sqrt{3}\epsilon)}{1 - \lambda_2},
$$

where step (a) is due to triangle inequality, and step (b) is due to Cauchy-Schwartz inequality and triangle inequality, step (c) comes from Lemma 7 and the fact that $\forall k \in [m], \|\frac{\mathbf{1}\mathbf{1}^T}{N} - \mathbf{A}^{m-k}\| \leq \lambda_2^{m-k}$ (see Mokhtari et al. (2018) for details), and step (d) is due to the property of geometric series.

By noticing $\hat{\mathbf{y}}_1 = \tilde{\mathbf{y}}_1 = \mathbf{c}$, we complete the proof of Equation (11) in Lemma 8.

### B.8.2 Proof of Equation (12)

Similarly, for any $m \in \{2, \cdots, T/K\}$, we have

$$
\sqrt{\sum_{i=1}^{N} \|\hat{\mathbf{y}}_m - \mathbf{y}_{m+1}^i\|^2} \overset{(31)}{=} \left\| \left( \frac{\mathbf{1}\mathbf{1}^T}{N} \otimes \mathbf{I} \right) \tilde{\mathbf{y}}'_m - \mathbf{y}'_{m+1} \right\|
$$

$$
\overset{(29)}{=} \left\| \sum_{k=1}^{m-1} \left( \left( \frac{\mathbf{1}\mathbf{1}^T}{N} - \mathbf{A}^{m-k} \right) \otimes \mathbf{I} \right) \mathbf{r}'_{k+1} + \sum_{k=1}^{m-1} \left( \left( \frac{\mathbf{1}\mathbf{1}^T}{N} - \mathbf{A}^{m-k} \right) \otimes \mathbf{I} \right) \eta \mathbf{g}'_k - \eta \mathbf{g}'_m \right\|
$$

$$
\overset{(a)}{\leq} \left\| \sum_{k=1}^{m-1} \left( \left( \frac{\mathbf{1}\mathbf{1}^T}{N} - \mathbf{A}^{m-k} \right) \otimes \mathbf{I} \right) \mathbf{r}'_{k+1} \right\| + \left\| \sum_{k=1}^{m-1} \left( \left( \frac{\mathbf{1}\mathbf{1}^T}{N} - \mathbf{A}^{m-k} \right) \otimes \mathbf{I} \right) \eta \mathbf{g}'_k \right\| + \|\eta \mathbf{g}'_m\|
$$

$$
\overset{(b)}{\leq} \sum_{k=1}^{m-1} \left\| \frac{\mathbf{1}\mathbf{1}^T}{N} - \mathbf{A}^{m-k} \right\| \|\mathbf{r}'_{k+1}\| + \sum_{k=1}^{m-1} \left\| \frac{\mathbf{1}\mathbf{1}^T}{N} - \mathbf{A}^{m-k} \right\| \|\eta \mathbf{g}'_k\| + \|\eta \mathbf{g}'_m\|
$$

$$
\overset{(c)}{\leq} \sqrt{N} \sum_{k=1}^{m} \lambda_2^{m-k} (3\eta K G + 2\sqrt{3}\epsilon) \overset{(d)}{\leq} \frac{\sqrt{N}(3\eta K G + 2\sqrt{3}\epsilon)}{1 - \lambda_2},
$$

where step (a) is due to triangle inequality, and step (b) is due to Cauchy-Schwartz inequality and triangle inequality, step (c) comes from Lemma 7 and the fact that $\forall k \in [m], \|\frac{\mathbf{1}\mathbf{1}^T}{N} - \mathbf{A}^{m-k}\| \leq \lambda_2^{m-k}$ (see Mokhtari et al. (2018) for details), and step (d) is due to the property of geometric series.

When $m = 1$, $\hat{\mathbf{y}}_1 = \tilde{\mathbf{y}}_1^i = \sum_{j \in \mathcal{N}_i} a_{ij} \tilde{\mathbf{y}}_1^j = \mathbf{c}$. Due to Line (10) in Algorithm 1, we have

$$
\sqrt{\sum_{i=1}^{N} \|\hat{\mathbf{y}}_1 - \mathbf{y}_2^i\|^2} = \sqrt{\sum_{i=1}^{N} \left\| \sum_{j \in \mathcal{N}_i} a_{ij} \tilde{\mathbf{y}}_1^j - \mathbf{y}_2^i \right\|^2} = \sqrt{\sum_{i=1}^{N} \left\| \sum_{t \in \mathcal{T}_m} \tilde{\mathbf{g}}_{t,i}(\mathbf{x}_m^i) \right\|^2} \leq \sqrt{N} \eta K G.
$$

By noticing that $\sqrt{N}\eta K G < \frac{\sqrt{N}(3\eta K G + 2\sqrt{3}\epsilon)}{1 - \lambda_2}$, we complete the proof of Equation (12).

### B.8.3 Proof of Equation (13)

For any $m \in [T/K]$, we notice that

$$
\sum_{i=1}^{N} \|\mathbf{x}_m^i - \mathbf{x}_m^j\| \leq \sum_{i=1}^{N} \|\mathbf{x}_m^i - \bar{\mathbf{x}}_m + \bar{\mathbf{x}}_m - \mathbf{x}_m^j\| \leq \sum_{i=1}^{N} \|\mathbf{x}_m^i - \bar{\mathbf{x}}_m\| + N\|\bar{\mathbf{x}}_m - \mathbf{x}_m^j\|
$$

$$
\leq \left(\sqrt{N} + N\right) \sqrt{\sum_{i=1}^{N} \|\mathbf{x}_m^i - \bar{\mathbf{x}}_m\|^2}. \tag{32}
$$

Moreover, for any $i \in [N]$ and $m \in \{2, \ldots, T/K\}$, we have

$$
\|\bar{\mathbf{x}}_m - \mathbf{x}_m^i\|^2 \leq \|\bar{\mathbf{x}}_m - \hat{\mathbf{y}}_m + \hat{\mathbf{y}}_m - \tilde{\mathbf{y}}_m^i + \tilde{\mathbf{y}}_m^i - \mathbf{x}_m^i\|^2
$$

$$
\overset{(a)}{\leq} 3\|\bar{\mathbf{x}}_m - \hat{\mathbf{y}}_m\|^2 + 3\|\hat{\mathbf{y}}_m - \tilde{\mathbf{y}}_m^i\|^2 + 3\|\tilde{\mathbf{y}}_m^i - \mathbf{x}_m^i\|^2
$$

$$
\overset{(b)}{\leq} \frac{3}{N} \sum_{j=1}^{N} \|\mathbf{x}_m^j - \tilde{\mathbf{y}}_m^j\|^2 + 3\|\hat{\mathbf{y}}_m - \tilde{\mathbf{y}}_m^i\|^2 + 3\|\tilde{\mathbf{y}}_m^i - \mathbf{x}_m^i\|^2
$$

$$
\overset{(c)}{\leq} 18\epsilon + 3\|\hat{\mathbf{y}}_m - \tilde{\mathbf{y}}_m^i\|^2,
$$

where step (a) utilizes Cauchy-Schwarz inequality, step (b) utilizes Cauchy-Schwarz inequality and the definition of $\hat{\mathbf{y}}_m$ and $\bar{\mathbf{x}}_m$, and step (c) comes from Lemma 1. When $m = 1$, $\|\bar{\mathbf{x}}_1 - \mathbf{x}_1^i\|^2 = \mathbf{0} \leq 18\epsilon + 3\|\hat{\mathbf{y}}_m - \tilde{\mathbf{y}}_m^i\|^2$, which leads us to conclude that for any $m \in [T/K]$,

$$
\|\bar{\mathbf{x}}_m - \mathbf{x}_m^i\|^2 \leq 18\epsilon + 3\|\hat{\mathbf{y}}_m - \tilde{\mathbf{y}}_m^i\|^2.
$$

Thus, for any $m \in [T/K]$, we have

$$
\sqrt{\sum_{i=1}^{N} \|\bar{\mathbf{x}}_m - \mathbf{x}_m^i\|^2} \leq \sqrt{\sum_{i=1}^{N} (18\epsilon + 3\|\hat{\mathbf{y}}_m - \tilde{\mathbf{y}}_m^i\|^2)}
$$

$$
\overset{(a)}{\leq} 3\sqrt{2N\epsilon} + \sqrt{3 \sum_{i=1}^{N} \|\hat{\mathbf{y}}_m - \tilde{\mathbf{y}}_m^i\|^2}
$$

$$
\overset{(b)}{\leq} 3\sqrt{2N\epsilon} + \frac{\sqrt{3N}(3\eta KG + 2\sqrt{3}\epsilon)}{1 - \lambda_2} \tag{33}
$$

where step (a) is due to triangle inequality and step (b) follows by Equation (11) in Lemma 8.

Finally, by substituting Equation (33) into Equation (32), for any $m \in [T/K]$, we have

$$
\sum_{i=1}^{N} \|\mathbf{x}_m^i - \mathbf{x}_m^j\| \leq (\sqrt{N} + N) \sqrt{\sum_{i=1}^{N} \|\mathbf{x}_m^i - \bar{\mathbf{x}}_m\|^2}
$$

$$
\leq \left(3\sqrt{2\epsilon} + \frac{(3\eta KG + 2\sqrt{3}\epsilon)}{1 - \lambda_2}\right)(N^{3/2} + N).
$$

## C Bandit Feedback for Trivial Query Functions

In this section, we describe and discuss the variation of `DOCLO` to handle bandit feedback for functions with trivial query oracle. The detailed implementation is given in the Algorithm 2, and here we provide proof of Theorem 3. This algorithm requires additional input from the user: smoothing parameter $\delta \leq \alpha$, shrunk set $\hat{\mathcal{K}}_\delta$, linear space $\mathcal{L}_0$.

*Proof.* Let $\mathcal{A}'$ denote Algorithm 2 and $\mathcal{A}$ denote Algorithm 1. Let $\hat{f}_{t,j}$ denote a $\delta$-smoothed version of $f_{t,i}$. Let $\mathbf{x}^* \in \arg\max_{\mathbf{x} \in \mathcal{K}} \sum_{t=1}^{T} \sum_{j=1}^{N} f_{t,j}(\mathbf{x})$ and $\hat{\mathbf{x}}^* \in \arg\max_{\mathbf{x} \in \hat{\mathcal{K}}_\delta} \sum_{t=1}^{T} \sum_{j=1}^{N} \hat{f}_{t,j}(\mathbf{x})$. Following our description in Section 5.1, Algorithm 2 is equivalent to running Algorithm 1 on $\hat{f}_{t,i}$, a $\delta$ smoothed version of $f_{t,i}$ over a shrunk set $\hat{\mathcal{K}}_\delta$.

By the definition of regret, we have

$$
\begin{aligned}
\mathbb{E}\left[\mathcal{R}_\alpha^{i,\mathcal{A}'}\right] - \mathbb{E}\left[\mathcal{R}_\alpha^{i,\mathcal{A}}\right] &= \frac{1}{N}\mathbb{E}\left[\alpha \sum_{t=1}^{T}\sum_{j=1}^{N} f_{t,j}(\mathbf{x}^*) - \sum_{t=1}^{T}\sum_{j=1}^{N} f_{t,j}(h(\mathbf{x}_t^i) + \delta \mathbf{v}_t^i)\right] \\
&\quad - \frac{1}{N}\mathbb{E}\left[\alpha \sum_{t=1}^{T}\sum_{j=1}^{N} \hat{f}_{t,j}(\hat{\mathbf{x}}^*) - \sum_{t=1}^{T}\sum_{j=1}^{N} \hat{f}_{t,j}(h(\mathbf{x}_t^i))\right] \\
&= \frac{1}{N}\mathbb{E}\left[\left(\sum_{t=1}^{T}\sum_{j=1}^{N} \hat{f}_{t,j}(h(\mathbf{x}_t^i)) - \sum_{t=1}^{T}\sum_{j=1}^{N} f_{t,j}(h(\mathbf{x}_t^i) + \delta \mathbf{v}_t^i)\right)\right. \\
&\quad \left. + \alpha\left(\sum_{t=1}^{T}\sum_{j=1}^{N} f_{t,j}(\mathbf{x}^*) - \sum_{t=1}^{T}\sum_{j=1}^{N} \hat{f}_{t,j}(\hat{\mathbf{x}}^*)\right)\right].
\end{aligned}
\tag{34}
$$

Based on Lemma 3 proved by Pedramfar et al. (2023), we have $|\hat{f}_{t,j}(h(\mathbf{x}_t^i)) - f_{t,j}(h(\mathbf{x}_t^i))| \leq \delta M_1$, and $\hat{f}_{t,j}$ is $M_1$-Lipschitz continuous as well. Thus, we have

$$
|f_{t,j}(h(\mathbf{x}_t^i) + \delta \mathbf{v}_t^i) - \hat{f}_{t,j}(h(\mathbf{x}_t^i))| \leq |f_{t,j}(h(\mathbf{x}_t^i) + \delta \mathbf{v}_t^i) - f_{t,j}(h(\mathbf{x}_t^i))| + |f_{t,j}(h(\mathbf{x}_t^i)) - \hat{f}_{t,j}(h(\mathbf{x}_t^i))| \leq 2\delta M_1.
\tag{35}
$$

Meanwhile, for the second part of Equation 34, we have

$$
\begin{aligned}
\sum_{t=1}^{T}\sum_{j=1}^{N} \hat{f}_{t,j}(\hat{\mathbf{x}}^*) &= \max_{\hat{\mathbf{x}} \in \hat{\mathcal{K}}_\alpha} \sum_{t=1}^{T}\sum_{j=1}^{N} \hat{f}_{t,j}(\hat{\mathbf{x}}) \\
&\stackrel{(a)}{\geq} -N\delta M_1 T + \max_{\hat{\mathbf{x}} \in \hat{\mathcal{K}}_\alpha} \sum_{t=1}^{T}\sum_{j=1}^{N} f_{t,j}(\hat{\mathbf{x}}) \\
&\stackrel{(b)}{=} -N\delta M_1 T + \max_{\mathbf{x} \in \mathcal{K}} \sum_{t=1}^{T}\sum_{j=1}^{N} f_{t,j}\left(\left(1 - \frac{\delta}{r}\right)\mathbf{x} + \frac{\delta}{r}\mathbf{c}\right) \\
&= -N\delta M_1 T + \max_{\mathbf{x} \in \mathcal{K}} \sum_{t=1}^{T}\sum_{j=1}^{N} f_{t,j}\left(\mathbf{x} + \frac{\delta}{r}(\mathbf{c} - \mathbf{x})\right) \\
&\stackrel{(c)}{\geq} -N\delta M_1 T + \max_{\mathbf{x} \in \mathcal{K}} \sum_{t=1}^{T}\sum_{j=1}^{N} \left(f_{t,j}(\mathbf{x}) - \frac{4\delta M_1 R}{r}\right) \\
&= -\left(1 + \frac{4R}{r}\right)N\delta M_1 T + \sum_{t=1}^{T}\sum_{j=1}^{N} f_{t,j}(\mathbf{x}^*)
\end{aligned}
$$

where step (a) follows from Lemma 3 by Pedramfar et al. (2023), step (b) follows from the definition of $\hat{\mathcal{K}}_\delta$, and step (c) is due to the $M_1$-Lipschitz continuity of $f_{t,i}$'s.

Putting it together with Equation 34 and Equation 35, we have

$$
\mathcal{R}_\alpha^{i,\mathcal{A}'} - \mathcal{R}_\alpha^{i,\mathcal{A}} \leq \frac{1}{N}\left(2N\delta M_1 T + \left(1 + \frac{4R}{r}\right)N\delta M_1 T\right) = \left(3 + \frac{4R}{r}\right)\delta M_1 T.
$$

Thus we have

$$\mathcal{R}_\alpha^{i,\mathcal{A}'} \le \mathcal{R}_\alpha^{i,\mathcal{A}} + \left(3 + \frac{4R}{r}\right)\delta M_1 T.$$

Assuming the zeroth order oracle is bounded by $B_0$, from Line 10 of Algorithm 2 we see that the gradient sample that is being passed to $\mathcal{A}$ is bounded by $G = \frac{k}{\delta}B_0 = O(\delta^{-1})$. Substituting results from Theorem 1, we see that

$$\mathbb{E}\left[\mathcal{R}_\alpha^{i,\mathcal{A}'}\right] = O\left(\mathcal{R}_\alpha^{i,\mathcal{A}} + \delta T\right) = O\left(\frac{1}{\eta} + \eta TK\delta^{-2} + \delta T\right).$$

Since we are doing same amount of infeasible projection operation and communication operation in Algorithm 2, LOO calls and communication complexity for Algorithm 2 remains the same as Algorithm 1. □

## D   Semi-Bandit Feedback for Non-Trivial Query Functions

To transform `DOCLO` into an algorithm that can handle semi-bandit feedback when we are dealing with functions with non-trivial queries, we pass $\frac{T}{L}$ as time horizon to `DOCLO`. In each of those $\frac{T}{L}$ blocks, we consider functions $\left(\hat{f}_{q,i}\right)_{1\le q\le T/L, 1\le i\le N}$, where $\hat{f}_{q,i} = \frac{1}{L}\sum_{t=(q-1)L+1}^{qL} f_{t,i}$. We note that in Algorithm 3, for any $\mathbf{x}\in\mathcal{K}$ and $1\le q\le T/L$, we have $\mathbb{E}[f_{t'_q}(\mathbf{x})] = \hat{f}_q(\mathbf{x})$, and if $f_t$ are differentiable, $\mathbb{E}[\nabla f_{t'_q}(\mathbf{x})] = \nabla\hat{f}_q(\mathbf{x})$. This way, the transformed algorithm only queries once at the point of action per block, thus semi-bandit.

*Proof of Theorem 4.* Let $\mathcal{A}'$ denote Algorithm 3 and $\mathcal{A}$ denote Algorithm 1. Following our description in Section 5.2, Algorithm 3 is equivalent to running Algorithm 1 on $\hat{f}_{q,i}(\mathbf{x}) = \frac{1}{L}\sum_{t=(q-1)L+1}^{qL} f_{t,i}(\mathbf{x})$, an average of $f_{t,i}$ over block $q$. In consistence with Algorithm 3 description, we let $\mathbf{z}_t^i$ denote the action taken by agent $i$ at time-step $t$, whether it be $\hat{\mathbf{x}}_q$, point of action selected by `DOCLO`, or $\hat{\mathbf{y}}_q$, point of query selected by `DOCLO`. Thus, the regret of Algorithm 3 over horizon $T$ is

$$
\begin{aligned}
\mathbb{E}\left[\mathcal{R}_{\alpha,T}^{i,\mathcal{A}'}\right] &= \frac{1}{N}\mathbb{E}\left[\alpha\max_{\mathbf{u}\in\mathcal{K}}\sum_{t=1}^{T}\sum_{j=1}^{N} f_{t,j}(\mathbf{u}) - \sum_{t=1}^{T}\sum_{j=1}^{N} f_{t,j}(\mathbf{z}_t^i)\right] \\
&= \frac{L}{N}\mathbb{E}\left[\alpha\max_{\mathbf{u}\in\mathcal{K}}\frac{1}{L}\sum_{t=1}^{T}\sum_{j=1}^{N} f_{t,j}(\mathbf{u}) - \frac{1}{L}\sum_{t=1}^{T}\sum_{j=1}^{N} f_{t,j}(\mathbf{z}_t^i)\right] \\
&= \frac{L}{N}\mathbb{E}\left[\alpha\max_{\mathbf{u}\in\mathcal{K}}\frac{1}{L}\sum_{j=1}^{N}\sum_{q=1}^{T/L}\sum_{t=(q-1)L+1}^{qL} f_{t,j}(\mathbf{u}) - \frac{1}{L}\sum_{j=1}^{N}\sum_{q=1}^{T/L}\sum_{t=(q-1)L+1}^{qL} f_{t,j}(\mathbf{z}_t^i)\right] \\
&= \frac{1}{N}\mathbb{E}\left[\sum_{j=1}^{N}\sum_{q=1}^{T/L}\sum_{t=(q-1)L+1}^{qL}\left(f_{t,j}(\hat{\mathbf{x}}_q^i) - f_{t,j}(\mathbf{z}_t^i)\right) + L\left(\alpha\max_{\mathbf{u}\in\mathcal{K}}\sum_{j=1}^{N}\sum_{q=1}^{T/L}\hat{f}_{q,j}(\mathbf{u}) - \sum_{j=1}^{N}\sum_{q=1}^{T/L}\hat{f}_{q,j}(\hat{\mathbf{x}}_q^i)\right)\right]
\end{aligned}
\tag{36}
$$

Algorithm 3 ensures that in each block with a given $q$, there is only 1 iteration where $\mathbf{z}_t^i \ne \hat{\mathbf{x}}_q^i$, otherwise $\mathbf{z}_t^i = \hat{\mathbf{x}}_q^i$. Since $f_{t,j}$ are $M_1$-Lipschitz continuous, we have $|f_{t,j}(\hat{\mathbf{x}}_q^i) - f_{t,j}(\hat{\mathbf{y}}_t^i)| \le M_1|\hat{\mathbf{x}}_q^i - \hat{\mathbf{y}}_t^i| \le 2M_1 R$ where the second equation comes from the restraint on $\mathcal{K}$. Since $\hat{\mathcal{K}}_\delta \subseteq \mathcal{K}$, we have $\max_{\mathbf{x}\in\hat{\mathcal{K}}_\delta}\|\mathbf{x}\| \le \max_{\mathbf{x}\in\mathcal{K}}\|\mathbf{x}\| = R$. Thus, we have

$$\sum_{j=1}^{N}\sum_{q=1}^{T/L}\sum_{t=(q-1)L+1}^{qL}\left|f_{t,j}(\hat{\mathbf{x}}_q^i) - f_{t,j}(\mathbf{x}_t^i)\right| \le \sum_{j=1}^{N}\sum_{q=1}^{T/L}\left(0*(L-1) + 2M_1 R*1\right) = \frac{2NTM_1 R}{L} \tag{37}$$

The second part of Equation 36 can be seen as the regret of running Algorithm 1 against $\left(\hat{f}_{q,i}\right)_{1 \leq q \leq T/L, 1 \leq i \leq N}$, over horizon $T/L$ instead of $T$. We denote it with $\mathcal{R}_{\alpha, T/L}^{i, \mathcal{A}}$. Applying Theorem 1, we have

$$\mathbb{E}\left[R_{\alpha, T/L}^{i, \mathcal{A}}\right] = O\left(\frac{1}{\eta} + \frac{\eta T K G^2}{L}\right).$$

Putting together with Equation 36 and Equation 37, we have

$$\mathbb{E}\left[\mathcal{R}_{\alpha, T}^{i, \mathcal{A}'}\right] \leq \frac{2 T M_1 R}{L} + L \mathbb{E}\left[\mathcal{R}_{\alpha, T/L}^{i, \mathcal{A}}\right]$$

which means

$$\mathbb{E}\left[\mathcal{R}_\alpha^i\right] = O\left(\frac{L}{\eta} + \eta T K G^2 + \frac{T}{L}\right).$$

Based on the implementation described in Algorithm 3, it queries oracle every $L$ iterations, and communicate and make updates with infeasible projection operation every $KL$ iteration. Thus, communication complexity for Algorithm 3 is $O(\frac{T}{KL})$, while LOO calls are $O(\frac{T}{\epsilon KL})$.

$\square$

# E  Zeroth Order Full-Information Feedback for Non-Trivial Query Functions

In this section, we describe and discuss the variation of DOCLO to handle zeroth-order full-information feedback for functions with non-trivial query oracle. The detailed implementation is given in the Algorithm 4 table, followed by proof of Theorem 5. Per request of FOTZO, Algorithm 4 requires additional input from the user: smoothing parameter $\delta \leq \alpha$, shrunk set $\hat{\mathcal{K}}_\delta$, linear space $\mathcal{L}_0$. In the case of non-trivial query oracle, $h(\cdot)$ is not necessarily an identity function. In the following, we give proof of Theorem 5.

*Proof of Theorem 5.* Let $\mathcal{A}'$ denote Algorithm 4 and $\mathcal{A}$ denote Algorithm 1. Let $\hat{f}_{t,j}$ denote a $\delta$-smoothed version of $f_{t,i}$. Let $\mathbf{x}^* \in \operatorname{argmax}_{\mathbf{x} \in \mathcal{K}} \sum_{t=1}^T \sum_{j=1}^N f_{t,j}(\mathbf{x})$ and $\hat{\mathbf{x}}^* \in \operatorname{argmax}_{\mathbf{x} \in \hat{\mathcal{K}}_\delta} \sum_{t=1}^T \sum_{j=1}^N \hat{f}_{t,j}(\mathbf{x})$. Following our description in Section 5.3, Algorithm 4 is equivalent to running Algorithm 1 on $\hat{f}_{t,i}$, a $\delta$ smoothed version of $f_{t,i}$ over a shrunk set $\hat{\mathcal{K}}_\delta$.

By the definition of regret, we have

$$
\begin{aligned}
\mathbb{E}\left[\mathcal{R}_\alpha^{i, \mathcal{A}'}\right] - \mathbb{E}\left[\mathcal{R}_\alpha^{i, \mathcal{A}}\right] &= \frac{1}{N} \mathbb{E}\left[\alpha \sum_{t=1}^T \sum_{j=1}^N f_{t,j}(\mathbf{x}^*) - \sum_{t=1}^T \sum_{j=1}^N f_{t,j}(h(\mathbf{x}_t^i))\right] \\
&\quad - \frac{1}{N} \mathbb{E}\left[\alpha \sum_{t=1}^T \sum_{j=1}^N \hat{f}_{t,j}(\hat{\mathbf{x}}^*) - \sum_{t=1}^T \sum_{j=1}^N \hat{f}_{t,j}(h(\mathbf{x}_t^i))\right] \\
&= \frac{1}{N} \mathbb{E}\left[\left(\sum_{t=1}^T \sum_{j=1}^N \hat{f}_{t,j}(h(\mathbf{x}_t^i)) - \sum_{t=1}^T \sum_{j=1}^N f_{t,j}(h(\mathbf{x}_t^i))\right)\right. \\
&\quad \left. + \alpha\left(\sum_{t=1}^T \sum_{j=1}^N f_{t,j}(\mathbf{x}^*) - \sum_{t=1}^T \sum_{j=1}^N \hat{f}_{t,j}(\hat{\mathbf{x}}^*)\right)\right].
\end{aligned}
\tag{38}
$$

Based on Lemma 3 proved by Pedramfar et al. (2023), we have $|\hat{f}_{t,j}(h(\mathbf{x}_t^i)) - f_{t,j}(h(\mathbf{x}_t^i))| \leq \delta M_1 < 2\delta M_1$.

It can be shown that the second part of Equation 38 follows the same upper bound as Equation 35.

Thus, putting it together, we have for Algorithm 2, the regret

$$\mathcal{R}_\alpha^{i,\mathcal{A}'} \le \mathcal{R}_\alpha^{i,\mathcal{A}} + \frac{1}{N}\left(2N\delta M_1 T + \left(1 + \frac{4R}{r}\right)N\delta M_1 T\right) = \mathcal{R}_\alpha^{i,\mathcal{A}} + \left(3 + \frac{4R}{r}\right)\delta M_1 T$$

Assuming the zeroth order oracle is bounded by $B_0$, from Line 10 of Algorithm 4 we see that the gradient sample that is being passed to $\mathcal{A}$ is bounded by $G = \frac{k}{\delta}B_0 = O(\delta^{-1})$. Substituting results from Theorem 1, we see that

$$\mathbb{E}\left[\mathcal{R}_\alpha^i\right] = O\left(\frac{1}{\eta} + \eta T K \delta^{-2} + \delta T\right).$$

Since we are doing same amount of infeasible projection operation and communication operation in Algorithm 4, LOO calls and communication complexity for Algorithm 4 remains the same as Algorithm 1. $\qquad\square$

## F  Bandit Feedback for Non-trivial Query Function

In this section, we extend `DOCLO` over functions with nontrivial query oracle to handle bandit feedback, i.e., trivial query and zero-order feedback. We achieve this by applying `FOTZO` to handle zero-order full-information feedback, then applying `SFTT` to transform the algorithm into trivial queries. In the following, we provide proof of Theorem 6.

*Proof of Theorem 6.* Let $\mathcal{A}'$ denote Algorithm 5 and $\mathcal{A}$ denote Algorithm 4. Following our description in Section 5.4, Algorithm 5 is equivalent to running Algorithm 4 on $\hat{f}_{q,i}(\mathbf{x}) = \frac{1}{L}\sum_{t=(q-1)L+1}^{qL}\hat{f}_{t,i}(\mathbf{x})$, an average of $\hat{f}_{t,i}$ over block $q$, where $\hat{f}_{t,i}$ is a $\delta$-smoothed version of $f_{t,i}$. In consistence with Algorithm 5 description, we let $\mathbf{x}_t^i$ denote the action taken by agent $i$ at iteration $t$, whether it be $\hat{\mathbf{x}}_q$, point of action selected by `DOCLO`, or $\hat{\mathbf{y}}_q$, point of query selected by `DOCLO`. Thus, the regret of Algorithm 5 over horizon T:

$$
\begin{aligned}
\mathbb{E}\left[\mathcal{R}_\alpha^{i,\mathcal{A}'}\right] &= \frac{1}{N}\mathbb{E}\left[\alpha\max_{\mathbf{u}\in\mathcal{K}}\sum_{t=1}^{T}\sum_{j=1}^{N}\hat{f}_{t,j}(\mathbf{u}) - \sum_{t=1}^{T}\sum_{j=1}^{N}\hat{f}_{t,j}(\mathbf{z}_t^i)\right]\\
&= \frac{L}{N}\mathbb{E}\left[\alpha\max_{\mathbf{u}\in\mathcal{K}}\frac{1}{L}\sum_{t=1}^{T}\sum_{j=1}^{N}\hat{f}_{t,j}(\mathbf{u}) - \frac{1}{L}\sum_{t=1}^{T}\sum_{j=1}^{N}\hat{f}_{t,j}(\mathbf{z}_t^i)\right]\\
&= \frac{L}{N}\mathbb{E}\left[\alpha\max_{\mathbf{u}\in\mathcal{K}}\frac{1}{L}\sum_{j=1}^{N}\sum_{q=1}^{T/L}\sum_{t=(q-1)L+1}^{qL}\hat{f}_{t,j}(\mathbf{u}) - \frac{1}{L}\sum_{j=1}^{N}\sum_{q=1}^{T/L}\sum_{t=(q-1)L+1}^{qL}\hat{f}_{t,j}(\mathbf{z}_t^i)\right]\\
&= \frac{1}{N}\mathbb{E}\left[\sum_{j=1}^{N}\sum_{q=1}^{T/L}\sum_{t=(q-1)L+1}^{qL}\left(\hat{f}_{t,j}(\hat{\mathbf{x}}_q^i) - \hat{f}_{t,j}(\mathbf{z}_t^i)\right)\right.\\
&\qquad\left.+L\left(\alpha\max_{\mathbf{u}\in\mathcal{K}}\sum_{j=1}^{N}\sum_{q=1}^{T/L}\hat{f}_{q,j}(\mathbf{u}) - \sum_{j=1}^{N}\sum_{q=1}^{T/L}\hat{f}_{q,j}(\hat{\mathbf{x}}_q^i)\right)\right]
\end{aligned}
\tag{39}
$$

Algorithm 3 ensures that in each block $q$, there is only 1 iteration where $\mathbf{z}_t^i = \hat{\mathbf{y}}_t^i \ne \hat{\mathbf{x}}_t^i$, otherwise $\mathbf{x}_t^i = \hat{\mathbf{x}}_t^i$. Based on Lemma 3 proposed by Pedramfar et al. (2023), $\hat{f}_{t,j}$ is $M_1$-Lipscitz continuous if $f_{t,i}$ is $M_1$-Lipscitz continuous, i.e., $|f_{t,j}(\hat{\mathbf{x}}_q^i) - f_{t,j}(\hat{\mathbf{y}}_t^i)| \le M_1|\hat{\mathbf{x}}_q^i - \hat{\mathbf{y}}_t^i| \le 2M_1 R$ where the second equation comes from the restraint on $\mathcal{K}$. Since $\hat{\mathcal{K}}_\delta \subseteq \mathcal{K}$, we have $\max_{\mathbf{x}\in\hat{\mathcal{K}}_\delta}\|\mathbf{x}\| \le \max_{\mathbf{x}\in\mathcal{K}}\|\mathbf{x}\| = R$. Thus, we have

$$\sum_{j=1}^{N}\sum_{q=1}^{T/L}\sum_{t=(q-1)L+1}^{qL}\left(\hat{f}_{t,j}(\hat{\mathbf{x}}_q^i) - \hat{f}_{t,j}(\mathbf{x}_t^i)\right) = \sum_{j=1}^{N}\sum_{q=1}^{T/L}\left(0*(L-1) + 2M_1 R*1\right) = \frac{2NTM_1 R}{L}\tag{40}$$

The second part of Equation 39 can be seen as the regret of running Algorithm 4 against $\left(\hat{f}_{q,i}\right)_{1 \leq q \leq T/L, 1 \leq i \leq N}$, over horizon $T/L$ instead of $T$. We denote it with $\mathcal{R}^{i,\mathcal{A}}_{\alpha,T/L}$. Applying Theorem 5, we have

$$\mathbb{E}\left[R^{i,\mathcal{A}}_{\alpha,T/L}\right] = O\left(\frac{1}{\eta} + \frac{\eta T K \delta^{-2}}{L} + \frac{\delta T}{L}\right).$$

Putting it together with Equation 36 and Equation 37, we have

$$\mathbb{E}\left[\mathcal{R}^{i,\mathcal{A}'}_{\alpha}\right] \leq \frac{2TM_1 R}{L} + L\mathbb{E}\left[\mathcal{R}^{i,\mathcal{A}}_{\alpha,T/L}\right],$$

which means that

$$\mathbb{E}\left[\mathcal{R}^i_{\alpha}\right] = O\left(\frac{L}{\eta} + \eta T K \delta^{-2} + \delta T + \frac{T}{L}\right).$$

Based on the implementation described in Algorithm 3, it queries oracle every $L$ iterations, and communicate and make updates with infeasible projection operation every $KL$ iteration. Thus, communication complexity for Algorithm 3 is $O(\frac{T}{KL})$, while LOO calls are $O(\frac{T}{\epsilon KL})$. □

