# OpenReview forum: "Decentralized Projection-free Online Upper-Linearizable Optimization with Applications to DR-Submodular Optimization"
_TMLR — Rejected by TMLR_

### Review · Reviewer_UYTH · 2025-05-19

**Summary Of Contributions:**

The broad context for this paper is online learning, where the typical "regret" notion is formulated using a sum of loss functions which are not convex, but fall into the category of "weakly up-concave functions". The particular focus of this paper is that of the *decentralized* setting, where there are $N$ separate agents, each of which receives their own (potentially adversarial) loss function at each round, and the ultimate goal is to obtain a small regret, having aggregated over all agents, noting that communication between agents is allowed.

To the best of my understanding, the main results are chiefly technical developments building directly upon existing work, in particular the algorithm design strategy of Pedramfar and Aggarwal (2024) used for "upper-linearizable" functions, originally in the centralized setting. Their results take the form of upper bounds on the (expected) regret associated with a general-purpose algorithm (their Algorithm 1), with different algorithmic settings used for different problem settings (e.g., different types of feedback availability), and they "parameterize" these bounds (see for example $\\theta$ in Table 1) to emphasize how regret bound and communication cost tradeoffs interact.

It should also be noted that another important trait of their approach is that it is "projection free"; the cost that is incurred instead is reflected via the "LOO calls" cost, namely the number of calls to a linear optimization oracle.

**Audience:**

Yes

**Broader Impact Concerns:**

Not applicable.

**Claims And Evidence:**

Yes

**Requested Changes:**

Please see the comments raised above. The paper represents a genuine effort, but I think the technical exposition needs more polish.

**Strengths And Weaknesses:**

The authors are quite methodical in organizing their exposition of previous literature and describing its relation to the new results obtained in this paper. The overall structure of the paper is good, with a key proof sketch, and highlights of key results summarized in a concise and fairly effective way.

On the other hand, I think the technical exposition still needs a fair bit of polish. Below I have listed a handful of points I tripped up on while reading.

- Page 1: the term "DR-submodular" is used throughout, but "DR" is not defined anywhere.

- Page 2, second last paragraph ("In this paper, ..."): the parameter $\\theta$ is given implicitly; I think it should receive some kind of comment, saying how it allows for explicit control of tradeoffs.

- Table 1 (and elsewhere): I take it that Table 1 presents existing known results only for projection free methods; is this correct? I think it would be useful to comment on how known guarantees change (if it all) when using projection-based methods in the problem setting of interest.

__Page 4:__
- First case of "eigen-value" is hyphenated.
- $\\mathcal{F}$ is given implicitly, only to be given more structure later.
- The authors start with high generality in the "agent query", allowing for $\\mathbf{x}\_{t}^{i}$ and $\\mathbf{y}\_{t}^{i}$ to be distinct outside the "trivial" agent query function case, but then in the following paragraph give the ultimate objective in terms of a sum of losses evaluated at $\\mathbf{x}\_{t}^{i}$ rather than $\\mathbf{y}\_{t}^{i}$; is the "non-trivial" case even needed here?
- In the last paragraph, the actions penalized by the aggregated loss function are missing their $t$ subscript, i.e., just $\\mathbf{x}^{i}$.

__Page 5:__
- The regret formulated at the top of the page assumes that the functions $f\_{t,j}$ return values where larger values are *more desirable*, correct? If so, why are these called "losses", and not "rewards" or something similar?
- The notion of $\\alpha$-regret is given here without additional comment; does this $\\alpha$-based generality have any meaning in this paper whatsoever? It doesn't appear anywhere else. Furthermore, in the critical definition of upper-linearizable functions, once again "$\\alpha$" appears, but it is not clear if the authors intend this $\\alpha$ to align with the $\\alpha$ that appears in the regret definition. All this is sloppy formulation.
- The query function is assumed to be "$L\_{1}$-Lipschitz", but is this with respect to both arguments?
- The authors use the terms "first-order oracle" without additional comment, in fact, they use this term to *define* their term "semi-bandit feedback", but I think using an undefined term to define a new term is bad practice. Yes, I know what the authors have in mind, but clarity is important.
- The authors say that the query function is Lipschitz continuous if "the up-concave function is $L$-smooth", but if you aren't going to show how the Lipschitz coefficient depends on smoothness coefficient $L$, why even bother write $L$ explicitly?
- Lemma 1: I know what the authors intend by $R\\mathcal{B}$, but this notation is undefined (the "product" of a scalar and a set).

__Algorithm 1:__
- $\\mathcal{T}\_{m}$ is undefined, it seems?
- $\\mathcal{N}\_{i}$ doesn't match $N\_{i}$ defined at the start of section 3.
- Is $h(\\mathbf{x}\_{m}^{i})$ supposed to correspond to $\\mathbf{y}\_{m}^{i}$ for the inner loop rounds? Unclear to me if this has any bearing elsewhere on the algorithm.
- Is the notation $\\mathcal{G}$ really needed here? Everything of essence seems to be characterized by the assumptions made on the noisy query response, unless I'm missing something.

__Misc.__

- The regret bounds on page 6 (and onward) are for "$R\_{\\alpha}^{i}$"; this should be $\\mathcal{R}\_{\\alpha}^{i}$, correct?

---

> ### Author Response · Authors · 2025-06-04
>
> Thank you for taking the time to review our paper. We have revised paper based on your comments, uploaded the revised version for your review, and will address your questions in the following:
>
> >Page 1: the term "DR-submodular" is used throughout, but "DR" is not defined anywhere.
>
> We have changed the first "DR-submodular" to "Diminishing-Return Submodular (DR-submodular)" in the paper.
>
> > Page 2: second last paragraph ("In this paper, ..."): the parameter $\theta$ is given implicitly; I think it should receive some kind of comment, saying how it allows for explicit control of tradeoffs.
>
> In the revision, after the ("In this paper, ...") sentence we have added: "In particular, for optimizing monotone upper-linearizable functions over convex set containing the origin, ... , where $\theta \in [0,1]$ is a trade-off parameter that controls the blocking effect. When $\theta$ decreases, the total calls to the linear optimization oracle would decrease, while regret and communication complexity would increase." And then the sentence is followed by ("We note that for $\theta=1$, ...")
>
> > Table 1 (and elsewhere): I take it that Table 1 presents existing known results only for projection free methods; is this correct? I think it would be useful to comment on how known guarantees change (if it all) when using projection-based methods in the problem setting of interest.
>
> Yes, all the results summarized in Table 1 are projection-free methods, as indicated in the title of Table 1: "Decentralized *Projection-free* up-concave maximization". Since our methods are projection-free, the table focused on projection-free methods for comparison.
>
> For the comparison to the projection-based methods, we have mentioned this comparison in the related work. We have "On the other hand, the projection-based Decentralized Online Boosting Gradient Ascent algorithm proposed in Zhang et al. (2023a) obtains $(1 − 1/e)$-regret of $O(\sqrt{T})$ with a communication complexity of $O(T)$." Note that we have also mentioned that "In first order full-information case, for $\theta = 1$, we show that the regret and the communication complexity matches the best known projection-based algorithm in (Zhang et al., 2023a)". The projection based method has only been studied in this case, where the results are no better than than achieved by our projection-free results. The other setups have not been studied for projection-based approaches either, to the best of our knowledge.
>
>
> > Page 4: First case of "eigen-value" is hyphenated.
>
> We have changed it to "eigenvalue".
>
> > Page 4:  $\mathcal{F}$ is given implicitly, only to be given more structure later.
>
> We have added the definition of $\mathcal{F}$ in **Notations and Definitions** in Preliminaries sections: `Given a set $\mathcal{K}$, we define a function class $\mathcal{F}$ as a subset of all real-valued functions over $\mathcal{K}$.", and then introduce the structure formally in Assumptions environment in Assumption 2.
>
> > Page 4: The authors start with high generality in the "agent query", allowing for $\mathbf{x}_t^i$ and $\mathbf{y}_t^i$ to be distinct outside the "trivial" agent query function case, but then in the following paragraph give the ultimate objective in terms of a sum of losses evaluated at $\mathbf{x}_t^i$ rather than $\mathbf{y}_t^i$; is the "non-trivial" case even needed here?
>
> Although the metric *regret* is defined on the action played, $\mathbf{x}_ t ^i$, the point of query $\mathbf{y}_ t^i$ will change the information we receive from the oracle, and hence add complexity. In the "Trivial" case, i.e., if we query at the point of action, we would observe $f_ {t,i}(\mathbf{y}_ t^i)=f_ {t,i}(\mathbf{x}_ t^i)$, and we can directly apply it to *regret* calculation. In the "Non-Trivial" case, since $\mathbf{x}_ t^i \neq \mathbf{y}_ t^i$, we cannot observe $f_ {t,i}(\mathbf{x}_ t^i)$, as we only know $f_ {t,i}(\mathbf{y}_ t^i)$, which makes it non-trivial since $f_ {t,i}(\mathbf{x}_ t^i)$ is necessary for *regret* calculation, and we would add extra layers to the algorithm to deal with that. For example, Algorithm 6 and 7 require querying at a point other than $\mathbf{x}_ t^i \neq \mathbf{y}_ t^i$. The query is at $z*\mathbf{x}_ t^i$ for Algorithm~6 and $\frac{z}{2}(\mathbf{x}_ t^i-\underline{\mathbf{x}_ t^i})+\underline{\mathbf{x}_ t^i}$.
>
>
> > Page 4: In the last paragraph, the actions penalized by the aggregated loss function are missing their $t$ subscript, i.e., just $\mathbf{x}^i$.
>
> Thanks for pointing this out, in the revised version, we have added the subscript $t$ to the action.
>
> > Page 5: The regret formulated at the top of the page assumes that the functions $f_{t,j}$ return values where larger values are more desirable, correct? If so, why are these called "losses", and not "rewards" or something similar?
>
> We agree that we are maximizing the objective, so it is not a loss. We have changed our reference to $f_{t,i}$ to *objective functions* instead of *losses*.
>
> (to be continued)

---

> ### Author Response · Authors · 2025-06-04
> **Response 2 for Reviewer UYTH**
>
> (following response 1)
>
> > Page 5: The notion of $\alpha$-regret is given here without additional comment; does this $\alpha$-based generality have any meaning in this paper whatsoever? It doesn't appear anywhere else. Furthermore, in the critical definition of upper-linearizable functions, once again $\alpha$ appears, but it is not clear if the authors intend this $\alpha$ to align with the $\alpha$ that appears in the regret definition. All this is sloppy formulation.
>
> We note that the problem is NP-hard even in the centralized offline setup.
> For example, when $f$ is a monotone continuous DR-submodular function and $\mathcal{K} \subseteq [0,1]^d$ contains the origin, finding a point $\mathbf{x} \in \mathcal{K}$ such that $f(\mathbf{x})$ is optimal is NP-hard.
> More generally, for any $\epsilon > 0$, finding any point $\mathbf{x} \in \mathcal{K}$ such that $f(\mathbf{x})$ is at least $(1 - e^{-1} + \epsilon)$ times the optimal value is NP-hard. (See Proposition 3 in [1])
> However, there are polynomial times algorithms that achieve the ratio of $1 - e^{-1}$.
> The ratio with this property is refered to as the *optimal approximation ratio*, in this case being $1 - e^{-1}$.
> In the three settings considered in this paper, (in the special case of $\gamma=1$) the approximation ratios for monotone function over convex sets containing the origin and non-monotone functions over general convex sets are known to be optimal (See [1] and [2])
> In the case of monotone functions over general convex sets (when $\gamma=1$), the approximation ratio of $1/2$ is the best known in the literature so far and it is conjectured to be optimal (See [3])
>
> We note that the notion of approximation ratio dates further back than the works we mentioned.
> For example, optimization of discrete submodular setting is more widely studied than its continuous counterpart and, for monotone set functions with cardinality constraints, it has been proven by [4] that finding $(1 - e^{-1} + \epsilon)$ times the optimal value is NP-hard, while a greedy algorithm achieves the approximation ratio of $1 - e^{-1}$.
>
> We note that this $\alpha$ is intended to be the same in the definition of upper linearizable function, since both indicate the approximation coefficient. With the $\alpha$ in the current place in the definition of linearizable function, we find $\alpha$-regret. Thus, both are meant to have same meanings, which justifies the use of the same variable.
>
> [1] *Guaranteed Non-convex Optimization: Submodular Maximization over Continuous Domains*, Bian et al., 2017
>
> [2] *Resolving the Approximability of Offline and Online Non-monotone DR-Submodular Maximization over General Convex Sets*, Mualem \\& Feldman, 2023.
>
> [3] *A Unified Approach for Maximizing Continuous DR-submodular Functions*, Pedramfar et al., 2023.
>
> [4] *A threshold of ln n for approximating set cover*, Feige, 1998.
>
> > Page 5: The query function is assumed to be "$L_1$-Lipschitz", but is this with respect to both arguments?
>
> It is $L_1$-Lipschitz continuous with respect to the second argument, and we have added this clarification when we introduce the Assumption 4: "We further assume that the query function $\mathfrak{g}(f_{t,i},\mathbf{x})$ is $L_1$-Lipschitz  with respect to the second argument, $\mathbf{x}$."
>
> > The authors use the terms "first-order oracle" without additional comment, in fact, they use this term to define their term "semi-bandit feedback", but I think using an undefined term to define a new term is bad practice. Yes, I know what the authors have in mind, but clarity is important.
>
> In the revision, we add "We say the oracle is *first-order* if the oracle gives information about the gradient of the $f_{t,i}$. We say the oracle is *zero-order* if the oracle gives information about the value of $f_{t,i}$." before defining feedback types.
>
> > The authors say that the query function is Lipschitz continuous if "the up-concave function is $L$-smooth", but if you aren't going to show how the Lipschitz coefficient depends on smoothness coefficient $L$, why even bother write $L$ explicitly?
>
> Thank you for pointing this out, we have removed $L$ in the revised version and intstead used "Lipschitz smooth".
>
> > Lemma 1: I know what the authors intend by $R\mathcal{B}$, but this notation is undefined (the "product" of a scalar and a set).
>
> We have added explanation "(where $R\mathcal{B}$ means a ball with radius $R$)".
>
> (to be continued)

---

> ### Author Response · Authors · 2025-06-04
> **Response 3 for Reviewer UYTH**
>
> (following response 2)
>
> > Algorithm 1: $\mathcal{T}_m$ is undefined, it seems?
>
> We have added "we denote the set of all time steps in block $m$ as $\mathcal{T}_m$".
>
> > Algorithm 1: $\mathcal{N}_i$ doesn't match $N_i$ defined at the start of section 3.
>
> We have changed the notation $N_i$ in the beginning of section 3 to $\mathcal{N}_i$
>
> > Algorithm 1: Is $h(\mathbf{x}_m^i)$ supposed to correspond to $\mathbf{y}_m^i$ for the inner loop rounds? Unclear to me if this has any bearing elsewhere on the algorithm.
>
> We understand that there was a confusion due to notations. The $\mathbf{y}_t^i$ in the non-trivial query is determined by the query algorithm ($z*\mathbf{x}$ for Algorithm 6 and $\frac{z}{2}(\mathbf{x}-\underline{\mathbf{x}})+\underline{\mathbf{x}}$ for Algorithm 7), but the $\mathbf{y}_m^i$ in the inner loop is not the same. In the revision, we have changed the notation for the query point to $\mathbf{w}_t^i$ rather than $\mathbf{y}_t^i$ and then the non-trivial query here is $\mathbf{w}_t^i$ while $\mathbf{y}_m^i$ in the algorithm has a different meaning unrelated to the query function.
> Also note that $\mathbf{w}_t^i \neq h(\mathbf{x}_t^i)$ (for Case (ii), monotone up-concave over convex set containing the origin, the point of query is $z*\mathbf{x}$ but $h(\mathbf{x})=\mathbf{x}$).
>
> > Algorithm 1: Is the notation $\mathcal{G}$ really needed here? Everything of essence seems to be characterized by the assumptions made on the noisy query response, unless I'm missing something.
>
> $\mathcal{G}$ stands for the querying algorithm that has built in the point of
>
> > Misc: The regret bounds on page 6 (and onward) are for "$R_{\alpha}^i$"; this should be $\mathcal{R}_{\alpha}^i$, correct?
>
> We agree, and have changed $R_{\alpha}^i$ to $\mathcal{R}_{\alpha}^i$.
>
> (the end)

---

### Review · Reviewer_8F2k · 2025-05-19

**Summary Of Contributions:**

This paper studies online, decentralized non-convex optimization. Specifically, functions with a property termed "upper-linearizable" are considered. A projection-free algorithm for decentralized regret minimization of such functions is proposed using semi-bandit (i.e. gradient) feedback. This algorithm is then generalized to other settings, e.g. bandit/zeroth-order.

**Audience:**

Yes

**Claims And Evidence:**

No

**Requested Changes:**

See above.

**Strengths And Weaknesses:**

**Meta-concerns:**

I found this paper hard to read, and it is not clear to me that it is a good fit for TMLR. I will elaborate on this below, but I have three overarching concerns:
1. **Lack of Motivation and Context:** Why should the reader care about the various classes of functions mentioned in the introduction? A range of practical applications are mentioned, but none of them are elaborated on. I recommend the authors pick a single application (e.g. mean-field inference) and explain clearly why the objective function in this case has (some subset of) the properties continuous, adversarial, $\gamma$-weakly up-concave, submodularity. Relatedly, I recommend the authors highlight an application where the decentralized nature of the problem is essential.
2. **Lack of experiments:** Theory in ML is important, but in my opinion it needs to feed back into practice. Neither this paper nor the related work [1] contain implementations of the proposed algorithms, or any benchmarking against existing algorithms. I would like to see some of this.
3. **Writing, Length, Number of Results:** In the contributions, the authors mention that this paper proposes 10 new algorithms, 9 of which are novel. However, the majority of the proofs relating to these algorithms (as well as all save one of these algorithms) are in the appendix. Because this is a theory paper, these proofs should not be considered supplementary; they need to be thoroughly checked for correctness. Thus, this feels like an attempt to circumvent the page limit of TMLR. I would suggest the authors consider whether this work is more appropriate for a venue like JMLR, which has a longer page limit and a longer review period.

-----
**Strengths:**

 1. Assuming correctness of the proofs, this work is a formidable work of theory, improving upon prior results in a number of ways.

----
**Comments and Questions:**

0. I would recommend choosing a different first sentence. Unless the reader is already acquainted with adversarial, $\gamma$-weakly up-concave functions, this is unlikely to be compelling.
1. Important definitions (up-concave, $\gamma$-weakly up-concave, DR-submodular) are hidden in the appendix. These need to be in the main text. See [2] for an example of how to do this succinctly.
2. There is significant overlap between the introduction of this work and that of [1] (some sentences are nearly identical). I recommend the authors address this by rewriting their introduction.
3. The link between work cited as motivation for the projection-free approach (Lacoste-Julien et al 2013, Chandrasekaran et al 2009, Hazan & Luo 2016) and the current setting is tenuous. Can the authors provide an example of a problem involving an upper-linearizable function class for which the cost of projection is the computational bottleneck? I note that an almost identical sentence occurs in [6]. I strongly recommend the authors reconsider their phrasing here.
4. Regarding Section 3:
    - Define clearly what sort of object $\mathcal{F}$ is. I guess it is a function class, but more detail would be nice.
    - Why use $\alpha$ regret instead of the simpler and more intuitive notion of regret? What role does the choice of $\alpha$ play?
    - "the adversary chooses a loss function $f_{t.i} \in \mathcal{F}$ **and a query oracle for $f_{t,i}$**" is confusing. I'm guessing the adversary can't switch between, say semi-bandit and bandit feedback at different rounds. That is, the adversary's choice of query is constrained.
    - In the definition of upper linearizable, what are the quantifiers? Does this need to hold for all $\mathbf{x},\mathbf{y} \in \mathcal{K}$? Also, is this $\alpha$ the same as that in $\alpha$-regret? Immediately after this definition, you state that $\mathfrak{q} = \mathfrak{g}$. But this doesn't make sense for bandit feedback, as in this case $\mathfrak{q}$ should be scalar-valued, but $\mathfrak{g}$ takes values in $\mathbb{R}^d$.
    - It is stated that the query function $\mathfrak{g}$ is assumed to be Lipschitz continuous. Do you mean Lipschitz in its second argument (the one taking input from $\mathcal{K}$)?
    - "$\mathcal{F}$ is $M_1$-Lipschitz." Do you mean that all functions within this class are $M_1$ Lipschitz?
    - In Lemma 1, what is the role of $\mathbf{x}_0$?
    - Within algorithm 1, there are two subroutines that seem hard to implement in practice: computing $h$ and computing $\mathcal{O}_{IP}$.  Are there classes of upper-linearizable functions for which $h$ is known, and not the identity? How hard is it to implement the infeasible projection oracle?
 5. In Section 4, at the top of pg. 7 the definition of DR-submodular is referred to. But, you have not defined what $\mathbf{x} \leq \mathbf{y}$ means. This relates to comment 1.
 6. In Section 5.1 $\mathcal{D}$ and $\mathcal{K}$ are used interchangeably. I suggest choosing one and sticking with it.
 7. The sliced one-point gradient estimator is interesting. I'm familiar with this in the unconstrained case ($\mathcal{K} = \mathbb{R}^d$) e.g. [3]. Could you elaborate on or provide a reference for, the equation given at the top of pg. 9. It would also be worth mentioning work of Flaxman and Zinkevich [4,5] here.

-----
**References:**

[1] _From linear to linearizable optimization: A novel framework with applications to stationary and non-stationary dr-submodular optimization_ Pedramfar and Aggarwal, 2024.

[2] _Communication-Efficient Decentralized Online Continuous DR-Submodular Maximization_ Zhang et al, 2023.

[3] _Random gradient-free minimization of convex functions_ Nesterov & Spokoiny 2017.

[4] _Online convex optimization in the bandit setting: gradient descent without a gradient_ Flaxman et al, 2004.

[5] _Online convex programming and generalized infinitesimal gradient ascent_ Zinkevich, 2003.

[6] _Improved projection-free online continuous submodular maximization_ Liao et al, 2023.

---

> ### Author Response · Authors · 2025-06-04
> **Response 1 to Reviewer 8F2k**
>
> Thank you for taking the time to review our paper. We have revised paper based on your comments, uploaded the revised version for your review, and will address your questions in the following:
>
> > **Lack of Motivation and Context:** Why should the reader care about the various classes of functions mentioned in the introduction? A range of practical applications are mentioned, but none of them are elaborated on. I recommend the authors pick a single application (e.g. mean-field inference) and explain clearly why the objective function in this case has (some subset of) the properties continuous, adversarial, $\gamma$-weakly up-concave, submodularity. Relatedly, I recommend the authors highlight an application where the decentralized nature of the problem is essential.
>
> Based on the comment, we have inserted this sentence into the first paragraph of our introduction to focus on the application in mean-field inference:
> "A prototypical example is **mean-field variational inference on edge devices** (Bian et al., 2019), where geographically dispersed sensors cooperatively maximize the evidence lower bound (ELBO) without sharing raw data.
> The ELBO in this setting is a \emph{continuous DR-submodular objective on the probability simplex}, therefore $\gamma$-weakly up-concave, whose curvature coefficients shift adversarially as fresh observations arrive, making a decentralized, online solution essential."
>
> > **Lack of experiments:** Theory in ML is important, but in my opinion it needs to feed back into practice. Neither this paper nor the related work [1] contain implementations of the proposed algorithms, or any benchmarking against existing algorithms. I would like to see some of this.
>
> While empirical validation is important, the contribution of this work is mainly theoretical, and we encourage future exploration in downstream application. We have provided our proof in details and in length, and the mathematical rigorous deduction has laid out the technical correctness for our results. In regards to benchmarking, we have compared our regret bound against other literature in Table 1.
>
> > **Writing, Length, Number of Results:** In the contributions, the authors mention that this paper proposes 10 new algorithms, 9 of which are novel. However, the majority of the proofs relating to these algorithms (as well as all save one of these algorithms) are in the appendix. Because this is a theory paper, these proofs should not be considered supplementary; they need to be thoroughly checked for correctness. Thus, this feels like an attempt to circumvent the page limit of TMLR. I would suggest the authors consider whether this work is more appropriate for a venue like JMLR, which has a longer page limit and a longer review period.
>
> Based on the comment, we have moved the Algorithm tables to the main text in the revision. We further note that it is typical in theoretical papers to have detaild proof in the Appendix. For your reference, we point a few recent TMLR papers which focus on theory with a similar structure with proofs in the Appendix: [1,2,3].
>
> [1] *Online Bandit Nonlinear Control with Dynamic Batch Length and Adaptive Learning Rate*, Kim \\& Lavaei, 2025.
>
> [2] *Group Fair Federated Learning via Stochastic Kernel Regularization*, Arif et al., 2025.
>
> [3] *Studying Exploration in RL: An Optimal Transport Analysis of Occupancy Measure Trajectories*, Nkhumise et al., 2025
>
> > 0. I would recommend choosing a different first sentence. Unless the reader is already acquainted with adversarial, $\gamma$-weakly up-concave functions, this is unlikely to be compelling.
>
> Based on the comment, we have changed the first sentence to: "Modern machine-learning systems increasingly face streaming, non-convex objectives that evolve with incoming data and require optimization in a decentralized fashion." and we have revised the rest of the first paragraph accordingly as indicated in our response to the first Meta-Concern: Lack of Motivation and Context.
>
> (to be continued)

---

> > ### Comment · Reviewer_8F2k · 2025-06-10
> > **Response to authors**
> >
> > Thanks for your thorough responses, and edits to your manuscript. The introduction is indeed much improved! Thanks also for the references papers [1--3]. I note that 2 and 3 have "long submission" listed as their submission length, not "regular submission".  All three have extensive experimental sections.

---

> > > ### Author Response · Authors · 2025-06-13
> > > **Response 5 to Reviewer 8F2k**
> > >
> > > Thank you for getting through our response so promptly and your comments. Here is our response:
> > >
> > > > I note that 2 and 3 have "long submission" listed as their submission length, not "regular submission". All three have extensive experimental sections.
> > >
> > > We note that our revision in this and previous rounds have brought the paper above 12 pages, and thus have marked this paper as a long paper in this revision.
> > >
> > > While empirical validation is important, the contribution of this work is mainly theoretical, and we encourage future exploration in downstream application. We have provided our proof in details and in length, and the mathematical rigorous deduction has laid out the technical correctness for our results. In regards to benchmarking, we have compared our regret bound against other literature in Table 1. While we agree that the previous mentioned papers had experiments, these are not directly related to DR-submodular optimization. In the following, we have three references in TMLR that do not have experiments [4,5,6].
> > >
> > > [1] Online Bandit Nonlinear Control with Dynamic Batch Length and Adaptive Learning Rate, Kim \& Lavaei, 2025.
> > >
> > > [2] Group Fair Federated Learning via Stochastic Kernel Regularization, Arif et al., 2025.
> > >
> > > [3] Studying Exploration in RL: An Optimal Transport Analysis of Occupancy Measure Trajectories, Nkhumise et al., 2025
> > >
> > > [4] Mathematical Characterization of Better-than-Random Multiclass Models, Sébastien Foulle, https://openreview.net/pdf?id=VdW9SkALSd
> > >
> > > [5] Wasserstein Distributionally Robust Policy Evaluation and Learning for Contextual Bandits, Shen et al., https://openreview.net/pdf?id=NmpjDHWIvg
> > >
> > > [6] Improved Regret Bounds for Linear Adversarial MDPs via Linear Optimization, Kong et al., https://openreview.net/pdf?id=KcmWZSk53y

---

> ### Author Response · Authors · 2025-06-04
> **Response 2 to Reviewer 8F2k**
>
> (following response 1)
>
> > 1. Important definitions (up-concave, $\gamma$-weakly up-concave, DR-submodular) are hidden in the appendix. These need to be in the main text. See [2] for an example of how to do this succinctly.
>
> Based on the comment, we have re-organized our Section 3: Preliminary into three parts: Notations and Definitions, Upper-Linearizable functions, and Problem Formulation. In "Notations and Definitions", we have included the definitions (up-concave, $\gamma$-weakly up-concave, DR-submodular) as follows:
> "For two vectors $\mathbf{x},\mathbf{y}\in\mathcal{K}$...";
> "Given $0 < \gamma \leq 1$, we say...";
> "A differentiable function $f : \mathcal{K} \to \mathbb{R}$ is..."
>
> > 2. There is significant overlap between the introduction of this work and that of [1] (some sentences are nearly identical). I recommend the authors address this by rewriting their introduction.
>
> Based on the comment, we have revised the first paragraph as:
> "One such challenge is handling adversarial continuous $\gamma$-weakly up-concave functions, which includes Diminishing-Return-Submodular (or DR-submodular) and concave functions.
> This problem can be formulated as an online interaction: at each time step, the learner selects an action from a convex domain, after which a reward function—satisfying $\gamma$-weak up-concavity—is revealed or partially observed. Depending on the feedback model, the learner may receive full gradient information, only the function value at the selected point, or noisy bandit-style responses. ... Such formulations also emerge in a wide array of other applications, including price optimization, inventory management, recommendation systems, and power network reconfiguration (Aldrighetti et al., 2021; Mishra et al., 2017; Ito & Fujimaki, 2016; Hassani et al., 2017; Mitra et al., 2021; Gu et al., 2023; Pedramfar
> et al., 2023; 2024)."
>
> > 3. The link between work cited as motivation for the projection-free approach (Lacoste-Julien et al 2013, Chandrasekaran et al 2009, Hazan & Luo 2016) and the current setting is tenuous. Can the authors provide an example of a problem involving an upper-linearizable function class for which the cost of projection is the computational bottleneck? I note that an almost identical sentence occurs in [6]. I strongly recommend the authors reconsider their phrasing here.
>
> Based on the comment, we have revised this paragraph as follows:
> "Existing methods for minimizing $\alpha$-regret in DR-submodular optimization typically fall into two broad classes. One class (Chen et al., 2018b; Zhang et al., 2022)  uses projection-based strategies which involve computing projections onto the feasible set at each round, while the other class (Chen et al., 2018a;b; Zhang et al., 2019; Liao et al., 2023)  leverages projection-free techniques which replace the projection step with linear optimization oracles to update decisions more efficiently.
> In certain machine learning tasks (Hazan & Luo, 2016; Lacoste-Julien et al., 2013), the computational cost of projection onto a complex feasible set can substantially exceed that of linear optimization. For example, in matrix completion over nuclear norm balls (Chandrasekaran et al., 2009), computing a projection requires full singular value decomposition, which is cubic in matrix size, whereas linear optimization only requires computing a leading singular vector pair. Notably, many of these tasks involve objective functions that either possess DR-submodular property or admit concave surrogates, both of which fall within the broader class of upper-linearizable functions. This connection motivates us to provide projection-free approaches to the more general upper-linearizable functions under decentralized online setting."
>
> We also note that the notion of upper-linearizable is quite new as it was introduced in the literature in the last year. Although such exploration has not been done specifically for upper linearizable functions, we observe similar pattern in DR-submodular functions as illustrated in the given literatures. Thus, as DR-submodular functions are special cases of upper-linearizable functions, it is safe to assume such pattern would persist among upper-linearizable objectives. Thank you for pointing this out and we rephrase it to highlight our focus.
>
> > 4. Regarding Section 3:
>
> > Define clearly what sort of object $\mathcal{F}$ is. I guess it is a function class, but more detail would be nice.
>
> We have added  "A function class $\mathcal{F}$ is a set of real-valued functions." in **Notations and Definitions** and in Assumption 2 we formally state that "We assume all functions $f\in\mathcal{F}:\mathcal{K}\to \mathbb{R}$ are $M_1$-Lipschitz continuous, differentiable and upper-linearizable with $\alpha, \beta,\mathfrak{g}$ and $h$ as described in Equation2."
>
> (to be continued)

---

> > ### Author Response · Authors · 2025-06-04
> > **Response 3 to Reviewer 8F2k**
> >
> > (following response 2)
> >
> > > Why use $\alpha$ regret instead of the simpler and more intuitive notion of regret? What role does the choice of $\alpha$ play?
> >
> > We note that the problem is NP-hard even in the centralized offline setup.
> > For example, when $f$ is a monotone continuous DR-submodular function and $\mathcal{K} \subseteq [0,1]^d$ contains the origin, finding a point $\mathbf{x} \in \mathcal{K}$ such that $f(\mathbf{x})$ is optimal is NP-hard.
> > More generally, for any $\epsilon > 0$, finding any point $\mathbf{x} \in \mathcal{K}$ such that $f(\mathbf{x})$ is at least $(1 - e^{-1} + \epsilon)$ times the optimal value is NP-hard. (See Proposition 3 in [1])
> > However, there are polynomial times algorithms that achieve the ratio of $1 - e^{-1}$.
> > The ratio with this property is refered to as the *optimal approximation ratio*, in this case being $1 - e^{-1}$.
> > In the three settings considered in this paper, (in the special case of $\gamma=1$) the approximation ratios for monotone function over convex sets containing the origin and non-monotone functions over general convex sets are known to be optimal (See [4] and [5])
> > In the case of monotone functions over general convex sets (when $\gamma=1$), the approximation ratio of $1/2$ is the best known in the literature so far and it is conjectured to be optimal (See [6])
> >
> > We note that the notion of approximation ratio dates further back than the works we mentioned.
> > For example, optimization of discrete submodular setting is more widely studied than its continuous counterpart and, for monotone set functions with cardinality constraints, it has been proven by [7] that finding $(1 - e^{-1} + \epsilon)$ times the optimal value is NP-hard, while a greedy algorithm achieves the approximation ratio of $1 - e^{-1}$.
> >
> > We note that this $\alpha$ is intended to be the same in the definition of upper linearizable function, since both indicate the approximation coefficient. With the $\alpha$ in the current place in the definition of linearizable function, we find $\alpha$-regret. Thus, both are meant to have same meanings, which justifies the use of the same variable.
> >
> > [4] *Guaranteed Non-convex Optimization: Submodular Maximization over Continuous Domains*, Bian et al., 2017
> >
> > [5] *Resolving the Approximability of Offline and Online Non-monotone DR-Submodular Maximization over General Convex Sets*, Mualem & Feldman, 2023.
> >
> > [6] *A Unified Approach for Maximizing Continuous DR-submodular Functions*, Pedramfar et al., 2023.
> >
> > [7] *A threshold of ln n for approximating set cover*, Feige, 1998.
> >
> > > "the adversary chooses a loss function $f_{t,i}\in \mathcal{F}$ and a query oracle for $\mathcal{F}$" is confusing. I'm guessing the adversary can't switch between, say semi-bandit and bandit feedback at different rounds. That is, the adversary's choice of query is constrained.
> >
> >
> > Yes, this is correct. In the same game, the adversary cannot switch between different feedback types between rounds. For instance, if the game is played in bandit setting, we require the adversary to always provide bandit feedback in all rounds of the game. If the game is played in semi-bandit setting, we require the adversary to always provide semi-bandit feedback in all rounds of the game.
> >
> > > In the definition of upper linearizable, what are the quantifiers? Does this need to hold for all $\mathbf{x},\mathbf{y}\in\mathcal{K}$? Also, is this $\alpha$ the same as that in  $\alpha$-regret? Immediately after this definition, you state that $\mathfrak{q}=\mathfrak{g}$. But this doesn't make sense for bandit feedback, as in this case $\mathfrak{q}$ should be scalar-valued, but $\mathfrak{g}$ takes values in $\mathbb{R}^d$.
> >
> > Yes, the condition should hold for any $\mathbf{x},\mathbf{y} \in \mathcal{K}$ and we have added to the new version.
> >
> > The $\alpha$ should be the same in the upper-linearizable function and the regret notion. In the new version, we first introduce upper-linearizable then introduce $\alpha$-regret. The notion of $\alpha$ in the upper-linearizable will give way to $\alpha$-approximation, and thus giving $\alpha$-regret in the online setups.
> >
> > We note that $\mathfrak{q}$ and $\mathfrak{g}$ are both scalar in the bandit setting. We clarify $\mathfrak{q}$ and $\mathfrak{g}$ with a simple bandit example.
> > Let us assume that there exists a class of upper linearizable functions $f$ with $\mathfrak{g}(f,\mathbf{x})=f(\mathbf{x})$. Thus, we query the oracle with $\mathfrak{q}(f,\mathbf{x}) = \mathfrak{g}(f,\mathbf{x})=f(\mathbf{x})$ at point $\mathbf{x}$, and the oracle will give a noisy response $\tilde{\mathfrak{q}}(f,\mathbf{x})$ of with expectation being $\mathfrak{q}(f,\mathbf{x})=f(\mathbf{x})$. The oracle is zeroth-order and the query is trivial (at point $\mathbf{x}$), thus the feedback is bandit. Since we assumed $f\in\mathcal{F}:\mathcal{K}\to\mathbb{R}$ (Assumption 2), $f(\mathbf{x})$ is a scalar vlaue, both $\mathfrak{q}$ and $\mathfrak{g}$ would take values in the scalar space.
> >
> > (to be continued)

---

> > > ### Comment · Reviewer_8F2k · 2025-06-10
> > > **Response to author response.**
> > >
> > > In the definition of upper linearizable, there is an inner product between $\mathfrak{g}(f,x)$ and $y - x$. So, $\mathfrak{g}(f,x)$ needs to be of the same dimension as $y-x$; it cannot be a scalar. So, I don't see how $f$ can be upper linearizble with $\mathfrak{g}(f,x) = f(x)$ unless $x \in \mathbb{R}$ too, which is uninteresting.
> > >
> > > In trying to understand Algorithm 6, I encountered further confusion. It is stated that a query algorithm $\mathcal{G}$ is required as input. In appendix E, that $\mathcal{G}(x)$ queries $\nabla f$. So, doesn't require first-order information?

---

> > > > ### Author Response · Authors · 2025-06-13
> > > > **Response 6 to Reviewer 8F2k**
> > > >
> > > > (following response 5)
> > > >
> > > > > In the definition of upper linearizable, there is an inner product between $\mathfrak{g}(f,\mathbf{x})$ and $\mathbf{y} - \mathbf{x}$. So, $\mathfrak{g}(f,\mathbf{x})$ needs to be of the same dimension as $\mathbf{y} - \mathbf{x}$; it cannot be a scalar. So, I don't see how $f$ can be upper linearizble with $\mathfrak{g}(f,\mathbf{x})=f(\mathbf{x})$ unless $\mathbf{x}\in \mathbb{R}$ too, which is uninteresting.
> > > >
> > > > Thank you for clarifying this. We understand the question, and noted there were some notation issues that we have fixed in this revision. We note that "$\mathfrak{q}=\mathfrak{g}$" holds for the first order setting, which is the case our main algorithm considered.
> > > > To address this in the revised version, we removed the "$\mathfrak{q}=\mathfrak{g}$" statement from the Assumption, and revised the problem formulation to show that this is only for the first order setting.
> > > > We mentioned the detail of query function where "For the upper-linearizable function, we use $\mathbf{w}_ t^i$ such that $\mathbb{E}[\nabla f(\mathbf{w}_ t^i)] = \mathfrak{g}(f_ {t,i},\mathbf{x}_ t^i)$". Further, we have for the query oracle that
> > > > "...returns a noisy response $\mathbf{o}_ {t,i}=\tilde{\mathfrak{q}}(\mathbf{w}_ t^i)$ with mean being either $\nabla f_ {t,i}(\mathbf{w}_ t^i)$ (first order feedback)...".
> > > >
> > > > For the zeroth order setting, such as the case considered in Section 5.1, 5.3, and 5.4, $\mathfrak{q}$ is a scalar while $\mathfrak{g}$ is a vector.
> > > > Thank you for pointing this out and we have revised the writing accordingly.
> > > >
> > > >
> > > > >  In trying to understand Algorithm 6, I encountered further confusion. It is stated that a query algorithm $\mathcal{G}$ is required as input. In Appendix E, that $\mathcal{G}$ queries $\nabla f$. So, doesn't require first-order information?
> > > >
> > > > We understand the confusion and have addressed this by changing the notations of query function and query oracle and making them more explicit.
> > > >
> > > > Algorithm 6 is Algorithm 4 in the revision. In the previous version, the query algorithm was comprised of two parts: select a point to query, and query the oracle (in this case, the $\nabla f$ you mentioned) for a response.
> > > > However, in the zeroth order setting, we were not running the "querying" part of the old $\mathcal{G}$; instead, we were using the "selecting" part and then we query another zero order oracle at the point of query selected by $\mathcal{G}$.
> > > >
> > > > In the revised version, we separate it into two parts: a query *function* $\mathcal{G}$ that returns a point of query, and an actual query *oracle* $\mathcal{Q}$ to give response of the query.
> > > > For this case specifically, the query function $\mathcal{G}$ would return a point for query, which is then used to query the "zeroth order oracle" $\mathcal{Q}$, which is zeroth order.
> > > >
> > > > Accordingly, we have revised the query function examples we have given in Appendix A.2 and A.3: we removed the "first order oracle" from Input and we removed "query the oracle" in the response. Please see the revised version for details.

---

> ### Author Response · Authors · 2025-06-04
> **Response 4 to Reviewer 8F2k**
>
> (following response 3)
>
> > It is stated that the query function $\mathfrak{g}$ is assumed to be Lipschitz continuous. Do you mean Lipschitz in its second argument (the one taking input from $\mathcal{K}$)?
>
> Yes, we make it more clear by stating that $\mathfrak{g}$ is Lipschitz continuous with respect to the second argument, $\mathbf{x}$ in Assumption 4.
>
> > "$\mathcal{F}$ is $M_1$-Lipschitz." Do you mean that all functions within this class are $M_1$-Lipschitz?
>
> Yes, we make it more clear by rephrasing it in Assumption 2 as stated above.
>
> > In Lemma 1, what is the role of $\mathbf{x}_0$?
>
> In Lemma 1, $\mathbf{x}_0$ is a feasible point to the constrainted set $\mathcal{K}$ (note that in the statement we assume $\mathbf{x}_0 \in \mathcal{K}$), and serves as an anchor for the infeasible projection. Otherwise, the algorithm would only produce an infeasible point $\tilde{\mathbf{y}}$ and unable to produce a feasible point $\mathbf{x}$ that is reasonably close.
>
> > Within algorithm 1, there are two subroutines that seem hard to implement in practice: computing $h$ and computing $\mathcal{O}_{IP}$. Are there classes of upper-linearizable functions for which $h$ is known, and not the identity? How hard is it to implement the infeasible projection oracle?
>
> For an example of non-identity $h(\cdot)$, we refer to Appendix A.3, nonmonotone DR-submodular functions over general convex set. In this case, $h(\mathbf{x})=\frac{\mathbf{x}+\underline{\mathbf{x}}}{2}$, where $\underline{\mathbf{x}}\in \mathcal{K}$ is a constant chosen by us. Thus, it is not computationally complex. For the other two cases, $h(\cdot)$ is identity function.
>
>
> For the infeasible projection, the oracle we are using are similar to that being used in [2]. This infeasible projection only solve linear optimization problem, and is less complex than using  projection operation which requires solving quadratic optimization problem. Thus, it is more efficient than performing a projection operation.
>
>
>
> > 5. In Section 4, at the top of pg. 7 the definition of DR-submodular is referred to. But, you have not defined what $\mathbf{x}\leq\mathbf{y}$ means. This relates to comment 1.
>
> The comparison is defined element-wise, and we have added "For two vectors $\mathbf{x}, \mathbf{y} \in \mathcal{K}$, we say $\mathbf{x} \leq \mathbf{y}$ if every element in $\mathbf{x}$ is less than or equal to the corresponding element in $\mathbf{y}$." in Section 3 along with our definition for DR-submodular.
>
> > 6. In Section 5.1 $\mathcal{D}$ and $\mathcal{K}$ are used interchangeably. I suggest choosing one and sticking with it.
>
> Thanks for the suggestion. In the revision, we have changed all $\mathcal{D}$ to $\mathcal{K}$.
>
> > 7. The sliced one-point gradient estimator is interesting. I'm familiar with this in the unconstrained case ($\mathcal{K}=\mathbb{R}^d$) e.g. [8]. Could you elaborate on or provide a reference for, the equation given at the top of pg. 9. It would also be worth mentioning work of Flaxman and Zinkevich [9,10] here.
>
> Thank you for pointing this out.
>
> The sliced one-point gradient esimator is a variant of the classical one-point gradient estimator that allows us to work in settings where we might need to consider lower dimensional subspaces.
> In this work, we tried to consider the setting in full generality and do not assume that that constraint set contains a $d$-dimensional ball.
> Thus, we are not able to use the classical version of the one-point gradient estimator.
>
> We have added a footnote to cite the more well known version of this equality: "When $k=d$ and therefore $\mathcal{L}_0 = \mathbb{R}^d$, this equality is well known, e.g. see Nemirovskii & IUdin, 1983; Flaxman et al., 2005.
> The more general case where $k \leq d$ may be found in Remark 4 in Pedramfar et al., 2023."
>
> [8]Random gradient-free minimization of convex functions Nesterov & Spokoiny 2017.
>
> [9] Online convex optimization in the bandit setting: gradient descent without a gradient Flaxman et al, 2004.
>
> [10] Online convex programming and generalized infinitesimal gradient ascent Zinkevich, 2003.

---

### Review · Reviewer_hqzv · 2025-05-26

**Summary Of Contributions:**

This paper studies the problem of decentralized projection-free optimization with upper-linearizable loss functions. For this problem, the authors obtained an interpolated guarantee of $O(T^{1-\theta/2})$ regret, with $O(T^\theta)$ communication complexity and $O(T^{2\theta})$ calls of linear optimization oracle calls.

This works investigate various problem setups, including whether the functions are monotone or not, whether the origin is contained in the feasible domain, and whether the feedback is full-information or bandit.

**Audience:**

Yes

**Broader Impact Concerns:**

N/A.

**Claims And Evidence:**

Yes

**Requested Changes:**

Please refer to the comments in the 'Strengths And Weaknesses' part.

**Strengths And Weaknesses:**

## Strengths
This work achieves various new results for the problem of decentralized projection-free optimization with upper-linearizable loss functions. Furthermore, the results in this work are interpolated, which can recover previous best-known guarantees as special cases.

## Weaknesses
1.  I am confused about why this paper focuses on the $\alpha$-regret, where $\alpha \in (0,1)$. As we all know, the standard notion of regret does not include such an $\alpha$. Therefore, importing an extra $\alpha$ in the regret definition seems to be a little weird. I guess the reason may be that the definition of upper-linearizable function contains the same $\alpha$, and importing a corresponding $\alpha$ in the regret definition would make the analysis easier? And I suggest that the authors could provide some explanations about this $\alpha$ in the next version.

2. In Sec 3, the authors listed multiple assumptions, e.g., 'We further assume that the query function $g(f_{t,i}, \mathbf{x})$ is $L_1$-Lipschitz. We assume that $\mathcal{F}$ is $M_1$-Lipschitz continuous and first order query oracles for $\mathcal{F}$ are bounded by $G$'. It seems that there are many assumptions only hidden in the text, making it hard to find out. I suggest that the authors could use formal environments for assumptions to make readers clearer about how many assumptions are required in this paper.

3. The concept of 'up-concave' is one of the most central notions of this paper. However, surprisingly, I do not find its formal definition in the main paper (except for the one in the third contribution). The formal definition of it is given in the appendix, which seems inappropriate. I suggest that the authors could move its definition into the main paper.

4. This paper claims 'projection-freeness' as one of the main contributions, as suggested in the title. However, if I understand the paper correctly, projection-freeness is achieved by using a simple gradient ascent update (Line 10 in Alg 1) with the infeasible projection, which is implemented by LOO calls. The former is standard, and the latter comes from the work of Liao et al. [2023]. As a result, I think achieving projection-freeness should not be claimed as one of the main contributions of this paper. Otherwise, it would lead to misunderstandings of the true contributions.

5. Could the authors provide some intuitive explanations of the upper-linearizable functions? This concept seems to be a little artificial. Besides, to validate the generality of this function class, it is suggested that the authors could use more space in the main paper to show how upper-linearizable functions can cover some of our well-known function classes as special cases.

6. Notice that there is an $h(\cdot)$ in the definition of upper-linearizable functions. Meanwhile, the algorithms require the knowledge of $h(\cdot)$ to give their submissions. Is this requirement hard to satisfy in some real-world applications? This issue coincides with the last one. I suggest that the authors could provide some explanations about what $h(\cdot)$ is when recovering some well-known function classes.

## Minor issues
1. It seems that the notion of `$F$' is not defined or explained.

2. In the second contribution, it should be 'tradeoff' but not 'tradeofff'.

---

> ### Author Response · Authors · 2025-06-04
> **Response 1 to Reviewer hqzv**
>
> Thank you for taking the time to review our paper. We have revised paper based on your comments, uploaded the revised version for your review, and will address your questions in the following:
>
> > I am confused about why this paper focuses on the $\alpha$-regret, where $\alpha\in(0,1)$. As we all know, the standard notion of regret does not include such an $\alpha$. Therefore, importing an extra $\alpha$ in the regret definition seems to be a little weird. I guess the reason may be that the definition of upper-linearizable function contains the same $\alpha$, and importing a corresponding $\alpha$ in the regret definition would make the analysis easier? And I suggest that the authors could provide some explanations about this $\alpha$ in the next version.
>
> We note that the problem is NP-hard even in the centralized offline setup.
> For example, when $f$ is a monotone continuous DR-submodular function and $\mathcal{K} \subseteq [0,1]^d$ contains the origin, finding a point $\mathbf{x} \in \mathcal{K}$ such that $f(\mathbf{x})$ is optimal is NP-hard.
> More generally, for any $\epsilon > 0$, finding any point $\mathbf{x} \in \mathcal{K}$ such that $f(\mathbf{x})$ is at least $(1 - e^{-1} + \epsilon)$ times the optimal value is NP-hard. (See Proposition 3 in [1])
> However, there are polynomial times algorithms that achieve the ratio of $1 - e^{-1}$.
> The ratio with this property is refered to as the *optimal approximation ratio*, in this case being $1 - e^{-1}$.
> In the three settings considered in this paper, (in the special case of $\gamma=1$) the approximation ratios for monotone function over convex sets containing the origin and non-monotone functions over general convex sets are known to be optimal (See [1] and [2])
> In the case of monotone functions over general convex sets (when $\gamma=1$), the approximation ratio of $1/2$ is the best known in the literature so far and it is conjectured to be optimal (See [3])
>
> We note that the notion of approximation ratio dates further back than the works we mentioned.
> For example, optimization of discrete submodular setting is more widely studied than its continuous counterpart and, for monotone set functions with cardinality constraints, it has been proven by [4] that finding $(1 - e^{-1} + \epsilon)$ times the optimal value is NP-hard, while a greedy algorithm achieves the approximation ratio of $1 - e^{-1}$.
>
> We note that this $\alpha$ is intended to be the same in the definition of upper linearizable function, since both indicate the approximation coefficient. With the $\alpha$ in the current place in the definition of linearizable function, we find $\alpha$-regret. Thus, both are meant to have same meanings, which justifies the use of the same variable.
>
> [1] *Guaranteed Non-convex Optimization: Submodular Maximization over Continuous Domains*, Bian et al., 2017
>
> [2] *Resolving the Approximability of Offline and Online Non-monotone DR-Submodular Maximization over General Convex Sets*, Mualem & Feldman, 2023.
>
> [3] *A Unified Approach for Maximizing Continuous DR-submodular Functions*, Pedramfar et al., 2023.
>
> [4] *A threshold of ln n for approximating set cover*, Feige, 1998.
>
> > In Sec 3, the authors listed multiple assumptions, e.g., 'We further assume that the query function $\mathfrak{g}(f_{t,i},\mathbf{x})$ is $L_1$-Lipschitz. We assume that $\mathcal{F}$ is $M_1$-Lipschitz continuous and first order query oracles for $\mathcal{F}$ are bounded by $G$'. It seems that there are many assumptions only hidden in the text, making it hard to find out. I suggest that the authors could use formal environments for assumptions to make readers clearer about how many assumptions are required in this paper.
>
> Thank you for this comment, in the revised version, we separate Section 3 (Preliminaries) into three parts: Notations and Definitions, Upper-Linearizable Functions, and Problem Formulations. We introduce assumptions formally in assumptions environment after introducing Problem Formulation.
>
> > The concept of 'up-concave' is one of the most central notions of this paper. However, surprisingly, I do not find its formal definition in the main paper (except for the one in the third contribution). The formal definition of it is given in the appendix, which seems inappropriate. I suggest that the authors could move its definition into the main paper.
>
> As described above, we include the definition of up-concave in the new version of Section 3.
>
> (to be continued)

---

> ### Author Response · Authors · 2025-06-04
> **Response 2 to Reviewer hqzv**
>
> (following Response 1)
>
> > This paper claims 'projection-freeness' as one of the main contributions, as suggested in the title. However, if I understand the paper correctly, projection-freeness is achieved by using a simple gradient ascent update (Line 10 in Alg 1) with the infeasible projection, which is implemented by LOO calls. The former is standard, and the latter comes from the work of Liao et al. [2023]. As a result, I think achieving projection-freeness should not be claimed as one of the main contributions of this paper. Otherwise, it would lead to misunderstandings of the true contributions.
>
> Thank you for mentioning this. We included "projection-free" in the title since our algorithm is projection-free, and in fairness, we compare our algorithm against other projection-free algorithm in the field. However, we did not claim anything related to "projection-freeness" as our main contribution, as illustrated at the end of the introduction section.
>
> > Could the authors provide some intuitive explanations of the upper-linearizable functions? This concept seems to be a little artificial. Besides, to validate the generality of this function class, it is suggested that the authors could use more space in the main paper to show how upper-linearizable functions can cover some of our well-known function classes as special cases.
>
> While the current version included examples in Appendix A, we have moved a discussion of that in Section 3 in the revision after the formal definition of upper-linearizable functions.
> So far, the main classes of functions that are covered by upper-linnearizable functions are concave functions and continuous DR-submodular functions (as described in the paper).
> However, we would like to point out that this concept has been introduced in the literature in the paper (Pedramfar & Aggarwal, 2024a) which was published in December 2024. Thus, given more time, we expect that this concept will be better understood and more examples of linearizable functions will be found.
>
>
> > Notice that there is an $h(\cdot)$ in the definition of upper-linearizable functions. Meanwhile, the algorithms require the knowledge of $h(\cdot)$ to give their submissions. Is this requirement hard to satisfy in some real-world applications? This issue coincides with the last one. I suggest that the authors could provide some explanations about what $h(\cdot)$ is when recovering some well-known function classes.
>
> It is not hard to satisfy, as we illustrated in the appendices: for case A.1 and A.2, $h(\mathbf{x})=\mathbf{x}$, while for case A.3,  $h(\mathbf{x})=\frac{\mathbf{x}+\bar{\mathbf{x}}}{2}$ for some constant $\bar{\mathbf{x}}\in\mathcal{K}$.
> In the revised version, we included such detailed discussion of the upper-linearizable functions in Section 3. We have added the corresponding Lemma to the functions, such that the readers can figure out the corresponding $\alpha$, $\beta$, $\mathfrak{g}$ and $h$: "More precisely, this class includes (i) ...(Lemma 2), (ii) ...(Lemma 3), and (iii) ...(Lemma 4). "
> Specifically regarding $h$, we have added: "We note that $h(\cdot)$ takes identity function for case (i) and (ii), while $h(\mathbf{x})=\frac{\mathbf{x}+\bar{\mathbf{x}}}{2}$ for some constant $\bar{\mathbf{x}}\in\mathcal{K}$ for case (iii). We refer the details to Appendix~A."
>
>
> > It seems that the notion of '$\mathcal{F}$' is not defined or explained.
>
> In the revision, we have added definition of $\mathcal{F}$ in Section 3 Notations and Definitions: "Given a set $\mathcal{K}$, we define a function class $\mathcal{F}$ as a subset of all real-valued functions over $\mathcal{K}$." and we formally introduce assumptions about $\mathbf{F}$ in Assumption 2.
>
> > In the second contribution, it should be 'tradeoff' but not 'tradeofff'.
>
> Thanks for pointing out this typo, we fixed it in the revision.

---

### Author Response · Authors · 2025-12-13
**Revised Version**

Please see the revised version of the paper, published in TMLR, at https://openreview.net/forum?id=bZ5WD2HUQr

---

### Decision · Action_Editor_eWgC · 2025-06-26

**Recommendation:** Reject

**Audience:**

Yes

**Audience Explanation:**

Possibly. The topic itself is relevant to TMLR’s audience, especially in the areas of online learning and decentralized optimization. However, the current presentation and lack of clarity may limit the impact and accessibility of the findings.

**Claims And Evidence:**

No

**Claims Explanation:**

Partially. While the theoretical analysis appears sound and the main results are technically justified, the clarity and organization of the presentation limit the overall persuasiveness of the claims. Some important aspects, such as motivation and interpretation of the bounds, could be better explained.